

**Assessing the impacts of reservoirs on the downstream flood frequency by coupling**
**the effect of the scheduling-related multivariate rainfall into an indicator of**
**reservoir effects**
Bin Xiong[1], Lihua Xiong[1*], Jun Xia[1], Chong-Yu Xu[1, 3], Cong Jiang[2], Tao Du[4]
1. State Key Laboratory of Water Resources and Hydropower Engineering Science, Wuhan
University, Wuhan 430072, P.R. China
2. School of Environmental Studies, China University of Geosciences (Wuhan), Wuhan 430074,
China
3. Department of Geosciences, University of Oslo, P.O. Box 1022 Blindern, N-0315 Oslo, Norway
4. Bureau of Hydrology, Changjiang Water Resources Commission, Wuhan 430010, China
*Corresponding author:*
Lihua Xiong, PhD, Professor
State Key Laboratory of Water Resources and Hydropower Engineering Science
Wuhan University, Wuhan 430072, P.R. China
E-mail: xionglh@whu.edu.cn
Telephone: +86-13871078660
Fax: +86-27-68773568



## Abstract:

Many studies have shown that the downstream flood regimes have been significantly altered by
upstream reservoir operation. Reservoir effects on the downstream flow regime are normally carried out
by comparing the pre-dam and post-dam frequencies of some streamflow indicators such as floods and
droughts. In this paper, a rainfall-reservoir composite index (RRCI) is developed to precisely quantify
reservoir impacts on downstream flood frequency under the framework of covariate-based flood
frequency analysis. The RRCI is derived from both the reservoir index (RI) of the previous study and
the joint cumulative probability (JCP) of multiple rainfall variables (i.e., the maximum, intensity,
volume and timing) of multiday rainfall input (MRI), and is calculated by a c-vine copula model. Then,
using RI or RRCI as covariate, a nonstationary generalized extreme value (NGEV) distribution model
with time-varying location and/or scale parameters is developed and used to analyze the annual
maximum daily flow (AMDF) of Ankang, Huangjiagang and Huangzhuang gauging stations of the
Hanjiang River, China with the Bayesian estimation method. The results show that regardless of using
RRCI or RI, nonstationary flood frequency analysis demonstrates that the overall flood risk of the basin
has been significantly reduced by reservoirs, and the reduction increases with the reservoir capacity.
What's more, compared with RI, RRCI through incorporating the effect of the scheduling-related
multivariate MRI can better explain the alteration of AMDF. And for a given reservoir capacity (i.e., a
specific RI), the flood risk (e.g., the Huangzhuang station) increases with the JCP of rainfall variables
and gradually approaches the risk of no reservoir (i.e., RI=0). This analysis, combining the reservoir

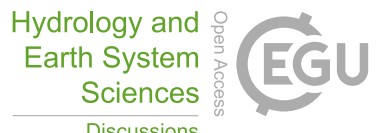

index with the scheduling-related multivariate MRI to account for the alteration in flood frequency,
provides a comprehensive approach and knowledge for downstream flood risk management under the
impacts of reservoirs.
**Keywords**: Nonstationary; reservoir; flood frequency analysis; multiday rainfall; generalized
extreme value distribution; the Hanjiang River

## 1 Introduction

River floods are generated by various complex nonlinear processes involving physical factors
including "hydrological pre-conditions (e.g. soil saturation, snow cover), meteorological conditions (e.g.
amount, intensity, and spatial and temporal distribution of rainfall), runoff generation processes as well
as river routing (e.g. superposition of flood waves in the main river and its tributaries)" (Nied et al.,
2013; Wyżga et al., 2016). In the absence of reservoirs, a nature extreme flow at the site is directly
related to the extreme rainfall in the drainage area. However, after the construction of large or medium-
sized reservoirs, the downstream extreme flow is the result of the reservoir scheduling mainly based on
reservoir capacity and inflow processes. In recent years, because of its importance for the risk
assessment of downstream floods, the study of flood frequency under the impacts of reservoirs has
received increasing attention. For example, Benito and Thorndycraft (2005) reported various significant
changes of the pre- and post-dam hydrologic regimes (e.g., minimum and maximum flows over
different durations) across the United States.





For conventional or stationary frequency analysis to be used in assessing the impact of dams on
downstream flood regimes, a basic hypothesis is that hydrologic time series keeps stationarity, i.e., "free
of trends, shifts or periodicity (cyclicity)" (Salas, 1993). However, in many cases, changing flooding
regimes and the nonstationarity of observed flood series have demonstrated that this strict assumption is
invalid (Kwon et al., 2008; Milly et al., 2008). Nonstationarity in the flood regimes downstream of
dams makes frequency analysis more complicate. Actually, the frequency of floods downstream of
dams is closely related to both climate variability and upstream flood operation. In recent years, there
are a lot of attempts linking flood generating mechanisms and reservoir operation to the frequency of
flood variable (Gilroy and Mccuen, 2012; Goel et al., 1997; Lee et al., 2017; Liang et al., 2017; Su and
Chen, 2018; Yan et al., 2016).
Previous studies have meaningfully increased knowledge related to the reservoir-induced
nonstationarity of downstream hydrological extreme frequency (Ayalew et al., 2013; López and Francés,
2013; Liang et al., 2017; Magilligan and Nislow, 2005; Su and Chen, 2018; Wang et al., 2017; Zhang et
al., 2015). There are two main approaches to incorporate the effects of reservoirs into flood frequency
analysis: the hydrological model simulation approach and the nonstationary frequency modeling
approach. In the first approach, a simulating regulated flood time series for the regulated frequency
analysis is available by three model components, i.e., stochastic rainfall generator the rainfall-runoff
model and the reservoir flood operation module (e.g., reservoir storage capacity, the size of release
structures and operation rules). However, as indicated by Ayalew et al. (2013), many simplifying



assumptions of this method is suitable just for small single reservoirs; for large single reservoirs or
reservoir systems, first approach would fail; besides, a lot of detailed information related to reservoir
flood operation module, especially for reservoir systems, may not be available. For this reason, our
attention is focused on the second method, the nonstationary frequency modeling approach.
Nonstationary distribution models have been widely used to deal with nonstationarity of extreme values.
In nonstationary distribution models, distribution parameters are expressed as the functions of
covariates to determine the conditional distributions of the extreme values. According to extreme value
theory, the maxima series can generally be described by the generalized extreme value distribution
(GEV). Thus, previous studies have used the nonstationary generalized extreme value distribution
(NGEV) to describe nonstationary maxima series. Scarf (1992) modeled the change in the location and
scale parameters of GEV over time through power function relationship. Coles (2001) introduced
several time-dependent structures (e.g., trend, quadratic and change-point) into the location, scale and
shape parameters of GEV. Adlouni et al. (2007) provided a general NGEV model with an improved
parameter estimate method. In recent years, "generalized additive models for location, scale and shape"
(GAMLSS) was widely used in nonstationary hydrological frequency analysis (Du et al., 2015; Jiang et
al., 2014; López and Francés, 2013; Rigby and Stasinopoulos, 2005; Villarini et al., 2009), but GEV is
rarely involved in candidate distributions of GAMLSS. In terms of parameter estimation method for
nonstationary distribution model, the maximum likelihood (ML) method is the most common parameter
estimate method. However, the ML method for NGEV model may diverge when using numerical



techniques to solve likelihood function with small sample. Another drawback of the ML method is that
it is not convenient to describe the uncertainty of the estimators. Adlouni et al. (2007) developed the
generalized maximum likelihood estimation method (GML) and demonstrated that the GML method
has better performance than the ML method in their all cases. Ouarda and El‐Adlouni (2011)
introduced the Bayesian nonstationary frequency analysis. The Bayesian method can directly describe
the uncertainty of extreme flood estimates through providing the prior and posterior distributions of the
shape parameter which controls the tail behavior of the NGEV.

In the nonstationary frequency modeling approach, a dimensionless reservoir index (RI), as an

indicator of reservoir effects, was proposed by López and Francés (2013), and it generally is used as
covariate for expression of distribution parameters (e.g., location parameter) (Jiang et al., 2014; López
and Francés, 2013). Liang et al. (2017) modified the reservoir index by replacing the mean annual
runoff in expression of *RI* with the annual runoff, so that the modified reservoir index can reflect the
impact of reservoirs on flood extremes under different total inflow conditions each year. However, this
improvement doesn't reveal more accurate effects of reservoirs on the downstream floods. In fact, the
effects of reservoirs, especially small or medium-sized reservoirs, may be closely related to not only
static reservoir capacity, but also dynamic reservoir operation associated with the multiday rainfall input
(MRI), not just annual runoff.

Therefore, the aim of the study is to develop an indicator of reservoir effects coupling the RI and

the effect of the scheduling-related multivariate rainfall (SRMR), named the rainfall-reservoir



composite index (RRIC), and then to assess the reservoir effects through a NGEV model with this
indicator as covariate. The specific objectives of this study are: (1) to calculate the RRCI to accurately
quantify reservoir effects; (2) to compare RRCI with RI through the covariate-based flood frequency
analysis; and (3) to quantify downstream flood risks based on the final NGEV model.

## 2 Methods

To quantify the effects of reservoirs on the frequency of the AMDF series, a three-step

framework (Figure 1), termed the covariate-based flood frequency analysis using the RRIC as covariate,
is established. In this section, the methods in this framework are introduced. First, RI of the previous
literature is presented. Second, based on RI, the RRCI is developed through incorporating the effect of
SRMR. And then, the C-vine copula model is used to construct the cumulative distribution function of
MRI variables which is treated as the measure function for the effect of SRMR. Fourth and last, the
NGEV model with the Bayesian estimation is clarified.

### 2.1 Reservoir index (RI)

Intuitively, the larger the reservoir capacity relative to the natural flow of a downstream gauging

station, the greater the effect of the reservoir on streamflow regime is possible. To quantify the
reservoir-induced alteration to streamflow regime, Batalla et al. (2004) proposed the impounded runoff
index (IRI), a ratio of reservoir capacity ($C$) to (unimpaired) mean annual flow ($R_m$), indicated as
$IRI = C / R_m$. For a single reservoir, the IRI is a good indicator of the extent to which the reservoir alters



streamflow. To analysis the effect of multi-reservior system on the alteration of flood frequency, López
and Francés (2013) proposed a dimensionless reservoir index to indicate the effects of reservoirs on the
hydrological regimes in a river. The reservoir index (RI) for a gauging station is defined as

$$RI = \sum_{i=1}^{N}\left(\frac{A_i}{A_T}\right)\cdot\left(\frac{C_i}{R_m}\right) \tag{1}$$

where $N$ is the total number of reservoirs upstream of the gauge station, $A_i$ is the total basin area
upstream of the $i$-th reservoir, $A_T$ is the total basin area upstream of the gauge station, $C_i$ is the water
storage capability of the $i$-th reservoir, and $R_m$ is the mean annual runoff at the gauge station. the Eq. (1)
indicates that for a reservoir system consisting of small and middle sized reservoirs, the RI of a gauging
station is generally less than 1, but for a system with some large reservoirs, e.g., multi-year regulating
storage reservoirs, the RI of the gauging station near this system may be close to 1 or higher. In the
following subsection, we will develop a new index to indicate the more precise effects of reservoirs on
flood regimes.
**2.2 Rainfall-reservoir composite index (RRCI)**
In addition to reservoir capacity, multiday rainfall input (MRI) is a key initial condition for the
scheduling results of the reservoir system. To add the effect of SRMR into the new indicator of
reservoir effects, the multiple scheduling-related variables (denoted as $x_1, x_2, ..., x_d$ ) of the MRI,



corresponding to the inflows to the reservoir which will be regulated to become downstream flood
events, are considered. The extraction procedure of the MRI is detailed in the section 3.2.
We propose a new index called rainfall-reservoir composite index (RRIC) for more
comprehensively assessing effects of reservoirs on floods by coupling the RI and the effects of SRMR,
defined as
$$RRCI = \begin{cases} \left(\bar{F}\left(x_1, x_2, \ldots x_d\right)\right)^{(1/RI-1)}, & 0 < RI \leq 1 \\ RI, & RI > 1 \end{cases} \qquad (2)$$

Where $\bar{F}(\cdot)$ is the measure function for the effects of SRMR. In this study, the OR-joint exceedance
probability $P_{MRI}^{\vee}\left(\bigcup_{i=1}^{d}\left(X_i > x_i\right)\right)$ of the multiple scheduling-related variables of the MRI is used as the
measure function. The closer the value of $P_{MRI}^{\vee}$ is to 0, the greater impact the MRI, inferring that the
reservoir scheduling is more inflexible, so that the resulting downstream flood is possibly greater, and
vice versa. The Eq. (2) can be expressed as
$$RRCI = \begin{cases} \left(P_{MRI}^{\vee}\left(\bigcup_{i=1}^{d}\left(X_i > x_i\right)\right)\right)^{(1/RI-1)} = \left(1 - F\left(x_1, x_2, \ldots x_d\right)\right)^{(1/RI-1)}, & 0 < RI \leq 1 \\ RI, & RI > 1 \end{cases} \qquad (3)$$

Where $F(\cdot)$ is the cumulative distribution function. Figure 2 illustrates the relationship in the Eq. (3),
which shows that the RRCI is conditional on the joint cumulative frequency of scheduling-related
rainfall variables when given the static capability of reservoirs (RI). The expectation of RRCI is as
follow





$$E\left(RRI\right) = \int_{\mathbb{R}^d} \left(1 - F\left(x_1, x_2, \ldots x_d\right)\right)^{(1/RI-1)} dF\left(x_1, x_2, \ldots x_d\right) = RI \tag{4}$$

In addition, for the OR case, we have

$$P_{MRI}^{\vee}\left(\bigcup_{i=1}^{d}\left(X_i > x_i\right)\right) \geq P_{MRI}^{\vee}\left(X_i > x_i\right) \tag{5}$$

The Eq. (3) and Eq. (5) indicate that for the given MRI, RRCI considering the multivariate MRI will be greater than or equal to RRCI considering the univariate MRI. To give a reasonable RRCI, unrelated rainfall variables should not be incorporated. We use four variables (i.e., extremes, intensities, volumes and timings) of the MRI as candidate variables to construct the $d$-dimensional ($d = 1, 2, 3, 4$) distribution for the calculation of RRCI. The identification of the scheduling-related variables from four candidate variables is based on the rank correlation coefficient between RRCI and the magnitude of AMDF. The construction method of $d$-dimensional ($d = 2, 3, 4$) distribution is described in the following subsection.

<Figure 2>

## 2.3 C-vine Copula model

In this subsection, a c-vine Copula model for the construction of continuous $d$-dimensional distribution $F\left(x_1, x_2, \ldots x_d\right)$ is clarified. The Sklar's theorem (Sklar, 1959) showed that for a continuous $d$-dimensional distribution, one-dimensional margins and dependence structure can be separated, and the dependence can be represented by a copula formula as follows

$$F\left(x_1, x_2, \ldots x_d \mid \boldsymbol{\theta}\right) = C\left(u_1, u_2, \ldots, u_d \mid \boldsymbol{\theta}_c\right), u_i = F_{X_i}\left(x_i \mid \boldsymbol{\theta}_i\right) \tag{6}$$





where $F_{X_i}(\cdot)$ is the univariate marginal distribution of $X_i$; $C(\cdot)$ is the copula function. $\boldsymbol{\theta}_c$ is the copula
parameter vector; $\boldsymbol{\theta}_i$ is the parameter vector of the corresponding margins. $\boldsymbol{\theta}=(\boldsymbol{\theta}_c,\boldsymbol{\theta}_1,\boldsymbol{\theta}_2,...,\boldsymbol{\theta}_d)$ is the
parameter vector of the whole *n*-dimensional distribution. Thus, the construction of $F(x_1,x_2,...x_d)$ can
be separated into two steps: first is the modeling of the univariate margins; second is the modeling of
the dependence structure. For the first step, we use the empirical distribution as univariate marginal
distributions. Then, for the second step, the copula construction for dependence modeling is based on
the pair-copula construction method which has been widely used in the previous research (Aas et al.,
2009; Xiong et al., 2015). According to Aas et al. (2009), the joint density function $f(x_1,x_2,...,x_d)$ is
written as
$$f(x_1,x_2,...,x_d|\boldsymbol{\theta})=c_{1...n}(u_1,u_2,...,u_d|\boldsymbol{\theta}_c)\prod_{i=1}^{d}f_{X_i}(x_i|\boldsymbol{\theta}_i),u_i=F_{X_i}(x_i|\boldsymbol{\theta}_i) \tag{7}$$

and the *n*-dimension copula density $c_{1...d}(u_1,u_2,...,u_d)$, which can be decomposed into $d(d-1)/2$
bivariate copulas, corresponding to a c-vine structure, is given by
$$c_{1...d}(u_1,u_2,...,u_d|\boldsymbol{\theta}_c)=\prod_{j=1}^{d-1}\prod_{i=1}^{d-j}c_{j,i+j|1,...,j-1}\left(F(u_j|u_1,...,u_{j-1}),F(u_{i+j}|u_1,...,u_{j-1})\Big|\boldsymbol{\theta}_{j,i|1,...,j-1}\right) \tag{8}$$

where $c_{j,i+j|1,...,j-1}$ is the density function of a bivariate pair copula and $\boldsymbol{\theta}_{j,i|1,...,j-1}$ is a parameter vector of
the corresponding bivariate pair copula. And the marginal conditional distribution is





$$F\left(u_{i+j}\left|u_1,...,u_{j-1}\right.\right)=$$

$$\frac{\partial C_{i+j,j-1|1,...,j-2}\left(F\left(u_{i+j}\left|u_1,...,u_{j-2}\right.\right),F\left(u_{j-1}\left|u_1,...,u_{j-2}\right.\right)\left|\boldsymbol{\theta}_{i+j,j-1|u_1,...,u_{j-2}}\right.\right)}{\partial F\left(u_{j-1}\left|u_1,...,u_{j-2}\right.\right)},$$ (9)

$$j=2,...,d-1;\ i=0,...,n-j$$

where $C_{i+j,j-1|1,...,j-2}$ is a bivariate copula distribution function. The maximum dimensionality covered in
this study is four. Thus for the four-dimension copula (of which the decomposition is shown in Figure
3), the general expression of Eq. (8) is
$$\begin{aligned}c_{1234}\left(u_1,u_2,u_3,u_4\left|\boldsymbol{\theta}_c\right.\right)&=c_{12}\left(u_1,u_2\left|\boldsymbol{\theta}_{12}\right.\right)c_{13}\left(u_1,u_3\left|\boldsymbol{\theta}_{13}\right.\right)c_{14}\left(u_1,u_4\left|\boldsymbol{\theta}_{14}\right.\right)\cdot\\ &c_{23|1}\left(F\left(u_2\left|u_1\right.\right),F\left(u_2\left|u_1\right.\right)\left|\boldsymbol{\theta}_{23|1}\right.\right)c_{24|1}\left(F\left(u_2\left|u_1\right.\right),F\left(u_4\left|u_1\right.\right)\left|\boldsymbol{\theta}_{24|1}\right.\right)\cdot\\ &c_{34|12}\left(F\left(u_3\left|u_1,u_2\right.\right),F\left(u_4\left|u_1,u_2\right.\right)\left|\boldsymbol{\theta}_{34|1}\right.\right)\end{aligned}$$ (10)

<Figure 3>
**2.4 Nonstationary generalized extreme value distribution model with the Bayesian estimation**
The covariate-based extreme frequency analysis is extensively concerned and used (Villarini et
al., 2009; Ouarda and El‐Adlouni, 2011; López and Francés, 2013; Xiong et al., 2018). In this
subsection, the NGEV model based on Bayesian estimation is developed for flood frequency analysis,
following Adlouni et al. (2007) and Ouarda and El‐Adlouni (2011).
Suppose that flood variable $Y_t$ obeys nonstationary distribution $f_{Y_t}\left(y_t\left|\boldsymbol{\eta}_t\right.\right)$ with parameters
$\boldsymbol{\eta}_t=[\mu_t,\sigma_t,\xi_t]$. This allows stationary frequency analysis to be incorporated into our framework,
because the procedure of model selection can identify whether the location, scale and shape parameters





change with the covariates. If they are constants, this process will be stationary frequency analysis. In
this study, the flood estimate of NGEV using the RRCI as covariate will be compared with the one
using the RI. Considering that the shape parameter is sensitive to quantile estimation of rare events
(more details about the shape parameter are referred in the section of parameter estimation), only
location and scale parameters are allowed to vary with covariates. Thus, a nonstationary GEV density
function is given by:
$$f_{Y_t}\left(y_t \mid \mu_t, \sigma_t, \xi_0\right) = \frac{1}{\sigma_t}\left[1 + \xi_0\left(\frac{y_t - \mu_t}{\sigma_t}\right)\right]^{-1/\xi_0 - 1} \exp\left\{-\left[1 + \xi_0\left(\frac{y_t - \mu_t}{\sigma_t}\right)\right]^{-1/\xi_0}\right\}$$
$$-\infty < \mu_t < \infty, \sigma_t > 0, -\infty < \xi_0 < \infty$$
(11)

According to the linear additive formulation of Generalized Additive Models for Location, Scale,
and Shape (GAMLSS) (Rigby and Stasinopoulos, 2005; Villarini et al., 2009), the NGEV models with
different formula of the location and scale parameters are show in Table 1.

<Table 1>

Take the model $\mathrm{NGEV}(\mu_t \sim \mathrm{RRCI}, \sigma_t \sim \mathrm{RRCI})$ as an example, the parameter vector
$\boldsymbol{\theta}_{\mathrm{NGEV}} = \left[\mu_0, \mu_1, \sigma_0, \sigma_1, \xi_0\right]$ is to be estimate. We use the Bayesian method to estimate $\boldsymbol{\theta}_{\mathrm{NGEV}}$. Let the
prior probability distribution be $\pi\left(\boldsymbol{\theta}_{\mathrm{NGEV}}\right)$ and observations $\boldsymbol{D}$ have the likelihood $l\left(\boldsymbol{D} \mid \boldsymbol{\theta}_{\mathrm{NGEV}}\right)$, then
the posterior probability distribution $p\left(\boldsymbol{\theta}_{\mathrm{NGEV}} \mid \boldsymbol{D}\right)$ can be calculated with Bayes' theorem, as follow
$$p\left(\boldsymbol{\theta}_{\mathrm{NGEV}} \mid \boldsymbol{D}\right) = \frac{l\left(\boldsymbol{D} \mid \boldsymbol{\theta}_{\mathrm{NGEV}}\right)\pi\left(\boldsymbol{\theta}_{\mathrm{NGEV}}\right)}{\int_\Omega l\left(\boldsymbol{D} \mid \boldsymbol{\theta}_{\mathrm{NGEV}}\right)\pi\left(\boldsymbol{\theta}_{\mathrm{NGEV}}\right)d\boldsymbol{\theta}_{\mathrm{NGEV}}} \propto l\left(\boldsymbol{D} \mid \boldsymbol{\theta}_{\mathrm{NGEV}}\right)\pi\left(\boldsymbol{\theta}_{\mathrm{NGEV}}\right)$$
(12)

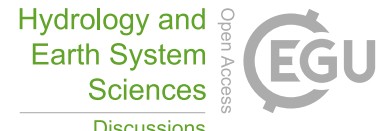

where the integral is the normalizing constant and $\mathbf{\Omega}$ is the whole parameter space. The obvious
difference between the Bayesian method and the frequentist method is that Bayesian method considers
the parameters $\theta_{\text{NGEV}}$ to be random variables, and the desired distribution of the random variables can
be obtained by a Markov chain which can constructed by using various Monte Carlo (MCMC)
algorithms (Reis Jr and Stedinger, 2005; Ribatet et al., 2007) to process Eq. (12). And in this study, we
use the Metropolis-Hastings algorithm (Chib and Greenberg, 1995; Viglione et al., 2013), which can be
done by aid of the R package "MHadaptive" (Chivers, 2012). We use a beta distribution function with
the parameters $u = 6$ and $v = 9$, which is suggested by Martins and Stedinger (2000); Martins and
Stedinger (2001), as the prior distribution on the shape parameter $\xi_0$. For the other parameters
$\mu_0, \mu_1, \sigma_0, \sigma_1$, the prior distributions are set to non-informative (flat) priors. There are two advantage of
the Bayesian method. First, as noted by Adlouni et al. (2007), this method allows the addition of the
other information, e.g., historical and regional information, through defining the prior distribution.
Second, the Bayesian method can provide an explicit way to account for the uncertainty of flood
estimation. In nonstationary case, in the $t$-year, the 95% credible interval for the estimation of the flood
quantile corresponding to a given probability $p$ can be obtained from a set of stable parameters
estimations $\hat{\theta}^i_{\text{NGEV}} (i = 1, 2, ..., M_c)$ in which $M_c$ is the length of the Markov chain.

The Akaike's information criterion (AIC) (Akaike, 1974) is used to rank the performance of the

NGEV models shown in Table 1, which is given by



$$AIC = -2\log\text{L}(M) + 2df \tag{13}$$
where $\log\text{L}(M)$ is the maximized log-likelihood of the NGEV model ($M$) and $df$ is the freedom degree.
In this study, to test the significance of the more complex model structure for nested models, we
refer to the chi-square test (Coles, 2001). Given the models $M_0 \subseteq M_1$, the deviance statistic for the test
is defined as
$$D = 2\left(\log\text{L}(M_1) - \log\text{L}(M_0)\right) \tag{14}$$
When the p-value of the chi-square distribution $\chi_k$ ($k$ is the difference of the number of the parameters
between the two models $M_0$ and $M_1$) is less than 0.05, the more complex model ($M_1$) is considered
significant. We use the quantile plot based on the diagnosis method suggested by Coles (2001) to check
whether the final model can well represent the data. In the nonstationary case, the diagnosis can be
performed by testing the standardized series (denoted as $\overline{Y_t}$) of the nonstationary flood series ($Y_t$),
conditional on the fitted parameter values. The variable $\overline{Y_t}$ is defined by
$$\overline{Y_t} = \frac{1}{\xi_0}\ln\left(1 + \xi_0\frac{Y_t - \mu_t}{\sigma_t}\right) \tag{15}$$
If the model is correct (i.e., $Y_t \sim GEV(\mu_t, \sigma_t, \xi_0)$), $\overline{Y_t}$ have a standard Gumbel distribution.
**3 Study area and data**
**3.1 Study area**
The Hanjiang River (Figure 4), with the coordinates of 30°30′-34°30′ N, 106°00′-114°00′ E and
a catchment area of 159000 km$^2$, is the largest tributary of the Yangtze River, China. Since 1960, many



reservoirs have been completed in the Hanjiang basin. The information of the middle-sized and large
reservoirs has been shown in Table 2, including the longitude, latitude, control area, time for completion
and capability. The Danjiangkou Reservoir in central China's Hubei province is the largest one in this
basin, which is completed by 1967. After the Danjiangkou Dam Extension Project in 2010, the
Danjiangkou Reservoir gained an additional capacity of 13.0 billion m$^3$ and an extra flood control
storage capacity of 3.3 billion m$^3$. Previous studies demonstrated that the effect of reservoirs on the flow
regime of the Hanjiang River is significant (Cong et al., 2013; GUO et al., 2008; Jiang et al., 2014; Lu
et al., 2009).
<Figure 4>
<Table 2>
**3.2 Data**
The assessment analysis of reservoir effects on flood frequency utilizes the streamflow data, the
reservoir data, and the rainfall data. The annual maximum daily flood series (AMDF) is extracted from
the daily streamflow records of the three gauges in the Hanjiang River basin, namely Ankang (AK)
station (with a drainage area of 38600 km$^2$), Huangjiagang (HJG) station (with a drainage area of
90491 km$^2$) and Huangzhuang (HZ) station (with a drainage area of 142056 km$^2$). The streamflow and
reservoir data are provided by the Hydrology Bureau of the Changjiang Water Resources Commission,
China (http://www.cjh.com.cn/en/index.html). Annual series of the maximum ($M$), mean intensity ($I$),

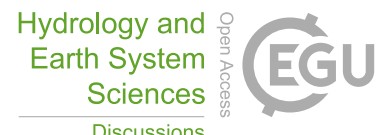

volume ($V$) and timing ($T$) in the annual critical MRI (defined as continuous daily rain records
matching the timing of AMDF, and in which any two consecutive days of rainfall values require more
than 0.2 mm) are obtained from the areal average daily rainfall series which are calculated using the
inverse distance weighting (IDW) method, based on the rainfall records of 16 sites (shown in Figure 4)
which are downloaded from the National Climate Center of the China Meteorological Administration
(source: http://www.cma.gov.cn/). For the Ankang and Huangzhuang stations, all records are available
from 1956 to 2015, while the records of the Huangjiagang station are available from 1956 to 2013.
**4 Results**
**4.1 The identification of the reservoir effects**
In this section, to confirm the impact of reservoirs on annual maximum daily flow (AMDF) in the study
area, the two statistical tests were performed, including the Mann–Kendall test (Kendall, 1975; Mann,
1945) for trend component and the Pettitt test (Pettitt, 1979) for change point, and then, the statistical
characteristics of AMDF before and after reservoir construction were analyzed. According to the Table
3, the mean and standard deviation of flood series in AK, HJG and HZ stations were significantly
reduced after reservoir construction. Taking the HJG station as an example, after the completion of two
large reservoirs (in 1966 and 1992, respectively), the mean of AMDF (1992-2013) is 4139 $m^3/s$, which
is only 0.28 time of 14951 $m^3/s$ (1956-1966) and the standard deviation is 4074 $m^3/s$, about 0.52 time of
7896 $m^3/s$ (1956-1966). Thus, the results of Table 3 indicate that the impact of reservoirs is significant
for floods in the Hanjiang River basin.





<Table 3>

Actually, although the Danjiangkou Reservoir (built in 1967) is a very large reservoir, its

construction time and change point of floods series are inconsistent. This could be caused by the effects
of other medium-large reservoirs (listed in Table 2) and rainfall factors (e.g., special extreme MRI may
limit or reduce the effects of the reservoir). Figure 5 presents the linear correlation between the four
variables of the MRI and AMDF. It is found that for the three stations (AK, HJG and HZ), except for
the timing ($T$) of the MRI in the AK station, Pearson correlation coefficients between each rainfall
variable and AMDF range from 0.27 to 0.71 (p-value>0.05), indicating that multivariate MRI
significantly affects AMDF. The further analysis for more precise effects of reservoirs is performed in
the 4.2 and 4.3 sections.
<Figure 5>
**4.2 Results for rainfall-reservoir composite index (RRCI)**

The C-vine copula model was applied to model the joint probability of the rainfall variables. To

identify the scheduling-related rainfall variables (i.e., the best subset from four rainfall variables), the
RRCI for all subsets were calculated and compared. The Pearson, Kendall, and Spearman correlation
coefficients between RRCI and AMDF are listed in Table 4. In Table 4, the ordering of the root nodes
(T1 of C-vine decomposition in Figure 3) determining the whole decomposition structure matches the
ordering of variables in the cell of the first column. As shown in the first row of Table 4, there is a
negative correlation between AMDF and RI for the AK, HJG and HZ stations. The values of the





Pearson correlation coefficient between AMDF and RI for the AK, HJG and HZ stations are -0.36, -
0.56 and -0.53, respectively, demonstrating that there is a significant relation between reservoirs
capacity and reduction of AMDF. And after introducing the effect of multivariate MRI (measured by the
OR-joint exceedance probability $P_{MRI}^{\vee}$ of the variables in MRI) into RRCI, the negative correlation
becomes stronger, indicating that RRCI can more accurately represent the impacts of the reservoirs. To
derive the RRCI, the rainfall variables are identified as the scheduling-related variables through the
highest Kendall correlation. According to the highest Kendall correlation, the scheduling-related
variables for the AK station are the maximum, intensity, volume and timing; those for the HJG station
are the intensity and timing; and those for the HZ station are the intensity, volume and timing.
<Table 4>
Table 5 is the results of the final C-vine copula model for modeling the joint distribution of the
scheduling-related variables, by aid of the R package "VineCopula" (https://CRAN.R-
project.org/package=VineCopula). For each bivariate pair in the third column of Table 5, three one-
parameter bivariate Archimedean copula families (i.e., the Gumbel, Frank, and Clayton copulas)
(Nelsen, 2006), are used to select from. As shown in Table 5, the results of the Cramer-von Mises test
(Genest et al., 2009) show that the C-vine copula model passed the test at the significant level of 0.05,
indicating the model can be effective for simulating the joint distribution of the scheduling-related
variables. Finally, the change of RI and RRCI over time is displayed in Figure 6. It is found that after
reservoir construction, for most years, the values of RRCI are larger (close to 1) than those of RI, which





implied that the effects of reservoirs on AMDF in these years may be underestimated by RI. On the
other hand, for few special years, because of special rainfall events, the effect of reservoirs on AMDF
may be overestimated by RI.

<Figure 6>

<Table 5>

**4.3 Flood frequency analysis**
In this section, nonstationary flood frequency analysis using RRCI or RI as covariate is performed to
investigate how reservoirs affect the frequency of AMDF. The results of AIC in Table 6 show that for
the models (M11, M12 and M13) using RI as covariate, the performance of model (M13) in which both
location and scale parameters of GEV distribution are time-varying is better than the models (M11 and
M12) in which only location or scale parameter are time-varying. And this situation is the same for the
models (M21, M22 and M23) using RRCI as covariate. Thus, to compare RRCI with RI, we only focus
on the model M13 and the model M23. Take the HZ station as an example. The time-varying GEV
distribution parameters for the HZ station are given as follows
(1) Model M13

$$\mu_t = 9333 - 12257RI$$
$$\sigma_t = \exp\left(9.048 - 2.531RI\right) \qquad (16)$$
$$\xi_0 = 0.099$$

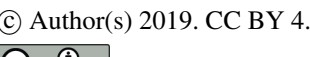


(2) Model M23

$$\mu_t = 11850 - 8937\,RRCI$$
$$\sigma_t = \exp\left(9.007 - 1.418\,RRCI\right) \tag{17}$$
$$\xi_0 = -0.065$$

For the model M13, the estimate values of $\mu_1$ and $\sigma_1$ are -12257 and -2.531, respectively, while for the

model M23, the estimate values of $\mu_1$ and $\sigma_1$ are -8937 and -1.418. The negative estimate values of $\mu_1$

and $\sigma_1$ in the Eq. (16) and Eq. (17) reveal reduction of both magnitude and scale of AMDF due to the

reservoir construction. As shown in Table 6, the best model for each gauging station is M23 with the

smallest AIC value, which means that RRCI as covariate is better than RI for explaining the alteration

of both location and scale parameters of GEV distribution. The results of the chi-square test in Table 6

indicate that the more complex structures of the best models (M23), compared to other models (except

for M21 in the AK station) are significant at the significant level of 0.05. The results of parameter

estimation of M23 are displayed in Table 7.

<Table 6>

<Table 7>

Figure 7 presents the performance of the best model (M23) for AK, HJG and HZ stations. The points in

the quantile-quantile plots of Figure 7 are close to 1:1 lines, indicating that the NGEV model (M23)

with RRCI as covariate is a reasonable model. And according to the centile curves plots of Figure 7, the

flood series is well fitted by M23 using the covariate RRCI. Take the case of the HZ station. After the

construction of Danjiangkou Reservoir (1967), due to flood operation, the magnitude and scale of





floods are significantly reduced. Moreover, considering the limits of the multivariate MRI to flood
operation, RRCI well explain the variation characteristics of flood after 1967.

\<Figure 7\>

The 100-year return levels with the 95% credible interval from the NGEV models (M13 and M23) for
the three stations are presented in Figure 8. For each station, compared to M13, M23 provides a lower
100-year return level and a smaller uncertainty range, which means that M13 with RI as covariate may
underestimate the effect of reservoirs on the floods. An overestimated return level of M13 is possibly
because in most years, the impact of the reservoir indicated by RI is less than that indicated by RRCI
considering the multivariate MRI. Further explanation from the perspective of reservoir operation is that
weak dependence relationships between the scheduling-related variables of MRI indicated by the
Kendall's tau of Table 5 is likely to reduce the flood magnitude during the periods of flood control.
Besides, the other advantage for RRCI is that the consideration of the multivariate MRI reduced the
uncertainty range of flood estimates.

\<Figure 8\>

Take the HZ gauging station as an example to illustrate the reservoir effects on the flood risk.

We investigate the risk rates (the exceedance probability) of four levels (8000, 12000, 16000 and 20000
$m^3/s$) corresponding to different flood losses according to the study of DUAN Weixin (2018). Figure 9
presents the risk rates of AMDF from M23 in the 1956-1966 (RI=0), 1967-1992 (RI=0.30), 1992-2012
(RI=0.32), and 2013-future (RI=0.50) periods. First, for every certain multivariate MRI, the risk rate on





each level of flood decreases with the increasing RI. Second, given a non-zero RI, the risk rate on each
level of flood increases with the increasing joint cumulative probability of the scheduling-related
rainfall variables (i.e., intensity, volume and timing) and gradually approaches the risk of no reservoir
effects (RI =0).
<Figure 9>

## 5 Discussions

Table 8 shows the top 5 floods and related variables after the construction (1966) of
Danjiangkou reservoir in the HZ station. It is found that the largest floods of 1967-2015 in the HZ
station occurred in 1983. For this flood event, the multiday rainfall input (MRI) can be considered rare
( $P_{MRI}^{\vee} = 0.474$ ranking the 3rd) due to the largest mean intensity (20.2 mm) and second late occurrence
(the 281th day). It is interest that the timings (ranking 2-6) of the extreme MRI for all of top 5
downstream floods seem to be overall later than those for the ordinary floods. It may be because near
the end of the major flood control period (July-October), the remained capacity of reservoir is not
sufficient to regulate the inflow floods caused by the late extreme MRI. Therefore, the timing of the
extreme MRI may be an important factor for producing the exceptional downstream flood events for the
HZ station.
In this study, the multivariate rainfall samples are obtained though corresponding the annual
maximum streamflow. This means that some extreme MRI samples due to correspond to non-maximum



flow are not included, resulting in the estimation error for $P_{MRI}^{\vee}$ . Nonetheless, the good performance of
frequency modeling demonstrates these multivariate rainfall samples may still have representativeness.
The peaks-over-threshold sampling method would be considered to obtain enough samples in the future
study.

## 6 Conclusions

We have shown that the regime of downstream floods in the study area was affected by the large
reservoirs. The real effect of reservoirs on floods is possibly related to both the reservoir capability and
the joint variation of scheduling-related rainfall variables (i.e., the maximum, intensity, volume and
timing of the MRI). It is found that the NGEV model using RRCI as covariate can lead to more accurate
flood estimations than either that using RI as covariate or the stationary GEV model. The result
demonstrates that the consideration of scheduling-related rainfall variables of the MRI is necessary for
assessing the impact of reservoirs on flood frequency.
Results in the identification of the reservoir effects show that the nonstationarity of AMDF is
significant and is possibly related to construction of the two large reservoirs (i.e., Danjiangkou and
Ankang reservoirs completed in 1967 and 1992, respectively). This is consistent with the results on the
effect of reservoirs on the flow regime in previous literature (Cong et al., 2013; GUO et al., 2008; Jiang
et al., 2014; Lu et al., 2009). The results of the C-vine copula show that the dependence relationships
between multiple rainfall variables are weak (Table 5). The Comparison between RRCI and RI (Figure
6) indicate that to some extent, for the given reservoir capacity, these weak dependency relationships



produced higher values of RRCI than that of RI in most case, however, some rare multivariate MRI still
would produce lower values of RRCI than that of RI. According to the interannual variation of RRCI,
for the downstream stations affected by reservoirs, it is expected to more significantly reduce the
number of large floods in most years; meanwhile, some unexpected large floods still will occur
conditional on rare high-impact multivariate MRI. Finally, the results of flood frequency analysis
demonstrated that as expected, RRCI better explains the interannual variability of AMDF than RI and
provided a lower flood estimation of 100-years return level with a smaller uncertainty range.

Accurately assessing the impact of reservoirs on downstream floods is an important issue for

flood risk management. In this study, to evaluate the more likely effects of reservoirs on downstream
flood risk of Hanjiang River, RRCI is derived from Eq. (3) which takes account of a combination of the
reservoir index and the joint frequency of scheduling-related rainfall variables. Then, the nonstationary
frequency model using RRCI as covariate is developed to obtain flood estimates and risk rates,
conditional on both multivariate rainfall frequencies and reservoir index. The flood risk corresponding
to a certain level of loss has been reduced by reservoirs; while, given the reservoir index, the flood risk
of rare multivariate MRI still is greater than that of ordinary multivariate MRI, but not higher than the
flood risk of no reservoir. Thus, during flood control periods, the prediction of multivariate MRI may
play an important role in assessing the downstream flood risk. The study provided a comprehensive
approach and knowledge for flood risk management to perform more accurate analysis of reservoir
effects.





## Acknowledgments

This research is financially supported jointly by the National Natural Science Foundation of China (NSFC Grants 41890822 and 51525902), the Research Council of Norway (FRINATEK Project 274310), and the "111 Project" Fund of China (B18037), all of which are greatly appreciated. No conflict of interest exists in the submission of the manuscript.

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




# Tables

Table 1. The NGEV models with different formula of the location and scale parameters for flood

frequency analysis.

| Model | ID | The formula of distribution parameters | |
| :---: | :---: | :---: | :---: |
| | | $\mu_t$ | $\ln(\sigma_t)$ |
| NGEV($\mu_t\sim1$, $\sigma_t\sim1$) | M0 | $\mu_0$ | $\sigma_0$ |
| NGEV($\mu_t\sim$RI, $\sigma\sim1$) | M11 | $\mu_0+\mu_1$RI | $\sigma_0$ |
| NGEV($\mu_t\sim1$, $\sigma\sim$RI) | M12 | $\mu_0$ | $\sigma_0+\sigma_1$RI |
| NGEV($\mu_t\sim$RI, $\sigma\sim$RI) | M13 | $\mu_0+\mu_1$RI | $\sigma_0+\sigma_1$RI |
| NGEV($\mu_t\sim$RRCI, $\sigma_t\sim1$) | M21 | $\mu_0+\mu_1$RRCI | $\sigma_0$ |
| NGEV($\mu_t\sim1$, $\sigma_t\sim$RRCI) | M22 | $\mu_0$ | $\sigma_0+\sigma_1$RRCI |
| NGEV($\mu_t\sim$RRCI, $\sigma_t\sim$RRCI) | M23 | $\mu_0+\mu_1$RRCI | $\sigma_0+\sigma_1$RRCI |





Table 2. The information of the reservoirs in the Hanjiang River basin.

| Reservoirs | Longitude | Latitude | Area (km$^2$) | Year | Capacity ($10^9$ m$^3$) |
|---|---|---|---|---|---|
| Shiquan | 108.05 | 33.04 | 23400 | 1974 | 0.566 |
| Ankang | 108.83 | 32.54 | 35700 | 1992 | 3.21 |
| Huanglongtan | 110.53 | 32.68 | 10688 | 1978 | 1.17 |
| Dangjiangkou | 111.51 | 32.54 | 95220 | 1967 | 21.0 |
| Dangjiangkou+ | 111.51 | 32.54 | 95220 | 2010 | 13.0 |
| Yahekou | 112.49 | 33.38 | 3030 | 1960 | 1.32 |






Table 3. The change in the mean and standard deviation of AMDF after the construction of the
two large reservoirs (i.e., the Danjiangkou reservoir built by 1966, and the Ankang reservoir built by

1992).

| Stations | Mean (m$^3$/s) | | | Standard deviation (m$^3$/s) | | |
|---|---|---|---|---|---|---|
| | 1956-1966 | 1967-1991 | 1992-2015 | 1956-1966 | 1967-1991 | 1992-2015 |
| AK | 9451 | 10468 | 6506 | 4341 | 4623 | 4454 |
| HJG | 14951 | 7524 | 4139 | 7896 | 5482 | 4074 |
| HZ | 16603 | 10120 | 5958 | 8833 | 5420 | 4721 |












Table 4. Correlation coefficients between the RRCI and the AMDF.

| Subset of rainfall variables | AK | | | HJG | | | HZ | | |
|---|---|---|---|---|---|---|---|---|---|
| | Pearson | Kendall | Spearman | Pearson | Kendall | Spearman | Pearson | Kendall | Spearman |
| -* | -0.36 | -0.22 | -0.29 | -0.56 | -0.42 | -0.55 | -0.53 | -0.40 | -0.53 |
| M | -0.29 | -0.28 | -0.38 | -0.67 | -0.53 | -0.74 | -0.46 | -0.38 | -0.51 |
| I | -0.34 | -0.28 | -0.37 | **-0.78** | -0.64 | -0.84 | -0.55 | -0.42 | -0.57 |
| V | -0.34 | -0.28 | -0.39 | -0.67 | -0.54 | -0.75 | -0.58 | -0.47 | -0.65 |
| T | -0.12 | -0.17 | -0.24 | -0.68 | -0.55 | -0.73 | -0.49 | -0.41 | -0.58 |
| M, I | -0.43 | -0.29 | -0.40 | -0.72 | -0.62 | -0.82 | -0.55 | -0.42 | -0.57 |
| M, V | -0.45 | -0.29 | -0.41 | -0.67 | -0.55 | -0.75 | -0.55 | -0.46 | -0.62 |
| M, T | -0.38 | -0.25 | -0.35 | -0.69 | -0.59 | -0.79 | -0.63 | -0.47 | -0.64 |
| I, V | -0.50 | -0.33 | -0.44 | -0.72 | -0.63 | -0.82 | -0.65 | -0.51 | -0.68 |
| I, T | -0.39 | -0.25 | -0.34 | -0.75 | **-0.65** | **-0.84** | -0.68 | -0.50 | -0.66 |
| V, T | -0.46 | -0.28 | -0.39 | -0.70 | -0.58 | -0.79 | -0.69 | -0.52 | -0.70 |
| M, I, V | -0.55 | -0.33 | -0.44 | -0.69 | -0.63 | -0.83 | -0.63 | -0.47 | -0.64 |
| M, I, T | -0.45 | -0.28 | -0.39 | -0.70 | -0.63 | -0.82 | -0.67 | -0.50 | -0.66 |
| M, V, T | -0.54 | -0.31 | -0.41 | -0.67 | -0.58 | -0.78 | -0.67 | -0.50 | -0.67 |
| I, V, T | -0.53 | -0.31 | -0.41 | -0.70 | -0.64 | -0.83 | **-0.71** | **-0.53** | **-0.70** |
| M, I, V, T | **-0.55** | **-0.33** | **-0.44** | -0.67 | -0.62 | -0.82 | -0.69 | -0.51 | -0.68 |

*The values in the first row are the correlation coefficients between RI and flood seires




Table 5. Results of copula models.

| Stations | Scheduling-related variables | Pairs | Copula type | Parameters $\theta_c$ | Kendall's tau | Goodness-of-fit test based on the empirical copula | |
|---|---|---|---|---|---|---|---|
| | | | | | | CvM* | p-value |
| AK | *M, I, V, T* | 14 | Clayton | 0.06 | 0.03 | 0.108 | 0.965 |
| | | 13 | Clayton | 0.47 | 0.19 | | |
| | | 12 | Clayton | 0.27 | 0.12 | | |
| | | 24\|1 | Frank | 1.13 | 0.12 | | |
| | | 23\|1 | Frank | -1.8 | -0.19 | | |
| | | 34\|12 | Clayton | 0.1 | 0.05 | | |
| HJG | *I, T* | 24 | Clayton | 0.95 | 0.32 | 0.624 | 0.755 |
| HZ | *I, V, T* | 24 | Gumbel | 1.07 | 0.07 | 0.107 | 0.95 |
| | | 23 | Clayton | 0.58 | 0.22 | | |
| | | 34\|2 | Clayton | 0.27 | 0.12 | | |

* CvM is the statistic of the Cramer-von Mises test; if the p-value of the C-vine copula model is less than the significance level of 0.05, the model is considered to be
not consistent with the empirical copula.






Table 6. The selection of the GEV models and the significance of the final model with the chi-
square test.

| Model ID | AK | | | HJG | | | HZ | | |
|---|---|---|---|---|---|---|---|---|---|
| | AIC | Chi-square test | | AIC | Chi-square test | | AIC | Chi-square test | |
| | | D | p-value | | D | p-value | | D | p-value |
| M0 | 1189.7 | 19.44 | 0.000 | 1158.9 | 64.17 | 0.000 | 1220.0 | 45.94 | 0.000 |
| M11 | 1182.4 | 12.12 | 0.000 | 1155.7 | 60.95 | 0.000 | 1214.5 | 40.50 | 0.000 |
| M12 | 1191.6 | 21.34 | 0.000 | 1155.8 | 61.10 | 0.000 | 1218.2 | 44.19 | 0.000 |
| M13 | 1184.5 | 14.25 | 0.000 | 1149.4 | 54.69 | 0.000 | 1207.0 | 32.98 | 0.000 |
| M21 | 1170.9 | 0.62 | 0.433 | 1146.8 | 52.10 | 0.000 | 1198.9 | 24.91 | 0.000 |
| M22 | 1191.6 | 21.39 | 0.000 | 1138.2 | 43.50 | 0.000 | 1201.7 | 27.66 | 0.000 |
| M23 | **1170.3** | 0.00 | 1.000 | **1094.7** | 0.00 | 1.000 | **1174.0** | 0.00 | 1.000 |




Table 7. The results of the parameter estimation for the best model (M23).

| Station | $\mu_t$ | $\sigma_t$ | $\xi_0$ |
|---|---|---|---|
| AK | 8323-5060RRCI | exp(8.353-0.508RRCI) | -0.065 |
| HJG | 12180-1053RRCI | exp(9.283-2.107RRCI) | 0.006 |
| HZ | 11850-8937RRCI | exp(9.007-1.418RRCI) | -0.065 |






Table 8. The top 5 floods and the corresponding RRCI, $P_{MRI}^{\vee}$ and scheduling-related rainfall
variables after the construction (1967) of Danjiangkou reservoir in the HZ station.

| Year | AMDF (m³/s) | Values (Ranking in 1967-2015) | | | | | |
|---|---|---|---|---|---|---|---|
| | | RRCI | RI | $P_{MRI}^{\vee}$ | I | V | T |
| 1983 | 25600 | 0.176 (3) | 0.301 (-) | 0.474 (3) | 20.2 (1) | 121.4 (19) | 281 (2) |
| 1975 | 19900 | 0.247 (6) | 0.299 (-) | 0.552 (6) | 9.6 (18) | 163.6 (13) | 277 (6) |
| 1974 | 18200 | 0.246 (5) | 0.299 (-) | 0.551 (5) | 12.0 (7) | 120.4 (20) | 278 (4) |
| 2005 | 16800 | 0.372 (12) | 0.318 (-) | 0.631 (11) | 8.2 (27) | 179.7 (10) | 278 (4) |
| 1984 | 16100 | 0.164 (1) | 0.301 (-) | 0.460 (2) | 9.9 (15) | 256.3 (4) | 273 (9) |







## Figures

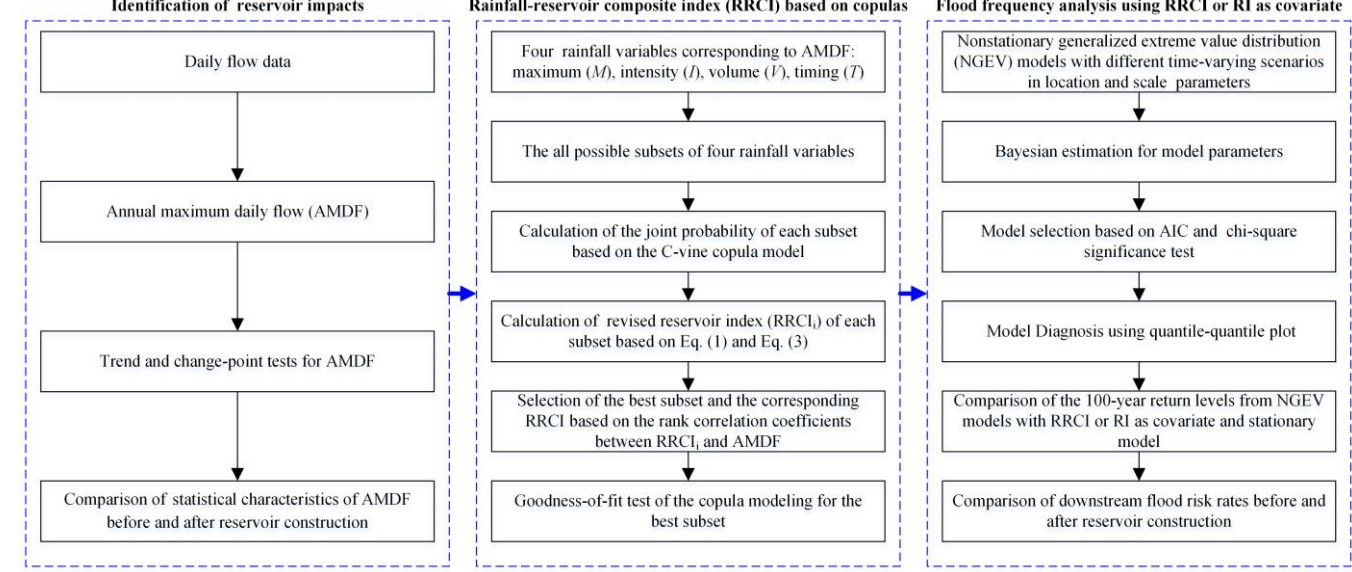

Figure 1. The flowchart of nonstationary flood frequency analysis with a rainfall-reservoir composite index (RRCI).



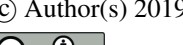



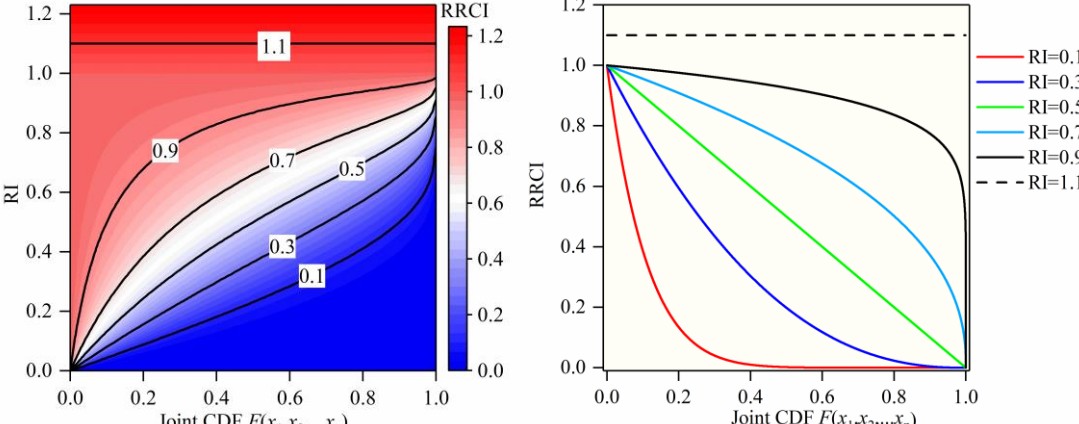

Figure 2. The relationship in the Eq. (3). The left panel is the contour of RRCI; the right panel is
the F-RRCI curves under the different values of RI.





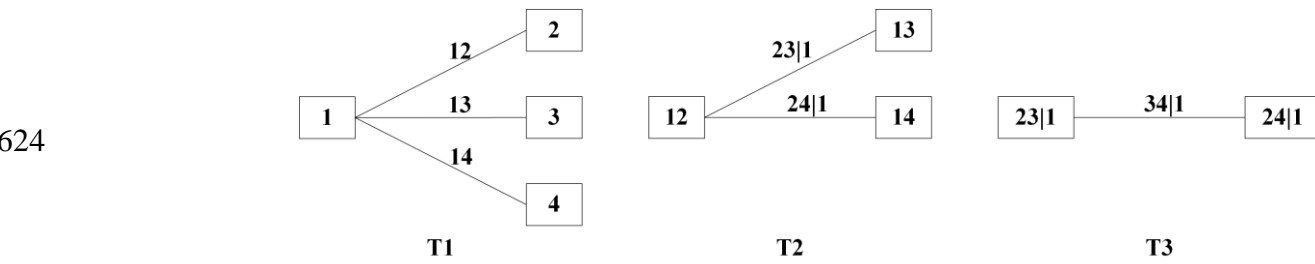

Figure 3. Decomposition of a C-vine copula with four variables and 3 trees (denoted by T1, T2 and T3).







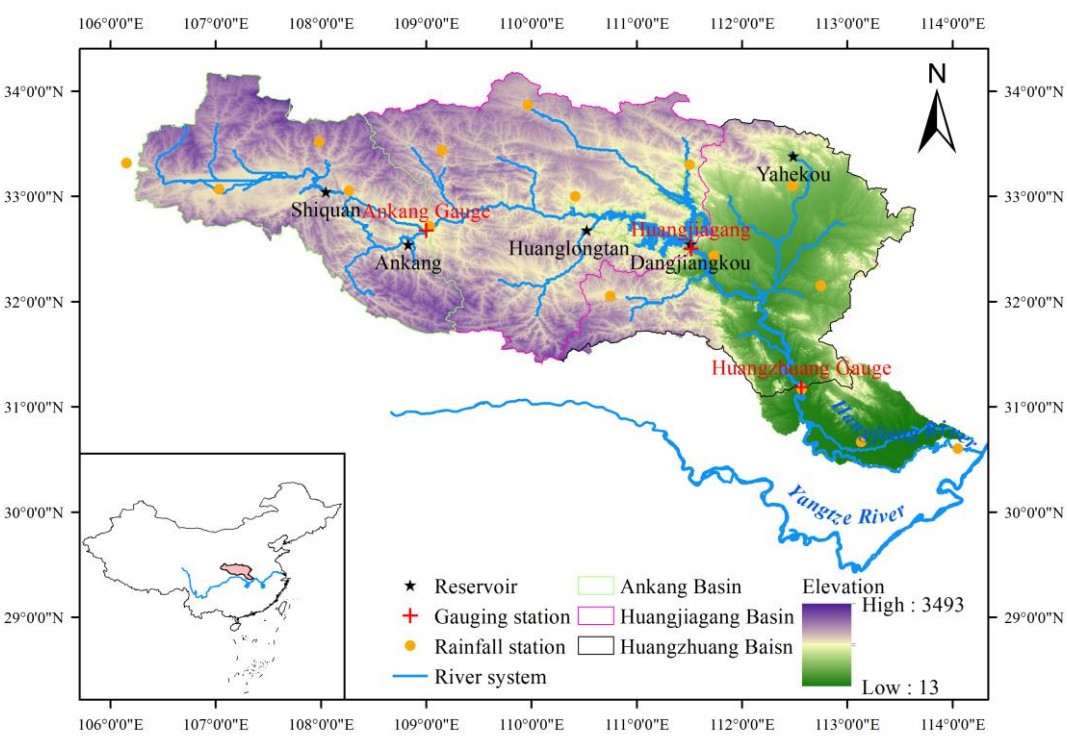


Figure 4. Geographic location of the reservoirs, gauging stations and rainfall stations in the

Hanjiang River.







Figure 5. Linear correlation between the variables of multivariate MRI and AMDF.









Figure 6. Variation of RI and RRCI.




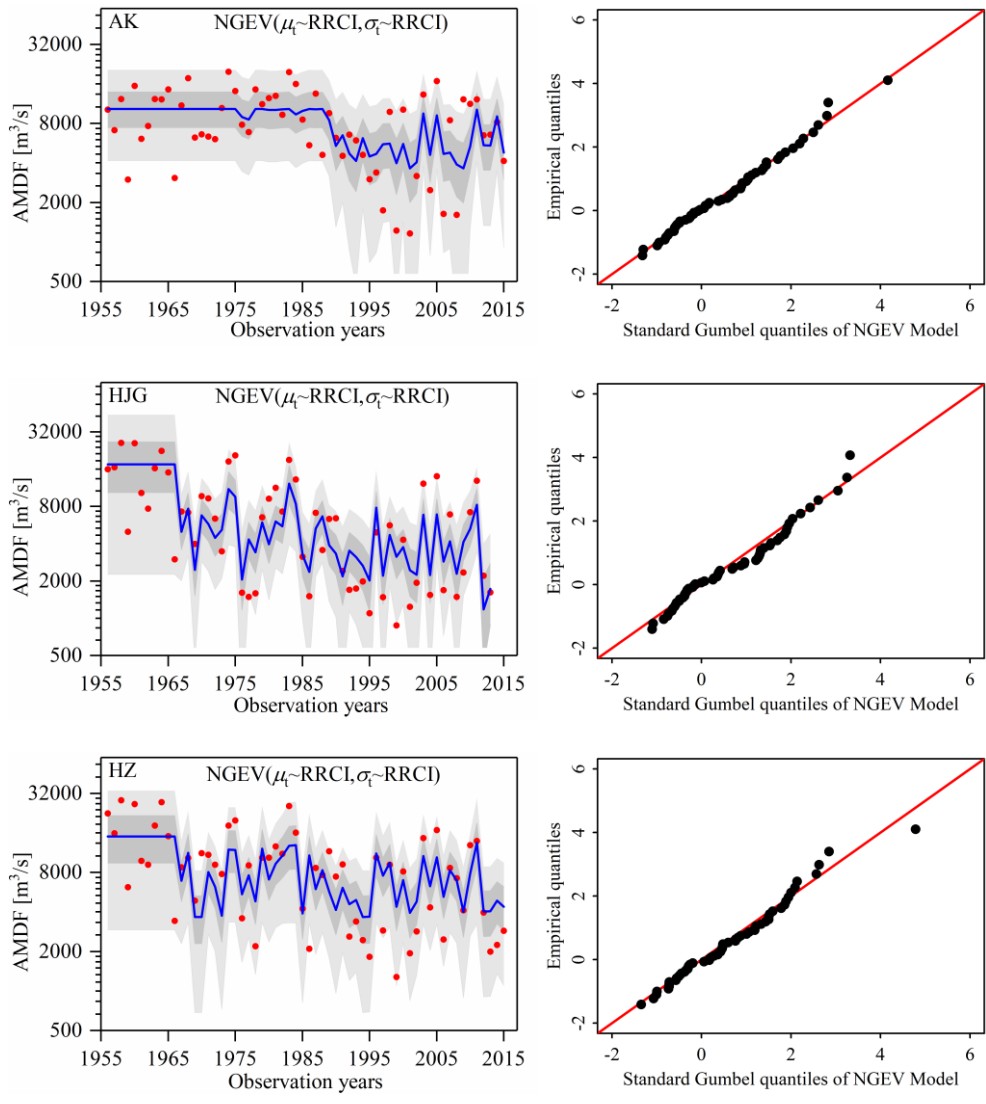




Figure 7. The performance of the best model (M23) for the Ankang (AK), Huangjiagang (HJG)

and Huangzhuang (HZ) stations. The left panel is the centile curves plots (the 50th centile curves are

indicated by thick blue; the light gray-filled areas are between the 5th and 95th centile curves; the dark

grey-filled areas are between the 25th and 75th centile curves; the filled red points indicate the observed



series). The right panel is the quantile-quantile plots based on Eq. (15); a reasonable model should have
the plotted points close to 1:1 line.






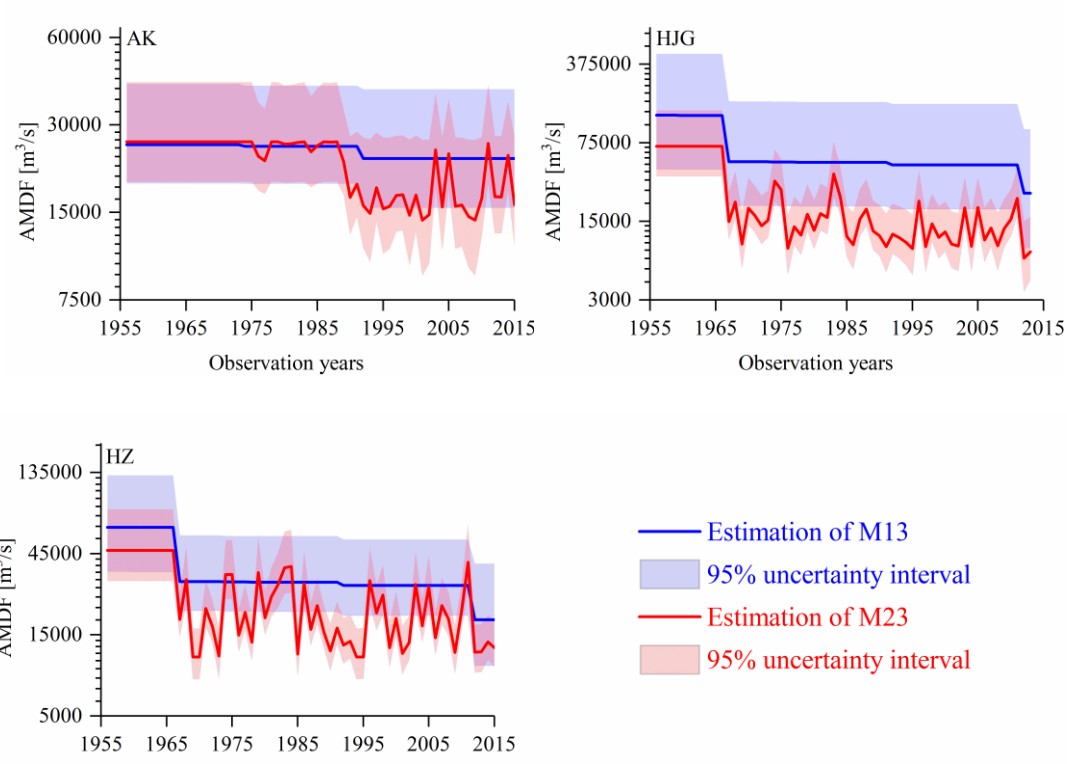

Figure 8. Statistical inference of the 100-year return levels from the models (M13 and M23) with the 95%

uncertainty interval.





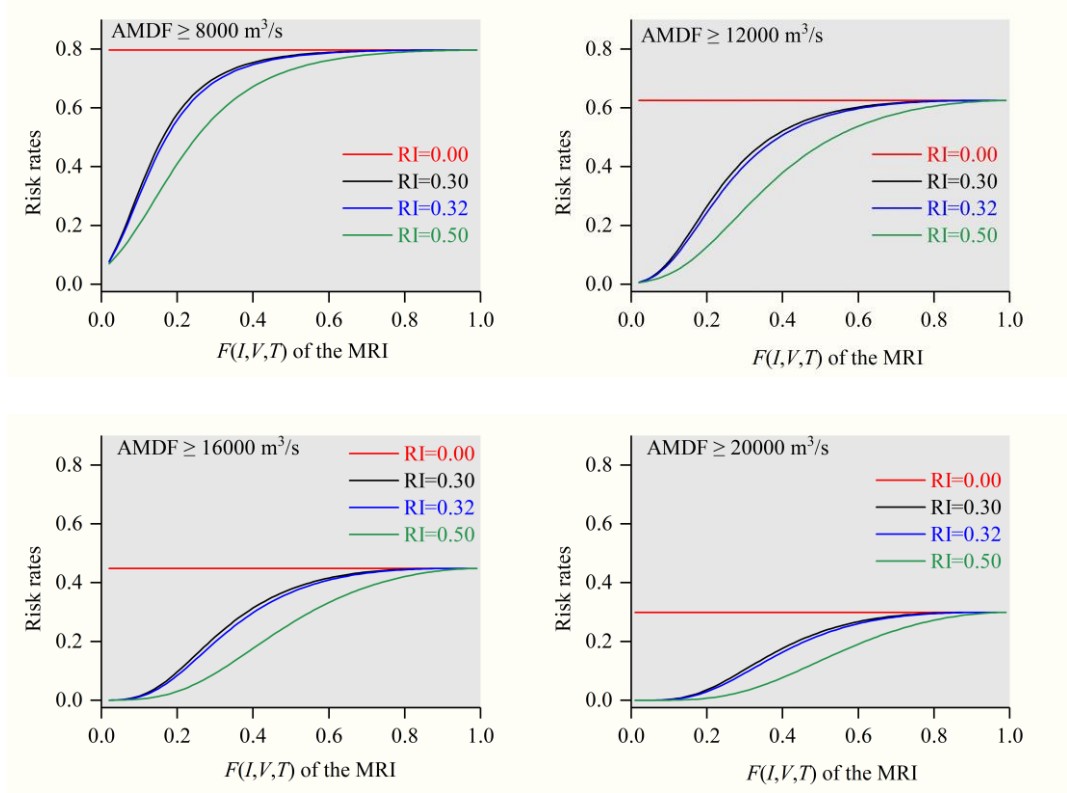

Figure 9. Comparison of risk rates of flood events corresponding to different levels of loss in the

HZ station between 1956-1966 (RI=0), 1967-1992 (RI=0.30), 1992-2012 (RI=0.32), and 2013-future

(RI=0.50) periods based on the final model (M23).