# Peer review of "Assessing the impacts of reservoirs on the downstream flood frequency by coupling the effect of the scheduling-related multivariate rainfall into an indicator of reservoir effects"

_Hydrology and Earth System Sciences, 2019_

## Referee Comment (RC1) · Anonymous Referee #1 · 16 Apr 2019

This study describes a modeling framework to account for the role of reservoirs in flood frequency analysis. While I think that the topic is generally of interest to the readership of this journal, I have a number of comments that should be addressed before considering it for publication.

- The manuscript needs to be proofread more carefully as there are several typos and unclear sentences. I will try point out some of these issues in the comments below, but this is not a complete list.

- Line 26: what "previous study"?
- Lines 46-49: which of the two references is the quote from?
- Line 49: "nature extreme flow" is unclear.
- Line 46: "this method makes it suitable"
- Line 77: "the first approach". Also, please add a reference to support the statement.
- Lines 95-96: unclear why you can't get the uncertainties in the estimates. Please clarify.
- Line 98: "all their cases"
- Line 104: "for the expression of the distribution"
- Line 106: "in the expression"
- Given that you use a GEV but leave the shape parameter constant (and this is fine), please add more 2-parameter distributions (e.g., lognormal, gamma, Weibull, Gumbel) which have only two parameters that you can make vary as a function of your covariates.
- Line 132: "To analyze"
- Line 139: "The Eq. (1)"
- If I get this right, you are assuming that the sediment trapping capability of the reservoir is negligible. However, over time the amount of storage decreases. To account for the role of sediment in reducing the reservoir capacity over time, I highly recommend the use of the Brune curve to account for it. If not Brune curve, please account for it in some fashion.
- Line 157: "the greater the MRI impact"
- Line 158: what does "inflexible" mean in this context?
- Line 161: "where"

- In terms of predictors, the spatial distribution of rainfall is not really captured. I can think of situations in which the same basin-averaged rainfall will have very different effects if most of the rainfall occurs far or close to the outlet. How is this addressed here?

- Line 185: "marginals"
- Line 204: "extensively concerned" is unclear.
- Line 208: what does "obeys nonstationary distribution" mean?

- What about model selection based on the SBC index? Would you get a more parsimonious model?

- Line 254: I don't think this statement is correct, given that you would be able to say whether a more complex model should be selected over a more complex one, not if the fit is good or bad.

- Line 266: ", and was completed"
- Line 281: what is the definition of "timing"?
- Line 303: what does "special" mean?
- Line 314: "was calculated"

- In fitting the copulas, the marginals were treating as stationary. Is this really the case? Please test for the presence of nonstationarities in the marginals of the predictors. If nonstationary, please account for it.

- The role of the Mann-Kendall and Pettitt tests is unclear to me. First of all, the results are discussed at a very basic and superficial level. Also, if the response variable tends to change with time but because the predictors you have selected change over time as well, then whether Y is stationary or not is not very important; however, whether
the relationship between predictors and predictand doesn't change over time becomes more relevant. Please fix this part.

- Lines 362-364: Please apply a correction to account for the fact you are performing multiple hypothesis testing

- Line 374: "explains"
- Line 391: "for every certain multivariate MRI" is unclear.
- Line 402: "It is of interest"
- Line 404: "the remaining capacity of the reservoir"
- Line 409: "due to correspond to" is unclear
- Line 423: "related to the construction"
- Line 427: "is weak"; "The comparison"
- Line 428: "indicates"
- Line 429: "in most cases"
- Line 435: "100-year"
- Line 649: "thick blue" what?
- Line 651: "The right panels are"

HESSD

---

## Referee Comment (RC2) · Anonymous Referee #2 · 24 Apr 2019

General Comments: The manuscript presents downstream flood frequency analysis framework using the annual maximum daily flows (AMDF). Joint cumulative probability of multiple rainfall variables (maximum, intensity, volume and timing) are considered as multiday rainfall input (MRI) and employed in C-vine copula model. Flood frequency model is defined by nonstationary generalized extreme value (NGEV) distribution model including uncertainty deliberation with Bayesian approach. Rainfall-reservoir composite index (RRCI) is proposed and used to quantify the reservoir effects as covariate for expression of distribution parameters. According to the different

metrics, the results of the proposed method outperforms typical reservoir index (RI) based flood frequency model which only accounts reservoir capacity and mean annual runoff. I believe the study is quite interesting for the readership of the journal and contributing to better modeling of downstream flood peak mechanism. The model results give reasonable outcomes and can be useful for regions where large reservoirs are located. The manuscript deserves publication after a major revision considering my below comments: - Language needs some refinements before publication. Also, there are some typos and repeated sentences, which make hard to follow and disturb the readability. It would be nice to revise the manuscript totally by dividing long sentences and eliminating the repeated ones. Same tense should be used (is or was) thought the text. - Studies dealing with downstream hydrograph alterations caused by dams are not discussed enough in the literature. - As stated in Lines 45-49, there are several factors for the generation of the floods. Authors focused on meteorological conditions, but also indicating the importance of hydrological conditions such as snow cover. The elevation range of the study area is guite wide (13 - 3493 m) and most upstream reservoirs (especially Ankang gauge) should be dominated by snowmelt. The response of the basin will be complex compared to lower altitude basins. There is not much information about the assessment of the snowmelt contribution of the catchments and their effects on operational decisions. It is also interesting to see that linear correlations between the timing variable of multivariate MRI and AMDF give lowest (almost zero) Pearson r for AK gauge in Figure 5. Would snowmelt be a reason for this? If this is the case, maybe RRCI is not enough to explain downstream peak floods for the regions where reservoirs fed by snowmelt? Temperature data can also be effective to estimate flood peaks in such cases. I believe this situation should be clarified. -In Data Section, the explanation of reservoir data is based on only their capacities. There is not much information how they are operated. For example, for what purposes they are operated, or how their reservoir pools are divided (flood control, conservation, dead storage etc.)? - It is not clear why inverse distance weighting (IDW) is selected for areal distribution of the rainfall records. The catchments are large and elevation
ranges in between 13-3493 m, so that this method may not be representative especially for mountainous regions. - Maybe it would be better to call "downstream flood frequency analysis" rather than "flood frequency analysis" throughout the manuscript? - Variation of RI and RRCI are quite different for AK gauge station in Figure 6. Please state the reason - Uncertainty of flood estimates are greater in AK stations (Figure 8) compared to the others. The reason should be explained. - Discussion section is comparatively short to conclusion part. In general the paper describes a usable approach but the main weakness is insufficient discussion of the available results. I mean, it is stated that the downstream flood regime should be altered by upstream reservoirs and the magnitude of flood peaks are reduced due to the storage capacity of them. This is expected in such a reservoir system by analyzing long period AMDF values (see Figure 7, observed AMDF). Rather, the author should elaborately clarify GEV model results in Discussion part. Main results should be given under discussion, and conclusion should be briefly summarize them. Considering these, I guess these two sections should be totally revised. - Figure and tables are appropriate. However, I have some doubts about the usefulness of Figure 9 to illustrate the reservoir effects on flood risk. It is not combining the results of the frequency model. It is not clear for what reason this figure stands for especially at the end of the result section. (I suggest removing this figure, as it is a bit confusing in terms of central theme of the paper). If authors would like to include it, I suggest them to re-organize its location through the manuscript and revise the descriptions to make it more clear (in Lines 387-395).

Specific comments: - There are too much abbreviation in the manuscript. Maybe a glossary would be useful for the readers. - Line 49, what is "nature extreme flow"? - Lines 50-52, what about the operational targets and other constraints? - Lines 52-54, requires more up-to-date references. - Lines 76-78, even a small reservoir could be very complex to derive operational strategies and a lot of detailed information might be required. I am not sure about this classification. Please consider revising this part. - Line 96, what type of uncertainty? - Line 84, which "previous studies"? - Line 108, it is a bit vague what do you mean by "more accurate effects on reservoirs?" - Lines 115-117,
please refer to Bayesian method in the objectives. - Line 143, what do you mean by "more precise effects of reservoirs"? - Line 146, please briefly explain "multiday rainfall input". - Lines 147-150, It is a bit confusing whether scheduling related multivariate (SRMR) and MRI are same or not? Could you give more detail for their explanations. - Line 155, why OR-joint exceedance probability is selected as measure function? -Line 158, what do you mean by "reservoir scheduling is more inflexible"? - Lines 170-172, selected four variable require more explanation. - Line 208, it is not clear "obeys nonstationary distribution". Please revise. - Line 280-286. The sentence is too long and difficult to understand. Please separate and revise. - Line 301, please revise "Actually, although..." - Line 303-304, it is not clear what do you mean by "(e.g., special extreme MRI may limit or reduce the effects of the reservoir)." - Line 314-315, please describe and relate calculated Spearman correlations in the text, otherwise remove them. -Lines 338-339, please clarify "special rainfall events" - Lines 412-413, please mention future studies in Conclusion part, not under Discussion. - Line 429, it is not clear what do you mean by "some rare multivariate MRI still would produce lower values of RRCI than that of RI". Please revise it.

Technical corrections: - Figure 1. The caption should be "The flowchart of nonstationary covariate-based flood frequency analysis with a rainfall-reservoir composite index (RRCI) - Figure 7. In the caption, "thick blue" should be "thick blue line". - Table 2. It would be better to not to duplicate "Dangjiangkou reservoir" and remove first row. The details should be given in the text only. - Line 26, please revise "of the previous study" - Line 35, please revise "What's more" - Lines 62-63, please revise the sentence. - In Line 73, it is stated three model components but not clear which of them are ordered since only two are given? - Line 76-78, too long sentence and hard to follow. Please revise it. - Line 119, please explain AMDF. - Line 114 and Line 120, "RRIC" should be "RRCI" - Line 115, "to calculate" should be replaced with "to develop" - Line 139, "the Eq. (1)" should be replaced with "Eq. (1)" Interactive comment

42, 2019.

---

## Author Comment (AC1) · 11 Jun 2019

**Reply to Referee #1**

This study describes a modeling framework to account for the role of reservoirs in flood frequency analysis. While I think that the topic is generally of interest to the readership of this journal, I have a number of comments that should be addressed before considering it for publication.

**Response:**

We are truly grateful for your positive comments and helpful suggestions. All your comments have been carefully addressed in the revised manuscript. Please see our point-by-point responses to your comments below.

-The manuscript needs to be proofread more carefully as there are several typos and unclear sentences. I will try point out some of these issues in the comments below, but this is not a complete list.

**Response:**

Thanks for your advice. We have carefully proofread the manuscript to correct all issues about typos and unclear expressions.

- Line 26: what "previous study"?

**Response:**

This is corrected as "López and Francés (2013)" in the revised manuscript.

- Lines 46-49: which of the two references is the quote from?

**Response:**

This quote is summarized by Wyżga et al. (2016). In the revision, this sentence has been changed as follows:

River floods are generated by various complex nonlinear processes involving physical factors including "hydrological pre-conditions (e.g. soil saturation, snow cover), meteorological conditions (e.g. amount, intensity, and spatial and temporal distribution of rainfall), runoff generation processes as well as river routing (e.g. superposition of flood waves in the main river and its tributaries)" (Wyżga et al., 2016).

- Line 49: "nature extreme flow" is unclear.

**Response:**

We have changed this sentence in the revised manuscript as follows:

In the absence of reservoirs, downstream flood extremes in most rain-dominated basins are mainly related to the corresponding extreme rainfall over the drainage area.…

- Line 46: "this method makes it suitable"

**Response:**

We can't find this sentence on Line 46. It may be on Line 75. In the revision, this sentence has been rephrased as follows:

The continuous simulation method can explicitly account for the reservoir effects on flood in a hypothetical basin. However, it is difficult to apply this approach to the most real cases (Volpi et al., 2018). The simplifying assumptions are just satisfied in a few of basins with single small reservoir. Even if some basins satisfy the simplifying assumptions, the detailed data and information required in this approach are probably unavailable.

- Line 77: "the first approach". Also, please add a reference to support the statement.

**Response:**

Corrected. In the revision, we have changed the statement for clarity as follows:

The continuous simulation method can explicitly account for the reservoir effects on flood in a hypothetical basin. However, it is difficult to apply this approach to the most real cases (Volpi et al., 2018). The simplifying assumptions are just satisfied in a few of basins with single small reservoir. Even if some basins satisfy the simplifying assumptions, the detailed data and information required in this approach are probably unavailable.

- Lines 95-96: unclear why you can't get the uncertainties in the estimates. Please clarify.

**Response:**

Thank you for pointing this out. We realize our statement is imprecise. This statement has been rephrased in the revised manuscript.

For model parameters, the ML can only get one estimate through maximization of the likelihood function, while the Bayesian inference can get multiple estimates, forming a posterior distribution of model parameters. Thus, the ML is inconvenient to describe the uncertainty of flood estimates associated with the model parameter uncertainty.

- Line 98: "all their cases"

Corrected.

- Line 104: "for the expression of the distribution"

**Response:**

Corrected.

- Line 106: "in the expression"

**Response:**

Corrected.

- Given that you use a GEV but leave the shape parameter constant (and this is fine), please add more 2-parameter distributions (e.g., lognormal, gamma, Weibull, Gumbel) which have only two parameters that you can make vary as a function of your covariates.

**Response:**

Thank you for this suggestion. In the revision, we have added the four 2-parameter distributions (i.e., lognormal, gamma, Weibull, Gumbel). The results are summarized in Table 7 (newly-added). The results indicate that for the AK and HZ station, the nonstationary WEI model with RRCI has a best performance, while for the HJG station, the nonstationary GA model with RRCI is the best model. In the revision, we have added Table 2 (newly-added) to summarize the used distributions. And the Table 6 and Table 7 are deleted. Detailed analyses of all new results will be included in the revised text. In the revised manuscript, all changes to Tables and Figures are listed as follows:

< Table 1> (revised)

<Table 2> (newly-added)

<Table 3> (Table 2 in the original manuscript; revised)

<Table 5> (Table 4 in the original manuscript; revised)

<Table 6> (Table 5 in the original manuscript; revised)

< Table 7> (newly-added)

< Table 8> (revised)

<Table 5 in the original manuscript> (deleted)

<Table 6 in the original manuscript> (deleted)

<Figure 1> (revised)

<Figure 5> (revised)

<Figure 6> (revised)

<Figure 7> (revised)

<Figure 8> (revised)

<Figure 9 in the original manuscript > (deleted)

- Line 132: "To analyze"

**Response:**

Corrected.

- Line 139: "The Eq. (1)"

**Response:**

Corrected.

- If I get this right, you are assuming that the sediment trapping capability of the reservoir is negligible. However, over time the amount of storage decreases. To account for the role of sediment in reducing the reservoir capacity over time, I highly recommend the use of the Brune curve to account for it. If not Brune curve, please account for it in some fashion.

**Response:**

Thank you for this good and insightful suggestion. To address your comment, RI is redefined to incorporate the impact of sediment on reducing the reservoir capacity over time in discussions. In the revision, RI is defined as

$$RI = \sum_{i=1}^{N} \left( \frac{A_i}{A_T} \right) \cdot \left( \frac{(1 - r_i^{Acc}) \cdot C_i}{R_m} \right) \tag{1}$$

where $r_i^{Acc}$ is the loss rate (%) of reservoir capacity in the $i$-th reservoir, due to the sediment deposition. The results indicate the loss of the reservoir capacity have an effect but not too big in this study (Figure S2). This is because so far, main reservoirs (i.e., Dangjiangkou and Ankang reservoirs) have a small loss rate no more than 15% (Figure S1). The estimation of $r_i^{Acc}$ has been presented in Supplementary Information (Please see Appendix A).

<Table S1> (newly-added)

<Figure S1> (newly-added)

<Figure S2> (newly-added)

Equation 1 is revised.

Equation S1 is newly-added.

Equation S2 is newly-added.

- Line 157: "the greater the MRI impact"

**Response:**

Corrected.

- Line 158: what does "inflexible" mean in this context?

**Response:**

We realize that the word "inflexible" may be inappropriate. Here, what we want to express is that the reservoir scheduling will have more constraints from the MRI. For example, when a large volume MRI occurs and its timing is near the end of flood season, the reservoir will probably face a large peak of inflow and a insufficient residual capacity due to reservoir impounding. The above explaination will been added in the revised manuscript.

- Line 161: "where"

**Response:**

Corrected.

- In terms of predictors, the spatial distribution of rainfall is not really captured. I can think of situations in which the same basin-averaged rainfall will have very different effects if most of the rainfall occurs far or close to the outlet. How is this addressed here?

**Response:**

Thank you for your comments. To capture the spatial distribution of rainfall, the distance ($L$) between the station with the maximum rainfall and the outlet have been considered. However, the results in Figure 5 (revised) show that for HZ station with the drainage area of 142056 km$^2$, there is a weak positive linear correlation (Pearson's r=0.24) between $L$ and AMDF, while for the AK station with the drainage area of 38600 km$^2$ and the HJG station 90491 km$^2$, the linear correlation between $L$ and AMDF is not significant. In the revised manuscript, this variable is considered as candidate to capture the spatial distribution of rainfall, but this variable is not selected for the calculation of RRCI, in consideration of both the non-significance correlation with floods of the study stations and the very complex fitting of 5-dimension copula.

- Line 185: "marginals"

**Response:**

Corrected.

**Response:**

- Line 208: what does "obeys nonstationary distribution" mean?

**Response:**

We have revised this statement as follows:

Suppose that flood variable $Y_t$ obeys distribution $f_{Y_t}(y_t|\boldsymbol{\eta}_t)$ with the covariate-dependent distribution parameters $\boldsymbol{\eta}_t$.

- What about model selection based on the SBC index? Would you get a more parsimonious model?

**Response:**

Thank you for your suggestion. In the revised manuscript, we have added the SBC index. And a more parsimonious model is selected based on the SBC criterion. After adding four 2-parameter distributions (i.e., lognormal, gamma, Weibull, Gumbel), the detailed results have been summarized in Table 7 (newly-added).

- Line 254: I don't think this statement is correct, given that you would be able to say whether a more complex model should be selected over a more complex one, not if the fit is good or bad.

**Response:**

Thank you. This statement has been deleted. In the revised manuscript, the chi-square test has been replaced by the SBC criterion.

**Response:**

Corrected.

**Response:**

The timing is defined as the time on which day of the year the annual maximum daily flood occurred. In the revision, the definition of "timing" will be added.

- Line 303: what does "special" mean?

**Response:**

In the revision, this sentence has been deleted.

- Line 314: "was calculated"

**Response:**

Corrected.

- In fitting the copulas, the marginals were treating as stationary. Is this really the case? Please test for the presence of nonstationarities in the marginals of the predictors. If nonstationary, please account for it.

**Response:**

Thanks. In the revision, the change-points of the variables are tested by the Pettitt test, and then, if any, the marginal with the change-point will be addressed by the estimation method (Xiong et al., 2015). The results in Table S2 show that there are the significant change-points in the mean intensity ($I$) of the AK and HJG stations and in the volume ($V$) of the HJG station. Results in Table 5 indicate that the consideration of the nonstationarity in these marginals makes little difference.

< Table S2> (newly-added)

- The role of the Mann-Kendall and Pettitt tests is unclear to me. First of all, the results are discussed at a very basic and superficial level. Also, if the response variable tends to change with time but because the predictors you have selected change over time as well, then whether Y is stationary or not is not very important; however, whether the relationship between predictors and predictand doesn't change over time becomes more relevant. Please fix this part.

**Response:**

Thanks. Here, the Mann-Kendall and Pettitt tests are indeed non-essential. We have deleted the Mann-Kendall and Pettitt tests in the revised manuscript.

It is hard to demonstrate whether the relationship between predictors and predictand doesn't change over time in this study. But this issue can be covered, because under the Bayesian framework, the uncertainty of the change of this relationship will be reflected in the posteriori distribution of model parameters.

- Lines 362-364: Please apply a correction to account for the fact you are performing multiple hypothesis testing

**Response:**

The correction has been made.

- Line 374: "explains"

**Response:**

Corrected.

- Line 391: "for every certain multivariate MRI" is unclear.

**Response:**

Revised.

- Line 402: "It is of interest"

**Response:**

Corrected.

- Line 404: "the remaining capacity of the reservoir"

**Response:**

Corrected.

- Line 409: "due to correspond to" is unclear

**Response:**

Revised.

- Line 423: "related to the construction"

**Response:**

Corrected.

- Line 427: "is weak"; "The comparison"

**Response:**

Corrected.

- Line 428: "indicates"

**Response:**

Corrected.

**Response:**

Corrected.

**Response:**

Corrected.

**Response:**

We have changed this in the revised manuscript as follows:
* * *
…the thick blue lines…

**Response:**

Corrected.

**Tables (revised and newly-added)**

Table 1. Seven scenarios for the formulas of the two distribution parameters (i.e., $\mu_t$ $\sigma_t$).

| Scenario codes | The formula of distribution parameters | |
| :---: | :---: | :---: |
| | $g_1(\mu_t)$ | $g_2(\sigma_t)$ |
| S0 | $\mu_0$ | $\sigma_0$ |
| S11 | $\mu_0 + \mu_1 RI$ | $\sigma_0$ |
| S12 | $\mu_0$ | $\sigma_0 + \sigma_1 RI$ |
| S13 | $\mu_0 + \mu_1 RI$ | $\sigma_0 + \sigma_1 RI$ |
| S21 | $\mu_0 + \mu_1 RRCI$ | $\sigma_0$ |
| S22 | $\mu_0$ | $\sigma_0 + \sigma_1 RRCI$ |
| S23 | $\mu_0 + \mu_1 RRCI$ | $\sigma_0 + \sigma_1 RRCI$ |

Table 2. Summary of the probability density functions and the used link functions for

nonstationary frequency modeling of the flood series.

[revised manuscript text omitted]

**Appendix A: Supplementary Information**

**The estimation of the loss rate (%) of reservoir capacity**

In this study, to estimate the variation of $r_i^{Acc}$ over time, it is assumed that there is the same amount of sediment in each year. Then, $r_i^{Acc}$ is estimated by

$$r_i^{Acc} = \frac{n_i L_i^m}{C_i} = \frac{n_i \bullet w_i^s \bullet T e_i}{\rho C_i} \tag{S2}$$

where $n_i$ is the number of years the $i$-th reservoir has been used, $L_i^m$ is the mean of annual loss of reservoir capacity (m$^3$), $w_i^s$ is the mean of annual inflow sediment mass (kg), $\rho$ is the density of the deposited sediment (kg/m$^3$) and $T e_i$ is the trap efficiency (%). Based on the Brune method (Brune, 1953; Mulu and Dwarakish, 2015), the trap efficiency is estimated with reservoir capacity-inflow ratio as follows

$$T e_i = 1 - \frac{0.5}{\sqrt{C_i/I_i}} \tag{S2}$$

where $I_i$ is the mean of annual inflow volume in the $i$-th reservoir (m$^3$/day ). The data in the previous literature (Guo, 1995; Hu, 2009; Liu, 2017) are collected to control the estimation errors of $L_i^m$. Please see Table S1.

Table S1. Summary for the calculation of the mean of annual loss of reservoir capacity

| Reservoirs | $C_i$ ($10^9$ m$^3$) | $I_i$ ($10^9$ m$^3$) | $w_i^s$ ($10^9$ kg) | $Te_i$ (%) | $L_i^m$ ($10^9$ m$^3$) | |
|---|---|---|---|---|---|---|
| | | | | | From previous studies | From Eq.(S2)* |
| Shiquan | 0.566 | 11.73 | 12.6 | 88% | 0.006 | 0.008 |
| Ankang | 3.21 | 19.17 | 27.1 | 94% | - | 0.018 |
| Huanglongtan | 1.17 | 6.12 | 8.58 | 94% | 0.007 | 0.006 |
| Dangjiangkou | 34.0 | 39.48 | 59.8 | 97% | 0.044 | 0.042 |
| Yahekou | 1.32 | 1.09 | - | 98% | 0.007 | - |

* $\rho = 1400$ kg/m$^3$ .

Table S2. Results of the change-point detection for the rainfall series.

| Variables | AK | | HJG | | HZ | |
|---|---|---|---|---|---|---|
| | change-point | p-value* | change-point | p-value | change-point | p-value |
| $M$ | 1976 | 1.037 | 1989 | 0.371 | 1971 | 1.278 |
| $I$ | 1987 | **0.031** | 1985 | **0.009** | 1990 | 0.080 |
| $V$ | 2009 | 0.746 | 1984 | **0.042** | 1984 | 0.769 |
| $T$ | 1992 | 1.180 | 1984 | 0.986 | 1984 | 1.367 |

*Less than 0.05 is considered significant.

[Figure]

Figure S1. Interannual variation of loss rate of reservior capacitity for each reservoir in the study area.

[Figure]

Figure S2. The impact of reservoir capacity loss on RI for AK, HJG and HZ stations.

[Figure]

Figure S3 Preliminary analysis of the snowmelt contribution of the catchment upstream the AK station. (a) is the timing of flood; (b) is the monthly average temperature; (c) is the monthly average streamflow; and (d) is the monthly average precipitation.

---

## Author Comment (AC2) · 11 Jun 2019

**Reply to Referee #2**

*General Comments:*

The manuscript presents downstream flood frequency analysis framework using the annual maximum daily flows (AMDF). Joint cumulative probability of multiple rainfall variables (maximum, intensity, volume and timing) are considered as multiday rainfall input (MRI) and employed in C-vine copula model. Flood frequency model is defined by nonstationary generalized extreme value (NGEV) distribution model including uncertainty deliberation with Bayesian approach. Rainfall reservoir composite index (RRCI) is proposed and used to quantify the reservoir effects as covariate for expression of distribution parameters. According to the different metrics, the results of the proposed method outperforms typical reservoir index (RI) based flood frequency model which only accounts reservoir capacity and mean annual runoff. I believe the study is quite interesting for the readership of the journal and contributing to better modeling of downstream flood peak mechanism. The model results give reasonable outcomes and can be useful for regions where large reservoirs are located. The manuscript deserves publication after a major revision considering my below comments.

**Response:**
Thank you very much for the good summary and the positive evaluation of the paper. All your valuable comments have been carefully addressed in the revision. Please see our point to point replay below.

- Language needs some refinements before publication. Also, there are some typos and repeated sentences, which make hard to follow and disturb the readability. It would be nice to revise the manuscript totally by dividing long sentences and eliminating the repeated ones. Same tense should be used (is or was) thought the text.

**Response:**
Thanks for your kind suggestion. We have carefully revised the text to correct all issues about typos, unclear long sentences, repeated sentences and different tenses.

- Studies dealing with downstream hydrograph alterations caused by dams are not discussed enough in the literature.

**Response:**
In the first paragraph of the modified version, we have added literature review on studies dealing with downstream hydrograph alterations caused by dams as follows:

Under intensified human activities, significant hydrological alterations caused by reservoirs are demonstrated in the many areas of the world. Graf (1999) shown that the dams probably have a greater effect on the streamflow than the global climate change in America. And the large dams have a strong downstream hydrologic effect (Graf, 2006). Batalla et al. (2004) demonstrated an evident reservoir-induced hydrologic alteration in the North-Eastern Spain. Yang et al. (2008) indicated the

spatial variability of the hydrological regimes alteration caused by the reservoirs in the middle and lower Yellow River, China. Mei et al. (2015) suggested that the Three Gorges Dam, as the largest dam in the world, has significantly changed the downstream hydrological regimes. In recent years, the cause-effect mechanisms of the downstream flood peak reduction were investigated in some literature (Ayalew et al., 2013; 2015; Volpi et al., 2018). For example, Volpi et al. (2018) demonstrated that for a single reservoir, the downstream flood peak reduction is mainly dependent on its position along the river, its spillway and its storage capacity based on a parsimonious instantaneous unit hydrograph-based model.

**Newly added literature**

Ayalew, T.B., Krajewski W.F., Mantilla R., 2015. Insights into Expected Changes in Regulated Flood Frequencies due to the Spatial Configuration of Flood Retention Ponds. Journal of Hydrologic Engineering, 20(10): 04015010.

Graf, W.L., 1999. Dam nation: A geographic census of American dams and their large‐scale hydrologic impacts. Water resources research, 35(4): 1305-1311.

Graf, W.L., 2006. Downstream hydrologic and geomorphic effects of large dams on American rivers. Geomorphology, 79(3-4): 336-360.

Mei, X., Dai, Z., Van Gelder, P.H.A.J.M., and Gao, J., 2015. Linking Three Gorges Dam and downstream hydrological regimes along the Yangtze River, China. Earth and Space Science, 2(4): 94-106.

Volpi, E., Di Lazzaro M., Bertola M., Viglione A., and Fiori A., 2018. Reservoir Effects on Flood Peak Discharge at the Catchment Scale. Water Resources Research, 54(11): 9623-9636.

Yang, T., Zhang Q., Chen Y.D., Tao X., Xu C.Y., and Chen X., 2008. A spatial assessment of hydrologic alteration caused by dam construction in the middle and lower Yellow River, China. Hydrological Processes: An International Journal, 22(18): 3829-3843.

- As stated in Lines 45-49, there are several factors for the generation of the floods. Authors focused on meteorological conditions, but also indicating the importance of hydrological conditions such as snow cover. The elevation range of the study area is quite wide (13 – 3493 m) and most upstream reservoirs (especially Ankang gauge) should be dominated by snowmelt. The response of the basin will be complex compared to lower altitude basins. There is not much information about the assessment of the snowmelt contribution of the catchments and their effects on operational decisions. It is also interesting to see that linear correlations between the timing variable of multivariate MRI and AMDF give lowest (almost zero) Pearson r for AK gauge in Figure 5. Would snowmelt be a reason for this? If this is the case, maybe RRCI is not enough to explain downstream peak floods for the regions where reservoirs fed by snowmelt? Temperature data can also be effective to estimate flood peaks in such cases. I believe this situation should be clarified.

**Response:**

Thank you for this comment. Although the elevation range of the study area is quite wide (13–3493 m), the study area is a rainfall-dominated area and the snowmelt contribution is quite limited. This area has a warm temperate semi-humid continental monsoon climate. The temperature in the basin is not much different from upstream to downstream. The timing of flood is the main rainfall period between June and September (Figure S3a, c and d). And the winter is warm as shown in Figure S3b. It is indicated that the rainfall is the main contribution for floods. The above information will be added in the revised manuscript.

<Figure S3> (newly-added)

[Figure]

Figure S3 Preliminary analysis of the snowmelt contribution of the catchment upstream the AK station. (a) is the timing of flood; (b) is the monthly average temperature; (c) is the monthly average streamflow; and (d) is the monthly average precipitation.

The reason why AK gauge has a weak linear correlation between the timing variable of multivariate MRI and the annual maximum flood in Figure 5 is probably that there is a non-significant effect of the staged operation of the reservoirs on the floods. The reservoirs upstream of AK station have a smaller capacity than HJG and HZ stations. There may be a random variation of the remaining storage capacity in each staged period of the flood season for AK station. Thus, in the long term, the reduction of the peaks of AK station tends to be not different in each staged period of the flood season.

In the revision, the above situation has been clarified. And Figure S3 has been added in Supplementary Information (Please see Appendix A).

- In Data Section, the explanation of reservoir data is based on only their capacities. There is not much information how they are operated. For example, for what purposes they are operated, or how their reservoir pools are divided (flood control, conservation, dead storage etc.)?

**Response:**

Agree. In the revision, more information on the reservoir operation has been added as follows:

…The Danjiangkou Reservoir in central China's Hubei province is the largest one in this basin, which is completed by 1967. As a multi-purpose reservoir, it mainly aims to supply water and control floods, and is also used for electricity generation and irrigation. The reservoir has the total storage capacity of 21.0 billion $m^3$, the dead storage capacity of 7.23 billion $m^3$, the effective storage capacity of 10.2 billion $m^3$, and the flood control capacity of 7.72 billion $m^3$. After the Danjiangkou Dam Extension Project in 2010, the Danjiangkou Reservoir gained an additional total storage capacity of 13.0 billion $m^3$ and an extra flood control storage capacity of 3.3 billion $m^3$…

- It is not clear why inverse distance weighting (IDW) is selected for areal distribution of the rainfall records. The catchments are large and elevation ranges in between 13-3493 m, so that this method may not be representative especially for mountainous regions.

**Response:**

The reason why IDW is selected is that IDW is a handy method. Due to both the data limitation (16 sites) and the unstable relationship between rainfall and elevation, it is hard for us to demonstrate whether the other methods (e.g., the Kriging methods) will be better. In this study, the rainfall records from all national meteorological stations in the study area are used. The precision of areal rainfall with the IDW method should be able to meet the requirement in the study. In the revision, the error of estimation of areal rainfall will be discussed to remind readers.

- Maybe it would be better to call "downstream flood frequency analysis" rather than "flood frequency analysis" throughout the manuscript?

**Response:**

Agree. We have made a revision for this throughout the manuscript.

- Variation of RI and RRCI are quite different for AK gauge station in Figure 6. Please state the reason

**Response:**

Thanks. The reason has been stated in the revised manuscript as follows:

For AK gauge, there is a difference in the variation of RI and RRCI. This is

because RRCI is dependent on both RI and the OR-joint exceedance probability (Figure 2). In spite of a low value of RI, the MRI with a high OR-joint exceedance probability will get a high RRCI. In fact, the reservoir effect on the downstream flood is great rather than small because of the fewer constraints from MRI. Thus, it is expected that RRCI can reflect a real reservoir effect more than RI.

- Uncertainty of flood estimates are greater in AK stations (Figure 8) compared to the others. The reason should be explained.

**Response:**
Thanks for this suggestion. The sample size of the regulated floods by the main reserviors upstream of AK station is smaller than HZ and HJG stations, and the dependent relationship between the floods and RRCI or RI in AK station is weaker. The greater uncertainty of the model parameters is produced. This explanation has been added in the revised manuscript.

- Discussion section is comparatively short to conclusion part. In general the paper describes a usable approach but the main weakness is insufficient discussion of the available results. I mean, it is stated that the downstream flood regime should be altered by upstream reservoirs and the magnitude of flood peaks are reduced due to the storage capacity of them. This is expected in such a reservoir system by analyzing long period AMDF values (see Figure 7, observed AMDF). Rather, the author should elaborately clarify GEV model results in Discussion part. Main results should be given under discussion, and conclusion should briefly summarize them. Considering these, I guess these two sections should be totally revised.

**Response:**
Agree. Discussion and Conclusion will be totally revised. And we will carefully modify these two sections according to your valuable suggestion.

- Figure and tables are appropriate. However, I have some doubts about the usefulness of Figure 9 to illustrate the reservoir effects on flood risk. It is not combining the results of the frequency model. It is not clear for what reason this figure stands for especially at the end of the result section. (I suggest removing this figure, as it is a bit confusing in terms of central theme of the paper). If authors would like to include it, I suggest them to re-organize its location through the manuscript and revise the descriptions to make it more clear (in Lines 387-395).

**Response:**
Agree. In order to highlight the central theme of the paper, Figure 9 has been deleted in the revised manuscript.

***Specific comments:***

-There are too much abbreviation in the manuscript. Maybe a glossary would be useful for the readers.

**Response:**

Thanks for this suggestion. We have added a glossary in Appendix B for these abbreviations.

- Line 49, what is "nature extreme flow"?

**Response:**

We have changed this sentence in the revised manuscript as follows:

In the absence of reservoirs, downstream flood extremes for most rain-dominated basins are mainly related to the corresponding extreme rainfall over the drainage area.…

- Lines 50-52, what about the operational targets and other constraints?

**Response:**

Thanks. Our statement exists imprecise. We have rephrased it as follows

However, after the construction of large or medium-sized reservoirs, the downstream extreme flow is probably dependent on the result of the reservoir scheduling which is based on a series constrainsts (e.g., the reservior capacity, the inflow or rainfall input, the operational targets and the spillways).

- Lines 52-54, requires more up-to-date references.

**Response:**

In the revision, we have added literature review on studies dealing with downstream hydrograph alterations caused by dams. Please see the response to the third comment.

- Lines 76-78, even a small reservoir could be very complex to derive operational strategies and a lot of detailed information might be required. I am not sure about this classification. Please consider revising this part.

**Response:**

Agree. A modification of this statement has been made as follows:

The continuous simulation method can explicitly account for the reservior effects on flood in the hypothetical case. However, it is difficult to apply this approach to the most real cases (Volpi et al., 2018). The simplifying assumptions are just satisfied in a few of basins with single small reservoir. Even if some basins meet the simplifying assumptions, the detailed information required in this approach are probably unavailable.

- Line 96, what type of uncertainty?

**Response:**

The uncertainty of flood estimates is associated with the uncertainty of the parameters estimates. For clarity, we have revised this sentence as follows:

For the model parameters, the ML can only get one estimate through maximization of the likelihood function, while the Bayesian inference can get multiple estimates, forming a posterior distribution of model parameters. Thus, the ML is inconvenient to describe the uncertainty of flood estimates associated with the model parameter uncertainty.

- Line 84, which "previous studies"?

**Response:**

Thanks. The correction has been made as follows:

Thus, previous studies (Adlouni et al., 2007; Caroni and Panagoulia, 2016; Ouarda and El‐Adlouni, 2011; Panagoulia et al., 2014) have used the nonstationary generalized extreme value distribution (NGEV) to describe nonstationary maxima series.

**Newly added literature**

Caroni, C., Panagoulia, D., 2016. Non-stationary modelling of extreme temperatures in a mountainous area of Greece. REVSTAT, 14: 217-228.

Panagoulia, D., Economou, P., Caroni, C., 2014. Stationary and nonstationary generalized extreme value modelling of extreme precipitation over a mountainous area under climate change. Environmetrics, 25(1): 29-43.

- Line 108, it is a bit vague what do you mean by "more accurate effects of reservoirs?"

**Response:**

Thanks. For clarity, a modification of this sentence has been made as follows:

The precision and accuracy in the quantitative analysis of the reservoir effects on the downstream floods need to be improved further.

- Lines 115-117, please refer to Bayesian method in the objectives.

**Response:**

Agree. In the revision, Bayesian method has been referred in the objectives.

- Line 143, what do you mean by "more precise effects of reservoirs"?

**Response:**

Thanks. A modification of this sentence has been made as follows:

We develop a new index to improve the precision and accuracy in the quantitative analysis of the reservoir effects on the downstream flood reduction.

- Line 146, please briefly explain "multiday rainfall input".

**Response:**

In the revision, the brief explanation has been added as follows:

In addition to reservoir capacity, multiday rainfall input (i.e., a multivariate event

with the continuous multi-day rainfall into the reservior system, MRI) is a key initial condition for the scheduling results of the reservoir system.

- Lines 147-150, It is a bit confusing whether scheduling related multivariate (SRMR) and MRI are same or not? Could you give more detail for their explanations.

**Response:**
Thanks. SRMR and MRI are different. All variables of SRMR are selected from the variables of MRI, and can be constraints for the reservoir operation. In the revised manuscript, these phrase will be accurately described and distinguished.

- Line 155, why OR-joint exceedance probability is selected as measure function?

**Response:**
We need a variable to measure the effect of a SRMR on the reservoir operation. The OR-joint exceedance probability is the likelihood that any variable in the given SRMR will be exceeded. The lower this likelihood, the greater degree of the constraint due to SRMR the reservoir operation has, so that the effect of the flood reduction is probably lower. The above explanation has been added in the revised manuscript.

- Line 158, what do you mean by "reservoir scheduling is more inflexible"?

**Response:**
We realize that the word "inflexible" may be inappropriate. Here, what we want to express is that the reservoir scheduling will have more constraints from the MRI. For example, when a large volume MRI occurs and its timing is near the end of flood season, the reservoir will probably face a large peak of inflow and a insufficient residual capacity due to reservoir impounding. The above explaination will been added in the revised manuscript.

- Lines 170-172, selected four variables require more explanation.

**Response:**
Agree. The more detailed explanation has been added.

- Line 208, it is not clear "obeys nonstationary distribution". Please revise.

**Response:**
The statement has been revised as follows:

Suppose that flood variable $Y_t$ obeys a distribution $f_{Y_t}(y_t|\boldsymbol{\eta}_t)$ with the covariate-dependent distribution parameters $\boldsymbol{\eta}_t$.

- Line 280-286. The sentence is too long and difficult to understand. Please separate and revise.

**Response:**

Thanks. The sentence has been separated and revised.

- Line 301, please revise "Actually, although: : :"

**Response:**

Thanks. In the revision, this sentence has been deleted.

- Line 303-304, it is not clear what do you mean by "(e.g., special extreme MRI may limit or reduce the effects of the reservoir)."

**Response:**

In the revision, this sentence has been deleted.

- Line 314-315, please describe and relate calculated Spearman correlations in the text, otherwise remove them.

**Response:**

The description and relation for the calculated Spearman correlations has been added in the revised text.

-Lines 338-339, please clarify "special rainfall events"

**Response:**

In the revision, we have clarified this phrase as follows:

On the other hand, for few special years, because of no considering the special rainfall events (i.e., the rainfall events with a very low OR-joint exceedance probability), RI probably overestimates the effect of reservoirs on AMDF.

- Lines 412-413, please mention future studies in Conclusion part, not under Discussion.

**Response:**

In the revision, we have followed your suggestion.

- Line 429, it is not clear what do you mean by "some rare multivariate MRI still would produce lower values of RRCI than that of RI". Please revise it.

**Response:**

Thanks. This sentence has been revised in the revised manuscript as follows:

The RRCI is lower than RI in some years, because of the rainfall events with a very low OR-joint exceedance probability in these years.

*Technical corrections:*

- Figure 1. The caption should be "The flowchart of nonstationary covariate-based flood frequency analysis with a rainfall-reservoir composite index (RRCI)

**Response:**
Thank you for this comment. We have revised the caption according to your suggestion.

- Figure 7. In the caption, "thick blue" should be "thick blue line".

**Response:**
Corrected.

- Table 2. It would be better to not to duplicate "Dangjiangkou reservoir" and remove first row. The details should be given in the text only.

**Response:**
Corrected.

- Line 26, please revise "of the previous study"

**Response:**
This is corrected as "López and Francés (2013)" in the revised manuscript.

- Line 35, please revise "What's more"

**Response:**
Revised.

- Lines 62-63, please revise the sentence.

**Response:**
In the revision, this sentence has been revised as follows:

In fact, the flood frequency downstream of dams is closely related to both the climate condition of the basin and the upstream reservoir operation.

– In Line 73, it is stated three model components but not clear which of them are ordered since only two are given?

**Response:**
Thanks. This is a mistake. We have corrected this sentence as follows:

…three model components, i.e., the stochastic rainfall generator model, the rainfall-runoff model and the reservoir flood operation module…

- Line 76-78, too long sentence and hard to follow. Please revise it.

**Response:**

Thanks. We have revised this sentence as follows:

The continuous simulation method can explicitly account for the reservoir effects on flood in a hypothetical basin. However, it is difficult to apply this approach to the most real cases (Volpi et al., 2018). The simplifying assumptions are just satisfied in a few of basins with single small reservoir. Even if some basins satisfy the simplifying assumptions, the detailed information required in this approach probably are probably unavailable.

- Line 119, please explain AMDF.

**Response:**

The explanation has been added in the revision as follows:

…the annual maximum daily flow series (AMDF) at the downstream stations

- Line 114 and Line 120, "RRIC" should be "RRCI"

**Response:**

Corrected.

- Line 115, "to calculate" should be replaced with "to develop"

**Response:**

Corrected.

- Line 139, "the Eq. (1)" should be replaced with "Eq. (1)"

**Response:**

Corrected.

**Appendix B: Glossary**

**AMDF** —Annual maximum daily flow

**CDF** —Cumulative distribution functions

**GEV** —Generalized extreme value distribution

**GML** —Generalized maximum likelihood estimation method

**IDW** —Inverse distance weighting method

**IRI** —Impounded runoff index, a ratio of reservoir capacity to mean annual flow

**JCP** —Joint cumulative probability

**ML** —Maximum likelihood method

**MRI** —Multiday rainfall input

**NGEV** —Nonstationary generalized extreme value distribution

**RRCI** —Rainfall-reservoir composite index

**RI** —Reservoir index

**SRMR** —Scheduling-related multivariate rainfall

---

## Author Response (AR1)

**Dear Editor,**

On behalf of my co-authors, I would like to express our sincere thanks to you and the anonymous reviewers for the efforts on reviewing our manuscript entitled "Assessing the impacts of reservoirs on the downstream flood frequency by coupling the effect of the scheduling-related multivariate rainfall into an indicator of reservoir effects"(ID: hess-2019-42).

According to the reviewer's comments, the manuscript has been revised. All the comments made by the reviewers are very professional. We have carefully addressed all the comments in the revision of the manuscript. A point-by-point response to the comments and the relevant changes made in the manuscript are presented as appendix to this letter. The revised manuscript with all revisions marked in red color is appended at the end of this document.

We hope the revision of the manuscript will meet with the approval of the reviewers and editors for publication in HESS.

With all best wishes.

Yours,

Lihua Xiong

On behalf of my co-authors

State Key Laboratory of Water Resources and Hydropower Engineering Science

Wuhan University, Wuhan 430072, China

E-mail: xionglh@whu.edu.cn

Telephone: +86-13871078660

Fax: +86-27-68773568

**Reply to Referee #1**

This study describes a modeling framework to account for the role of reservoirs in flood frequency analysis. While I think that the topic is generally of interest to the readership of this journal, I have a number of comments that should be addressed before considering it for publication.

**Response:**

We are truly grateful for your positive comments and helpful suggestions. All your comments have been carefully addressed in the revised manuscript. Please see our point-by-point responses to your comments below.

-The manuscript needs to be proofread more carefully as there are several typos and unclear sentences. I will try point out some of these issues in the comments below, but this is not a complete list.

**Response:**

Thanks for your advice. We have carefully proofread the manuscript to correct all issues about typos and unclear expressions.

- Line 26: what "previous study"?

**Response:**

This has been deleted in the revised manuscript.

- Lines 46-49: which of the two references is the quote from?

**Response:**

This quote is summarized by Wyżga et al. (2016). In the revision, the other reference has been deleted for clarity.

- Line 49: "nature extreme flow" is unclear.

**Response:**

For clarity, we have changed this sentence in the revised manuscript as follows:

In general, without reservoirs, the flood extremes downstream of most rain-dominated basins are mainly related to the extreme rainfall in the drainage area…

- Line 46: "this method makes it suitable"

**Response:**

We can't find this sentence on Line 46. It may be on Line 75. In the revision, this sentence has been rephrased as follows:

> The continuous simulation method can explicitly account for the reservoir effects on flood in the hypothetical case. However, it is difficult to apply this approach to the most real cases (Volpi et al., 2018), because the simplifying assumptions of this approach are just satisfied in a few of basins with single small reservoir. Furthermore, even if the basins meet the simplifying assumptions, the detailed information required in this approach are probably unavailable…

**Newly added literature:**

Volpi, E., Di Lazzaro M., Bertola M., Viglione A., and Fiori A., 2018. Reservoir Effects on Flood Peak Discharge at the Catchment Scale. Water Resources Research, 54(11): 9623-9636. https://doi.org/10.1029/2018WR023866

- Line 77: "the first approach". Also, please add a reference to support the statement.

**Response:**

Corrected. We have added the reference (please see the above response for "- Line 46, …").

- Lines 95-96: unclear why you can't get the uncertainties in the estimates. Please clarify.

**Response:**

Thank you for pointing this out. We realize our statement is imprecise. This statement has been rephrased in the revised manuscript.

> … Another drawback of the ML method is its inconvenience to describe the uncertainty of model parameters estimates, because the ML can only get one estimate of the model parameters through maximization of the likelihood function.…

- Line 98: "all their cases"

Corrected.

- Line 104: "for the expression of the distribution"

**Response:**

Corrected.

- Line 106: "in the expression"

**Response:**

Corrected.

- Given that you use a GEV but leave the shape parameter constant (and this is fine), please add more 2-parameter distributions (e.g., lognormal, gamma, Weibull, Gumbel) which have only two parameters that you can make vary as a function of your covariates.

**Response:**

Thank you for this suggestion. In the revision, we have added the four 2-parameter distributions (i.e., Lognormal, Gamma, Weibull, Gumbel). The results are summarized in Table 7 (newly-added). The results indicate that for the AK and HZ station, the nonstationary Weibull distribution with the RRCI-dependent scenario has a best performance, while for the HJG station, the nonstationary Gamma distribution with the RRCI-dependent scenario is the best model. In the revision, we have added Table 1 (newly-added) to summarize the used distributions. And the Table 6 and Table 7 are deleted. Detailed analyses of all new results have been included in the revised text. In the revised manuscript, all changes to Tables and Figures are listed as follows:

< Table 1> (newly-added)

<Table 2> (Table 1 in the original manuscript; revised)

<Table 3> (Table 2 in the original manuscript; revised)

<Table 5> (Table 4 in the original manuscript; revised)

<Table 6> (Table 5 in the original manuscript; revised)

< Table 7> (newly-added)

< Table 8> (revised)

<Table 5 in the original manuscript> (deleted)

<Table 6 in the original manuscript> (deleted)

<Figure 1> (revised)

<Figure 2> (revised)

<Figure 5> (revised)

<Figure 6> (revised)

<Figure 7> (newly-added)

<Figure 8> (Figure 7 in the original manuscript; revised)

<Figure 9> (Figure 8 in the original manuscript; revised)

<Figure 9 in the original manuscript > (deleted)

**Response:**

Corrected.

**Response:**

Corrected.

- If I get this right, you are assuming that the sediment trapping capability of the reservoir is negligible. However, over time the amount of storage decreases. To account for the role of sediment in reducing the reservoir capacity over time, I highly recommend the use of the Brune curve to account for it. If not Brune curve, please account for it in some fashion.

**Response:**

Thank you for this good and insightful suggestion. To address your comment, RI is redefined to incorporate the impact of sediment on reducing the reservoir capacity over time. In the revision, RI is defined as

$$\text{RI} = \sum_{i=1}^{N} \left( \frac{A_i}{A_T} \right) \cdot \left( \frac{(1 - \text{LR}_i) \cdot \text{RC}_i}{\bar{Q}} \right)$$

where $\text{LR}_i$ is the loss rate (%) of reservoir capacity in the $i$-th reservoir, due to the sediment deposition. RI is affected by the loss of the reservoir capacity but not too much (Figure S2), because the main reservoirs (i.e., Dangjiangkou and Ankang reservoirs) have a small loss rate no more than 15% (Table S1 and Figure S1). The estimation of $\text{LR}_i$ has been presented in Supplementary Information.

<Table S1> (newly-added)

<Figure S1> (newly-added)

<Figure S2> (newly-added)

Equation 1 is revised.

Equation S1 is newly-added.

Equation S2 is newly-added.

**Response:**

Corrected. Note that there is a modification of the name for MRI (revised as MARI) in the revised manuscript.

- Line 158: what does "inflexible" mean in this context?

**Response:**

We realize that the word "inflexible" may be inappropriate. Here, what we want to express is that the reservoir scheduling will have more constraints from the MARI. For example, when MARI with a large volume occurs and its timing is near the end of flood season, the reservoir with a operation strategy of increasing flood limit water level in stages will probably face a large peak of inflow and a insufficient residual capacity due to reservoir impounding. The above explaination will been added in the revised manuscript.

- Line 161: "where"

**Response:**

Corrected.

- In terms of predictors, the spatial distribution of rainfall is not really captured. I can think of situations in which the same basin-averaged rainfall will have very different effects if most of the rainfall occurs far or close to the outlet. How is this addressed here?

**Response:**

Thank you for your comments. To capture the spatial distribution of rainfall, for the MARI event, the distance ($L$) between the rainfall station with the maximum rainfall and the outlet have been considered. However, the results in Figure 5 (revised) show that for HZ station with the drainage area of 142056 km$^2$, there is a weak positive linear correlation (Pearson's r=0.24) between $L$ and AMDF, while for the AK station with the drainage area of 38600 km$^2$ and the HJG station 90491 km$^2$, the linear correlation between $L$ and AMDF is not significant. In the revised manuscript, this variable is considered as candidate to capture the spatial distribution of rainfall, but this variable is not selected for the calculation of RRCI, in consideration of both the non-significance correlation with AMDF of the study stations and the very complex fitting of 5-dimension copula.

- Line 185: "marginals"

**Response:**

Corrected.

- Line 204: "extensively concerned" is unclear.

**Response:**

Revised.

- Line 208: what does "obeys nonstationary distribution" mean?

**Response:**

We have revised this statement as follows:
* * *
Suppose that flood variable $Y_t$ obeys distribution $f_{Y_t}(y_t | \mathbf{\eta}_t)$ with the distribution parameters $\mathbf{\eta}_t = [\mu_t, \sigma_t, \xi]$.

- What about model selection based on the SBC index? Would you get a more parsimonious model?

**Response:**

Thank you for your suggestion. In the revised manuscript, we have added the SBC index. And the model selection is based on the SBC criterion. After adding four 2-parameter distributions (i.e., Lognormal, Gamma, Weibull, Gumbel), the detailed results have been summarized in Table 7 (newly-added).

- Line 254: I don't think this statement is correct, given that you would be able to say whether a more complex model should be selected over a more complex one, not if the fit is good or bad.

**Response:**

Thank you. This statement has been deleted. In the revised manuscript, the chi-square test has been replaced by the SBC criterion.

- Line 266: ", and was completed"

**Response:**

Corrected.

- Line 281: what is the definition of "timing"?

**Response:**

The timing is defined as the end time of MARI. In this study, the timing of MARI is equal to the occurrence time of AMDF in the year. In the revision, this definition of "timing" has been added.

**Response:**

In the revision, this sentence has been deleted.

**Response:**

Corrected.

- In fitting the copulas, the marginals were treating as stationary. Is this really the case? Please test for the presence of nonstationarities in the marginals of the predictors. If nonstationary, please account for it.

**Response:**

Thanks. In the revision, the change-points of the variables are tested by the Pettitt test, and then, if any, the marginal with the change-point will be addressed by the estimation method (Xiong et al., 2015). The results in Table S2 show that there are the significant change-points in the mean intensity ($I$) of the AK and HJG stations and in the volume ($V$) of the HJG station. Results in Table 5 (Table 4 in the original manuscript; revised) indicate that the consideration of the nonstationarity in these marginals makes little difference.

< Table S2> (newly-added)

- The role of the Mann-Kendall and Pettitt tests is unclear to me. First of all, the results are discussed at a very basic and superficial level. Also, if the response variable tends to change with time but because the predictors you have selected change over time as well, then whether Y is stationary or not is not very important; however, whether the relationship between predictors and predictand doesn't change over time becomes more relevant. Please fix this part.

**Response:**

Thanks. Here, the Mann-Kendall and Pettitt tests are indeed non-essential. We have deleted the Mann-Kendall and Pettitt tests in the revised manuscript.

It might be hard to demonstrate whether the relationship between predictors and predictand does not change over time in this study. But this issue can be covered, because under the Bayesian framework, the uncertainty of this relationship will be reflected in the posteriori distribution of model parameters.

- Lines 362-364: Please apply a correction to account for the fact you are performing multiple hypothesis testing

**Response:**

The correction has been made.

- Line 374: "explains"

**Response:**

Revised.

- Line 391: "for every certain multivariate MRI" is unclear.

**Response:**

Deleted.

- Line 402: "It is of interest"

**Response:**

Corrected.

- Line 404: "the remaining capacity of the reservoir"

**Response:**

Corrected.

- Line 409: "due to correspond to" is unclear

**Response:**

Revised.

- Line 423: "related to the construction"

**Response:**

Deleted.

- Line 427: "is weak"; "The comparison"

**Response:**

Deleted.

- Line 428: "indicates"

**Response:**

Corrected.

**Response:**

Corrected.

**Response:**

Corrected.

**Response:**

We have changed this in the revised manuscript as follows:

…the thick blue lines…

**Response:**

Corrected.

**Tables (revised and newly-added)**

Table 1. Summary of the probability density functions, the corresponding moments and the used link functions for nonstationary flood frequency analysis.

| Distributions | Probability density functions | Moments | Link functions |
|---|---|---|---|
| Gamma (GA) | $f_Y\left(y\mid\mu_t,\sigma_t\right)=\dfrac{(y)^{1/\sigma_t^2-1}}{\Gamma\left(1/\sigma_t^2\right)\left(\mu\sigma_t^2\right)^{1/\sigma_t^2}}\exp\left(-\dfrac{y}{\mu_t\sigma_t^2}\right)$  $y>0,\mu_t>0,\sigma_t>0$ | $E(Y)=\mu_t$  $Var(Y)=\mu_t^2\sigma_t^2$ | $g_1(\mu_t)=\ln(\mu_t)$  $g_2(\sigma_t)=\ln(\sigma_t)$ |
| Weibull (WEI) | $f_Y\left(y\mid\mu_t,\sigma_t\right)=\left(\dfrac{\sigma_t}{\mu_t}\right)\left(\dfrac{y}{\mu_t}\right)^{\sigma_t-1}\exp\left(-\left(\dfrac{y}{\mu_t}\right)^{\sigma_t}\right)$  $y>0,\mu_t>0,\sigma_t>0$ | $E(Y)=\mu_t\Gamma\left(1+1/\sigma_t\right)$  $Var(Y)=\mu_t^2\left[\Gamma\left(1+2/\sigma_t\right)-\Gamma^2\left(1+1/\sigma_t\right)\right]$ | $g_1(\mu_t)=\ln(\mu_t)$  $g_2(\sigma_t)=\ln(\sigma_t)$ |
| Lognormal (LOGNO) | $f_Y\left(y\mid\mu_t,\sigma_t\right)=\dfrac{1}{y\sigma_t\sqrt{2\pi}}\exp\left\{-\dfrac{\left[\log(y)-\mu_t\right]^2}{2\sigma_t^2}\right\}$  $y>0,-\infty<\mu_t<\infty,\sigma_t>0$ | $E(Y)=w^{1/2}\exp(\mu_t)$  $Var(Y)=w(w-1)\exp(2\mu_t)$  $w=\exp(\sigma_t^2)$ | $g_1(\mu_t)=\ln(\mu_t)$  $g_2(\sigma_t)=\ln(\sigma_t)$ |

[revised manuscript text omitted]

**Supplementary Information**

**Estimation of the loss rate (%) of reservoir capacity**

To estimate the variation of $\text{LR}_i$ over time, it is assumed that there is the same amount of sediment in each year. Then, $\text{LR}_i$ is estimated by

$$\text{LR}_i = \frac{n_i \cdot L_i^m}{\text{RC}_i} = \frac{n_i \cdot w_i^s \cdot \text{Te}_i}{\rho \cdot \text{RC}_i} \tag{S1}$$

where $n_i$ is the number of years which the $i$-th reservoir has been used, $L_i^m$ is the mean of annual loss of reservoir capacity (m$^3$) for the $i$-th reservoir, $w_i^s$ is the mean of annual inflow sediment mass (kg) for the $i$-th reservoir, $\rho$ is the density of the deposited sediment (kg/m$^3$) and $\text{Te}_i$ is the trap efficiency (%). Based on the Brune method (Brune, 1953; Mulu and Dwarakish, 2015), the trap efficiency is estimated with reservoir capacity-inflow ratio as follows

$$\text{Te}_i = 1 - \frac{0.5}{\sqrt{\text{RC}_i / I_i^m}} \tag{S2}$$

where $I_i^m$ is the mean of annual inflow volume in the $i$-th reservoir (m$^3$/day). The data in the previous literature (Guo, 1995; Hu, 2009; Liu, 2017) are collected to control the estimation errors of $L_i^m$. Please see Table S1.

**Table S1.** Summary for the calculation of the mean of annual loss of reservoir

capacity

| Reservoirs | $RC_i$ $(10^9\,\mathrm{m}^3)$ | $I_i^m$ $(10^9\,\mathrm{m}^3)$ | $w_i^s$ $(10^9\,\mathrm{kg})$ | $Te_i$ (%) | $L_i^m$ $(10^9\,\mathrm{m}^3)$ | |
|---|---|---|---|---|---|---|
| | | | | | From previous studies | From Eq.(S2)* |
| Shiquan | 0.566 | 11.73 | 12.6 | 88% | 0.006 | 0.008 |
| Ankang | 3.21 | 19.17 | 27.1 | 94% | - | 0.018 |
| Huanglongtan | 1.17 | 6.12 | 8.58 | 94% | 0.007 | 0.006 |
| Dangjiangkou | 34.0 | 39.48 | 59.8 | 97% | 0.044 | 0.042 |
| Yahekou | 1.32 | 1.09 | - | 98% | 0.007 | - |

\* $\rho = 1400\ \mathrm{kg/m}^3$

**Table S2.** Results of the change-point detection for the four MARI variables.

| Variables | AK | | HJG | | HZ | |
|---|---|---|---|---|---|---|
| | change-point | p-value* | change-point | p-value | change-point | p-value |
| *M* | 1976 | 1.037 | 1989 | 0.371 | 1971 | 1.278 |
| *I* | 1987 | **0.031** | 1985 | **0.009** | 1990 | 0.080 |
| *V* | 2009 | 0.746 | 1984 | **0.042** | 1984 | 0.769 |
| *T* | 1992 | 1.180 | 1984 | 0.986 | 1984 | 1.367 |

*Less than 0.05 is considered significant.

**Figure S1**. Interannual variation of loss rate of reservior capacitity for each reservoir in the study area.

[Figure]

**Figure S2**. Impact of reservoir capacity loss on RI for AK, HJG and HZ
stations.

[Figure]

**Figure S3**. Preliminary analysis of the snowmelt influences on the streamflow in the catchment upstream the AK station. (a) is the total number of times for AMDF in each month; (b) is the monthly average temperature; (c) is the monthly average streamflow; and (d) is the monthly average precipitation.

[Figure]

**Glossary and Notation:**

$\alpha_0, \alpha_1, \beta_0, \beta_1$ : parameters of nonstationary model.

$A_i$ : total basin area upstream of the $i$-th reservoir.

$A_T$ : total basin area upstream of the gauge station.

AIC: Akaike information criterion.

AK: Ankang (gauging station).

AMDF: annual maximum daily flow (series).

CDF: Cumulative distribution function

$d$ : dimension of copulas.

df : freedom degree.

GA: Gamma distribution

GEV: Generalized Extreme Value distribution.

GEV_S23: nonstationary GEV distribution with the S23 scenario.

GML: generalized maximum likelihood (method).

GU: Gumbel distribution.

HJG: Huangjiagang (gauging station).

HZ: Huang zhuang (gauging station).

$I$ : intensity, the mean of daily rainfall in MARI.

IDW: Inverse distance weighting method.

IRI: impounded runoff index, a ratio of reservoir capacity to mean annual runoff.

$\hat{l}$ : maximized likelihood of the model object.

$L$ : distance, the distance between the rainfall center and the outlet.

LOGNO: Lognormal distribution.

$\mathrm{LR}_i$ : loss rate (%) of total storage capacity of the $i$-th reservoir due to the sediment deposition.

$\mu_t$ : mu parameter of the distribution functions used.

$M_c$ : length of the Markov chain.

$M$ : maximum, the maximum of daily rainfall in MARI.

MARI: multiday antecedent rainfall input.

MCMC: Markov chain Monte Carlo.

ML: maximum likelihood (method).

$n$ : number of data points.

$N$ : total number of reservoirs upstream of the gauge station.

OR-JEP: OR-joint exceedance probability.

$P_{\text{MARI}}^{\vee}$ : OR-joint exceedance probability.

$\boldsymbol{\theta}_i$ : parameter vector of the $i$-th marginal distribution.

$\boldsymbol{\theta}_c$ : copula parameter vector.

$\boldsymbol{\theta}$ : parameter vector of the whole $n$-dimensional distribution.

$\boldsymbol{\theta}_{\text{GEV\_S23}}$ : parameters of the GEV_S23 model.

$\hat{\boldsymbol{\theta}}_{\text{GEV\_S23}}^{i}$ : an estimation for the parameters of the GEV_S23 model.

$\bar{Q}$ : mean annual runoff.

RRCI: rainfall-reservoir composite index.

RI: reservoir index.
RC: reservoir capacity.

$\text{RC}_i$ : total storage capacity of the $i$-th reservoir.

$\sigma_t$ : sigma parameter of the distribution functions used.

S0: constant scenario.

S1: RI-dependent scenarios.

S2: RRCI-dependent scenarios.

SBC: Schwarz Bayesian criterion.

$T$ : timing, the end time of MARI in the year.

$u_i$ : univariate marginal distribution of $X_i$.

$V$ : volume, the total of daily rainfall in MARI.

WEI: Weibull distribution.

$\xi$ : shape parameter of the Generalized Extreme Value distribution.

$X_1, X_2, ... X_d$ : scheduling-related MARI variables.

**Reply to Referee #2**

*General Comments:*

The manuscript presents downstream flood frequency analysis framework using the annual maximum daily flows (AMDF). Joint cumulative probability of multiple rainfall variables (maximum, intensity, volume and timing) are considered as multiday rainfall input (MRI) and employed in C-vine copula model. Flood frequency model is defined by nonstationary generalized extreme value (NGEV) distribution model including uncertainty deliberation with Bayesian approach. Rainfall reservoir composite index (RRCI) is proposed and used to quantify the reservoir effects as covariate for expression of distribution parameters. According to the different metrics, the results of the proposed method outperforms typical reservoir index (RI) based flood frequency model which only accounts reservoir capacity and mean annual runoff. I believe the study is quite interesting for the readership of the journal and contributing to better modeling of downstream flood peak mechanism. The model results give reasonable outcomes and can be useful for regions where large reservoirs are located. The manuscript deserves publication after a major revision considering my below comments.

**Response:**

Thank you very much for the good summary and the positive evaluation of the paper. All your valuable comments have been carefully addressed in the revision. Please see our point to point replay below.

- Language needs some refinements before publication. Also, there are some typos and repeated sentences, which make hard to follow and disturb the readability. It would be nice to revise the manuscript totally by dividing long sentences and eliminating the repeated ones. Same tense should be used (is or was) thought the text.

**Response:**

Thanks for your kind suggestion. We have carefully revised the text to correct all issues about typos, unclear long sentences, repeated sentences and different tenses.

- Studies dealing with downstream hydrograph alterations caused by dams are not discussed enough in the literature.

**Response:**

In the first paragraph of the modified version, we have added literature review on studies dealing with downstream hydrograph alterations caused by dams as follows:

….In the literature, the significant hydrological alterations caused by reservoirs are demonstrated in the many areas of the world. Graf (1999) showed that the dams have greater effects on the streamflow than the global climate change in America. Benito and Thorndycraft (2005) reported various significant changes of the

pre- and post-dam hydrologic regimes (e.g., minimum and maximum flows over different durations) across the United States. Batalla et al. (2004) demonstrated an evident reservoir-induced hydrologic alteration in the North-Eastern Spain. Yang et al. (2008) indicated the spatial variability of the hydrological regimes alteration caused by the reservoirs in the middle and lower Yellow River, China. Mei et al. (2015) found that the Three Gorges Dam, the largest dam in the world, has significantly changed the downstream hydrological regimes. In recent years, the cause-effect mechanisms of the downstream flood peak reduction were also investigated in some literature (Ayalew et al., 2013; 2015; Volpi et al., 2018). For example, Volpi et al. (2018) suggested that for a single reservoir, the downstream flood peak reduction is mainly dependent on its position along the river, its spillway and its storage capacity based on a parsimonious instantaneous unit hydrograph-based model. These studies have revealed that it is crucial to assess the impacts of reservoirs on downstream flood regimes for the success of downstream flood risk management.

**Newly added literature**

Ayalew, T.B., Krajewski W.F., Mantilla R., 2015. Insights into Expected Changes in Regulated Flood Frequencies due to the Spatial Configuration of Flood Retention Ponds. Journal of Hydrologic Engineering, 20(10): 04015010.

Graf, W.L., 1999. Dam nation: A geographic census of American dams and their large‐scale hydrologic impacts. Water resources research, 35(4): 1305-1311.

Mei, X., Dai, Z., Van Gelder, P.H.A.J.M., and Gao, J., 2015. Linking Three Gorges Dam and downstream hydrological regimes along the Yangtze River, China. Earth and Space Science, 2(4): 94-106.

Volpi, E., Di Lazzaro M., Bertola M., Viglione A., and Fiori A., 2018. Reservoir Effects on Flood Peak Discharge at the Catchment Scale. Water Resources Research, 54(11): 9623-9636.

Yang, T., Zhang Q., Chen Y.D., Tao X., Xu C.Y., and Chen X., 2008. A spatial assessment of hydrologic alteration caused by dam construction in the middle and lower Yellow River, China. Hydrological Processes: An International Journal, 22(18): 3829-3843.

- As stated in Lines 45-49, there are several factors for the generation of the floods. Authors focused on meteorological conditions, but also indicating the importance of hydrological conditions such as snow cover. The elevation range of the study area is quite wide (13 – 3493 m) and most upstream reservoirs (especially Ankang gauge) should be dominated by snowmelt. The response of the basin will be complex compared to lower altitude basins. There is not much information about the assessment of the snowmelt contribution of the catchments and their effects on operational decisions. It is also interesting to see that linear correlations between the timing variable of multivariate MRI and AMDF give lowest (almost zero) Pearson r for AK gauge in Figure 5. Would snowmelt be a reason for this? If this is the case, maybe RRCI is not enough to explain downstream peak floods for the regions where

reservoirs fed by snowmelt? Temperature data can also be effective to estimate flood peaks in such cases. I believe this situation should be clarified.

**Response:**

Thank you for this comment. Although the elevation range of the study area is quite wide (13–3493 m), the study area is a rainfall-dominated area and the snowmelt contribution is quite limited. This area has a warm temperate semi-humid continental monsoon climate. The temperature in the basin is not much different from upstream to downstream. The timing of flood is the main rainfall period from June to September (Figure S3a, c and d). And the winter is warm as shown in Figure S3b. It is indicated that the rainfall is the main contribution for floods. The above information will be added in the revised manuscript.

<Figure S3> (newly-added)

[Figure]

Figure S3. Preliminary analysis of the snowmelt influences on the streamflow in the catchment upstream the AK station. (a) is the total number of times for AMDF in each month; (b) is the monthly average temperature; (c) is the monthly average streamflow; and (d) is the monthly average precipitation.

The reason why AK gauge has a weak linear correlation between the timing variable of multivariate MRI and the annual maximum flood in Figure 5 is probably that there is a non-significant effect of the staged operation of the reservoirs on the floods. The reservoirs upstream of AK station have a smaller capacity than HJG and HZ stations. There may be a random variation of the remaining storage capacity in each staged period of the flood season for AK station. Thus, in the long term, the reduction of the peaks of AK station tends to be not different in each staged period of the flood season.

And Figure S3 has been added in Supplementary Information.

- In Data Section, the explanation of reservoir data is based on only their capacities. There is not much information how they are operated. For example, for what purposes they are operated, or how their reservoir pools are divided (flood control, conservation, dead storage etc.)?

**Response:**

Agree. In the revision, more information on the reservoir operation has been added as follows:

… The Danjiangkou Reservoir in central China's Hubei province is the largest one in this basin, and was completed by 1967. As a multi-purpose reservoir, it mainly aims to supply water and control floods, and is also used for electricity generation and irrigation. The reservoir has the total storage capacity of 21.0 billion m3, the dead storage capacity of 7.23 billion m3, the effective storage capacity of 10.2 billion m3, and the flood control capacity of 7.72 billion m3. After the Danjiangkou Dam Extension Project in 2010, the Danjiangkou Reservoir gained an additional capacity of 13.0 billion m3 and an extra flood control storage capacity of 3.3 billion m3. Besides, this reservoir is operated by the strategy of staged increasing flood limit water level in the flood control season (Zhang et al., 2009).

**Newly added literature**

Zhang L., Xu J., Huo, J., Chen J., 2009. Study on Stage Flood Control Water Level of Danjiangkou Reservoir. Journal of Yangtze River Scientific Research Institute, 26 (3): 13-14. (In Chinese)

- It is not clear why inverse distance weighting (IDW) is selected for areal distribution of the rainfall records. The catchments are large and elevation ranges in between 13-3493 m, so that this method may not be representative especially for mountainous regions.

**Response:**

The reason why IDW is selected is that IDW is a handy method. Due to both the data limitation (16 sites) and the unstable relationship between rainfall and elevation, it is hard for us to demonstrate whether the other methods (e.g., the Kriging methods) will be better. In this study, the rainfall records from all national meteorological stations in the study area are used. The precision of areal rainfall with the IDW method should be able to meet the requirement in the study. In the revision, the error of estimation of areal rainfall has been discussed to remind readers in the discussion as follows:

…The areal-averaged MARI is based on the records of 16 rainfall stations with the IDW method; the estimation error of areal-averaged rainfall may be transferred to the OR-JEP estimation error; the additional rainfall site data and spatial distribution information are needed to reduce the OR-JEP estimation error. Nonetheless, the good performance of downstream flood frequency modeling demonstrates the MARI samples still remain representative in this study.

- Maybe it would be better to call "downstream flood frequency analysis" rather than "flood frequency analysis" throughout the manuscript?

**Response:**

Agree. We have made a revision for this throughout the manuscript.

- Variation of RI and RRCI are quite different for AK gauge station in Figure 6. Please state the reason

**Response:**

Thanks. For AK gauging station, there is a quite difference in the variation of RI and RRCI. This is because RRCI is dependent on both RI and the OR-joint exceedance probability (OR-JEP). As shown in Figure 2 (revised), in spite of a low value of RI, the MARI with a high OR-JEP value can get a high RRCI. In fact, the reservoir effect on the downstream flood is great under the condition of the fewer constraints (high OR-JEP values) from MARI. Thus, it is expected that RRCI can reflect a real reservoir effect more than RI.

- Uncertainty of flood estimates are greater in AK stations (Figure 8) compared to the others. The reason should be explained.

**Response:**

Thanks for this suggestion. The uncertainty range of AK station is larger than HJG and HZ stations. The possible explanation to the larger uncertainty range is that the sample size (1993-2015) of the regulated floods at AK station is smaller, and, furthermore, the dependent relationship between RRCI and AMDF at AK station is weaker. This explanation has been added in the revised manuscript.

- Discussion section is comparatively short to conclusion part. In general the paper describes a usable approach but the main weakness is insufficient discussion of the available results. I mean, it is stated that the downstream flood regime should be altered by upstream reservoirs and the magnitude of flood peaks are reduced due to the storage capacity of them. This is expected in such a reservoir system by analyzing long period AMDF values (see Figure 7, observed AMDF). Rather, the author should elaborately clarify GEV model results in Discussion part. Main results should be given under discussion, and conclusion should briefly summarize them. Considering these, I guess these two sections should be totally revised.

**Response:**

Thank you very much for this comment. Discussion and Conclusion have been totally revised. Discussion has been put in the Section 4.4 as follows:

**4.4 Discussion**

[revised manuscript text omitted]

- Figure and tables are appropriate. However, I have some doubts about the usefulness of Figure 9 to illustrate the reservoir effects on flood risk. It is not combining the results of the frequency model. It is not clear for what reason this figure stands for especially at the end of the result section. (I suggest removing this figure, as it is a bit confusing in terms of central theme of the paper). If authors would like to include it, I suggest them to re-organize its location through the manuscript and revise the descriptions to make it more clear (in Lines 387-395).

**Response:**
Agree. In order to highlight the central theme of the paper, Figure 9 has been deleted in the revised manuscript.

*Specific comments:*

-There are too much abbreviation in the manuscript. Maybe a glossary would be useful for the readers.

**Response:**
Thanks for this suggestion. In Supplementary Information, we have added a glossary for these abbreviations as follows:

**Glossary and Notation:**

$\alpha_0, \alpha_1, \beta_0, \beta_1$ : parameters of nonstationary model.

$A_i$ : total basin area upstream of the $i$-th reservoir.

$A_T$ : total basin area upstream of the gauge station.

AIC: Akaike information criterion.

AK: Ankang (gauging station).

AMDF: annual maximum daily flow (series).

CDF: Cumulative distribution function

$d$ : dimension of copulas.

df : freedom degree.

GA: Gamma distribution

GEV: Generalized Extreme Value distribution.

GEV_S23: nonstationary GEV distribution with the S23 scenario.

GML: generalized maximum likelihood (method).

GU: Gumbel distribution.

HJG: Huangjiagang (gauging station).

HZ: Huang zhuang (gauging station).

$I$ : intensity, the mean of daily rainfall in MARI.

IDW: Inverse distance weighting method.

IRI: impounded runoff index, a ratio of reservoir capacity to mean annual runoff.

$\hat{l}$ : maximized likelihood of the model object.

$L$ : distance, the distance between the rainfall center and the outlet.

LOGNO: Lognormal distribution.

$\mathrm{LR}_i$ : loss rate (%) of total storage capacity of the $i$-th reservoir due to the sediment deposition.

$\mu_t$ : mu parameter of the distribution functions used.

$M_c$ : length of the Markov chain.

$M$ : maximum, the maximum of daily rainfall in MARI.

MARI: multiday antecedent rainfall input.

MCMC: Markov chain Monte Carlo.

ML: maximum likelihood (method).

$n$ : number of data points.

$N$ : total number of reservoirs upstream of the gauge station.

OR-JEP: OR-joint exceedance probability.

$P_{\mathrm{MARI}}^{\vee}$ : OR-joint exceedance probability.

$\boldsymbol{\theta}_i$ : parameter vector of the $i$-th marginal distribution.

$\boldsymbol{\theta}_c$ : copula parameter vector.

$\boldsymbol{\theta}$ : parameter vector of the whole $n$-dimensional distribution.

$\boldsymbol{\theta}_{\text{GEV\_S23}}$ : parameters of the GEV_S23 model.

$\hat{\boldsymbol{\theta}}^i_{\text{GEV\_S23}}$ : an estimation for the parameters of the GEV_S23 model.

$\bar{Q}$ : mean annual runoff.

RRCI: rainfall-reservoir composite index.

RI: reservoir index.
RC: reservoir capacity.

$\text{RC}_i$ : total storage capacity of the $i$-th reservoir.

$\sigma_t$ : sigma parameter of the distribution functions used.

S0: constant scenario.

S1: RI-dependent scenarios.

S2: RRCI-dependent scenarios.

SBC: Schwarz Bayesian criterion.

$T$ : timing, the end time of MARI in the year.

$u_i$ : univariate marginal distribution of $X_i$.

$V$ : volume, the total of daily rainfall in MARI.

WEI: Weibull distribution.

$\xi$ : shape parameter of the Generalized Extreme Value distribution.

$X_1, X_2, ...X_d$ : scheduling-related MARI variables.

- Line 49, what is "nature extreme flow"?

**Response:**
For clarity, we have changed this sentence in the revised manuscript as follows:

In general, without reservoirs, the flood extremes downstream of most rain-dominated basins are mainly related to the extreme rainfall in the drainage area…

- Lines 50-52, what about the operational targets and other constraints?

**Response:**
Thanks. Our statement exists imprecise. We have rephrased it as follows:

However, with reservoirs, the downstream flood regimes should be totally different due to upstream flood control scheduling.

- Lines 52-54, requires more up-to-date references.

**Response:**

In the revision, we have added literature review on studies dealing with downstream hydrograph alterations caused by dams. Please see the response for "- Studies dealing with downstream hydrograph alterations caused by dams are not discussed enough in the literature.".

- Lines 76-78, even a small reservoir could be very complex to derive operational strategies and a lot of detailed information might be required. I am not sure about this classification. Please consider revising this part.

**Response:**

Agree. A modification of this statement has been made as follows:

The continuous simulation method can explicitly account for the reservoir effects on flood in the hypothetical case. However, it is difficult to apply this approach to the most real cases (Volpi et al., 2018), because the simplifying assumptions of this approach are just satisfied in a few of basins with single small reservoir. Furthermore, even if the basins meet the simplifying assumptions, the detailed information required in this approach are probably unavailable.

- Line 96, what type of uncertainty?

**Response:**

The uncertainty of flood estimates is associated with the uncertainty of the parameters estimates. For clarity, we have revised this sentence as follows:

…Bayesian inference can get multiple estimates, forming a posterior distribution of model parameters. Thus, the Bayesian method is able to conveniently describe the uncertainty of flood estimates associated with the uncertainty of model parameters.

- Line 84, which "previous studies"?

**Response:**

Thanks. The correction has been made as follows:

Thus, previous studies (Adlouni et al., 2007; Ouarda and El‐Adlouni, 2011) have used the nonstationary Generalized Extreme Value distribution (NGEV) to describe nonstationary maxima series.

- Line 108, it is a bit vague what do you mean by "more accurate effects of reservoirs?"

**Response:**

Thanks. For clarity, a modification of this sentence has been made as follows:

The precision and accuracy in the quantitative analysis of the reservoir effects on the downstream floods need to be improved further.

- Lines 115-117, please refer to Bayesian method in the objectives.

**Response:**

Agree. In the revision, Bayesian method has been referred in the objectives.

- Line 143, what do you mean by "more precise effects of reservoirs"?

**Response:**

Thanks. We have deleted this sentence.

- Line 146, please briefly explain "multiday rainfall input".

**Response:**

Note that there is a modification of the name for MRI (revised as MARI) in the revised manuscript. In the revision, the brief explanation has been added as follows:

In addition to the reservoir capacity, multiday antecedent rainfall input (MARI), i.e., an event of the continuous multi-day multivariate rainfall forming the inflow event which will be regulated to become downstream extreme flow by the reservoir system is a key constraint for the scheduling of the reservoir system.

- Lines 147-150, It is a bit confusing whether scheduling related multivariate (SRMR) and MRI are same or not? Could you give more detail for their explanations.

**Response:**

Thanks.

SRMR and MRI are different. All variables of SRMR are selected from the variables of MRI. The SRMR variables is the scheduling related MARI variables. In the revised manuscript, for clarity, SRMR is deleted and MARI has been accurately described.

- Line 155, why OR-joint exceedance probability is selected as measure function?

**Response:**

We need a rainfall index to measure the effect of the antecedent rainfall on the reservoir operation. The OR-joint exceedance probability (OR-JEP) is the probability that any one of the given set of values ($x_1, x_2, ..., x_d$) for the scheduling-related MARI variables will be exceeded. The lower this probability, the greater effects on reservoir operation the MARI has, and then, it is expected that the downstream floods possibly obtain relative large values. The above explanation has been added in the revised manuscript.

- Line 158, what do you mean by "reservoir scheduling is more inflexible"?

**Response:**

We realize that the word "inflexible" may be inappropriate. Here, what we want to express is that the reservoir scheduling will have more constraints from the MARI. For example, when MARI with a large volume occurs and its timing is near the end of flood season, the reservoir with a operation strategy of increasing flood limit water

level in stages will probably face a large peak of inflow and a insufficient residual capacity due to reservoir impounding. The above explaination will been added in the revised manuscript.

- Lines 170-172, selected four variables require more explanation.

**Response:**
Agree. The more detailed explanation has been added as follows:

In this study, to add the antecedent rainfall effects into the new indicator of reservoir effects, the five variables are considered to describe MARI, i.e., the maximum $M$ (the maximum of daily rainfall in MARI), the intensity $I$ (the mean of daily rainfall in MARI), the volume $V$ (the total of daily rainfall in MARI), the timing $T$ (the end time of MARI in the year) and the distance $L$ (the distance between the rainfall center and the outlet). The reason that $M$, $I$, $V$, and $L$ are selected is that these variables will determine the peak, the total volume and the peak appearance time of the inflow event. The variable $T$ is utilized to capture the information of the remaining storage capacity, due to the staged operation strategies in the flood season for some reservoirs. For the operation strategy of increasing flood limit water level in stages, it is expected that if the timing of MARI is near the end of flood season, the downstream AMDF will be less affected by reservoirs, because of less remaining capacity in this period.

- Line 208, it is not clear "obeys nonstationary distribution". Please revise.

**Response:**
The statement has been revised as follows:

Suppose that flood variable $Y_t$ obeys distribution $f_{Y_t}\left(y_t \mid \mathbf{\eta}_t\right)$ with the distribution parameters $\mathbf{\eta}_t = \left[\mu_t, \sigma_t, \xi\right]$.

- Line 280-286. The sentence is too long and difficult to understand. Please separate and revise.

**Response:**
Thanks. The sentence has been separated and revised.

**Response:**

Thanks. In the revision, this sentence has been deleted.

- Line 303-304, it is not clear what do you mean by "(e.g., special extreme MRI may limit or reduce the effects of the reservoir)."

**Response:**
In the revision, this sentence has been deleted.

- Line 314-315, please describe and relate calculated Spearman correlations in the text, otherwise remove them.

**Response:**
The description and relation for the calculated Spearman correlations has been added in the revised text.

-Lines 338-339, please clarify "special rainfall events"

**Response:**
In the revision, this phrase has been deleted.

- Lines 412-413, please mention future studies in Conclusion part, not under Discussion.

**Response:**
In the revision, we have followed your suggestion.

- Line 429, it is not clear what do you mean by "some rare multivariate MRI still would produce lower values of RRCI than that of RI". Please revise it.

**Response:**
Thanks. This sentence has been revised in the revised manuscript.

***Technical corrections:***

- Figure 1. The caption should be "The flowchart of nonstationary covariate-based flood frequency analysis with a rainfall-reservoir composite index (RRCI)

**Response:**
Thank you for this comment. We have revised the caption according to your suggestion.

- Figure 7. In the caption, "thick blue" should be "thick blue line".

**Response:**
Corrected.

- Table 2. It would be better to not to duplicate "Dangjiangkou reservoir" and remove first row. The details should be given in the text only.

**Response:**
Corrected.

- Line 26, please revise "of the previous study"

**Response:**
Revised.

- Line 35, please revise "What's more"

**Response:**
Revised.

- Lines 62-63, please revise the sentence.

**Response:**
Revised.

– In Line 73, it is stated three model components but not clear which of them are ordered since only two are given?

**Response:**
Thanks. This is a mistake. We have corrected this sentence as follows:

… In the first approach, the regulated flood time series can be simulated by using three model components, i.e., the stochastic rainfall generator, the rainfall-runoff model and the reservoir flood operation module which includes the reservoir storage capacity, the size of release structures and the operation rules.…

- Line 76-78, too long sentence and hard to follow. Please revise it.

**Response:**
Thanks. We have revised this sentence. Please see the response for "- Lines 76-78, even a small reservoir could be very complex…"

- Line 119, please explain AMDF.

**Response:**
The explanation has been added in the revision as follows:

To quantify the effects of reservoirs on the frequency of the annual maximum daily flow series (AMDF) downstream of reservoirs,…

- Line 114 and Line 120, "RRIC" should be "RRCI"

**Response:**
Corrected.

- Line 115, "to calculate" should be replaced with "to develop"

**Response:**
Corrected.

- Line 139, "the Eq. (1)" should be replaced with "Eq. (1)"

**Response:**
Corrected.

[revised manuscript text omitted]
(\mathbf{\theta}_{\text{GEV\_S23}})$ and observations $\mathbf{D}$ have the likelihood

263   $l(\mathbf{D}|\mathbf{\theta}_{\text{GEV\_S23}})$, then the posterior probability distribution $p(\mathbf{\theta}_{\text{GEV\_S23}}|\mathbf{D})$ can be calculated with Bayes'

264   theorem, as follow

[revised manuscript text omitted]

---

## Author Response (AR2)

**Dear Editor,**

On behalf of my co-authors, I appreciate you and two reviewers very much for all the positive and constructive comments on our manuscript entitled "Assessing the impacts of reservoirs on downstream flood frequency by coupling the effect of scheduling-related multivariate rainfall into an indicator of reservoir effects"(ID: hess-2019-42).

The comments made by reviewer #1 have been addressed. According to your suggestion, we sent the manuscript to a professional English editor. Now, the language of the manuscript has been improved. A point-by-point response to the comments and the relevant changes made in the manuscript are presented as appendix to this letter. The revised manuscript with all revisions marked in red color is appended at the end of this document.

We hope the revision is acceptable, and I look forward to hearing from you soon.

With all best wishes.

Yours,

Lihua Xiong

On behalf of my co-authors

State Key Laboratory of Water Resources and Hydropower Engineering Science

Wuhan University, Wuhan 430072, China

E-mail: xionglh@whu.edu.cn

Telephone: +86-13871078660

Fax: +86-27-68773568

**Reply to Referee #1**

- Line 32: "Although most…"

**Response:**

Revised.

- Lines 35-37: This sentence reads a bit convoluted.

**Response:**

Thanks. We have rephrased this.

- Line 39: "estimated values"

**Response:**

Corrected.

- Lines 58-59: "in northeastern Spain"

**Response:**

Corrected.

- Lines 59-60: it is unclear what "indicated" really refers to.

**Response:**

Revised.

- Line 71: "series is stationary"

**Response:**

Corrected.

- Line 74: "complicated"

**Response:**

Corrected.

- Lines 109-113: with ML you get a point estimate, but also the associated standard error, which allows you to computer confidence intervals on the parameters.

**Response:**

Thank the reviewer for this comment. We have rephrased the two sentences to correct the inappropriate statements.

However, the ML method for a nonstationary distribution model can lead to very high quantile estimator variances when using numerical techniques to solve the likelihood function when using the small sample (Adlouni et al., 2007).

- Lines 220-223: can you rephrase this?

**Response:**

Thanks. We have rephrased this.

- Line 274: this sentence reads incomplete.

**Response:**

Revised.

- Line 343: I guess you mean that the p-value is smaller than 0.05. Please correct.

**Response:**

Corrected.

- Lines 344-347: I don't think this is appropriate as the relationship between L ad AMDF may become significant after you account for one or more of the other variables. Please fix it.

**Response:**

Thanks. This is a thoughtful comment. The statement of "indicating that the location of the rainfall may not be significantly related to the AMDF of the outlet" has been deleted. In addition, we understand that L may be important to explain the formation of flood peak in some basins when non-uniform rainfall in space occurs. However, to reduce the cost and complexity in this study, we have to use the simple method (linear relationship) to reduce the dimensionality for fitting copula.

**Response:**

Corrected.

**Response:**

Thanks. We have added the explanation of computing the uncertainties around the 99th percentile as follows:

In nonstationary case, the 95% credible interval in the t-year is calculated by a set of the (99th) percentile estimations which are obtained by the flood distribution functions determined by the values of both covariate in that year and posterior parameter samples.

**Assessing the impacts of reservoirs on downstream flood frequency by coupling the**

**effect of scheduling-related multivariate rainfall into an indicator of reservoir**

**effects**

Bin Xiong[1], Lihua Xiong[1*], Jun Xia[1], Chong-Yu Xu[1, 3], Cong Jiang[2], Tao Du[4]

1. State Key Laboratory of Water Resources and Hydropower Engineering Science, Wuhan

University, Wuhan 430072, China

2. School of Environmental Studies, China University of Geosciences, Wuhan 430074, China

3. Department of Geosciences, University of Oslo, P.O. Box 1022 Blindern, N-0315 Oslo, Norway

4. Bureau of Hydrology, Changjiang Water Resources Commission, Wuhan 430010, China

*Corresponding author:*

Lihua Xiong, PhD, Professor

State Key Laboratory of Water Resources and Hydropower Engineering Science

Wuhan University, Wuhan 430072, China

E-mail: xionglh@whu.edu.cn

Telephone: +86-13871078660

Fax: +86-27-68773568

**Abstract**

Many studies have shown that downstream flood regimes have been significantly altered by upstream reservoir operation. Reservoir effects on the downstream flow regime are normally performed by comparing the pre-dam and post-dam frequencies of certain streamflow indicators, such as floods and droughts. In this study, a rainfall-reservoir composite index (RRCI) is developed to precisely quantify reservoir impacts on downstream flood frequency under a framework of a covariate-based nonstationary flood frequency analysis using the Bayesian inference method. The RRCI is derived from a combination of both a reservoir index (RI) for measuring the effects of reservoir storage capacity and a rainfall index. More precisely, the OR-joint exceedance probability (OR-JEP) of certain scheduling-related variables selected out of five variables that describe the multiday antecedent rainfall input (MARI) is used to measure the effects of antecedent rainfall on reservoir operation. Then, the RI-dependent or RRCI-dependent distribution parameters and five distributions, the gamma, Weibull, lognormal, Gumbel, and generalized extreme value, are used to analyze the annual maximum daily flow (AMDF) of the Ankang, Huangjiagang, and Huangzhuang gauging stations of the Hanjiang River, China. A phenomenon is observed that although most of the floods that peak downstream of reservoirs have been reduced in magnitude by upstream reservoirs, some relatively large flood events still have occurred, such as at the Huangzhuang station in 1983. The results of nonstationary flood frequency analysis show that, in comparison to the RI, the RRCI that combines both the RI and the OR-JEP resulted a much better explanation for such phenomena of flood occurrences downstream of reservoirs. A Bayesian inference of the 100-year return level of the AMDF shows that the optimal RRCI-dependent distribution, compared to the RI-dependent one, results in relatively smaller estimated values. However, there exist exceptions due to some low OR-JEP values. In addition, it provides a smaller uncertainty range. This study highlights the necessity of including antecedent rainfall effects, in addition to the effects of reservoir storage capacity, on reservoir operation to assess the reservoir effects on downstream flood frequency. This analysis can provide a more comprehensive approach for downstream flood risk management under the impacts of reservoirs.

**Keywords**: nonstationary flood frequency analysis; downstream floods; reservoir; antecedent rainfall; Bayesian inference; Hanjiang River

**1 Introduction**

River floods are generated by various complex nonlinear processes involving physical factors including "hydrological pre-conditions (e.g., soil saturation, snow cover), meteorological conditions (e.g., amount, intensity, and the spatial and temporal distribution of rainfall), runoff generation processes, and river routing (e.g., superposition of flood waves in the main river and its tributaries)" (Wyżga et al., 2016). In general, without reservoirs, the downstream flood extremes of most rain-dominated basins are primarily related to extreme rainfall events in the drainage area. However, with reservoirs, the downstream flood regimes should be totally different due to upstream flood control scheduling. In the literature, the significant hydrological alterations caused by reservoirs have been demonstrated in the many areas of the world. Graf (1999) showed that dams have more significant effects on streamflow in America than global climate change. Benito and Thorndycraft (2005) reported various significant changes across the United States in pre- and post-dam hydrologic regimes (e.g., minimum and maximum flows over different durations). Batalla et al. (2004) demonstrated an evident reservoir-induced hydrologic alteration in northeastern Spain. Yang et al. (2008) demonstrated the spatial variability in hydrological regimes alterations caused by the reservoirs in the middle and lower Yellow River in China. Mei et al. (2015) found that the Three Gorges Dam, the largest dam in the world, has significantly changed downstream hydrological regimes. In recent years, the cause-effect mechanisms of downstream flood peak reductions were also investigated by some researchers (Ayalew et al., 2013; Ayalew et al., 2015; Volpi et al., 2018). For example, Volpi et al. (2018) suggested that for a single reservoir, the downstream flood peak reduction was primarily dependent on its position along the river, its spillway, and its storage capacity based on a parsimonious instantaneous unit hydrograph-based model. These studies have revealed that it is crucial to assess the impacts of reservoirs on downstream flood regimes for the success of downstream flood risk management.

Flood frequency analysis is the most common technique used by hydrologists to gain knowledge of flood regimes. In conventional or stationary frequency analyses, a basic hypothesis is that hydrologic time series maintains stationarity, i.e., "free of trends, shifts, or periodicity (cyclicity)" (Salas, 1993).

However, in many cases, observations of changes in flood regimes have demonstrated that this strict assumption is invalid (Kwon et al., 2008; Milly et al., 2008). Nonstationarity in downstream flood regimes of dams makes frequency analyses more complicated. Actually, the frequency of downstream floods of dams is closely related to upstream flood operations. In recent years, there have been many attempts to link flood generating mechanisms and reservoir operations to the frequency of downstream floods (Gilroy and Mccuen, 2012; Goel et al., 1997; Lee et al., 2017; Liang et al., 2017; Su and Chen,

2018; Yan et al., 2017).

Previous studies have meaningfully increased the knowledge about reservoir-induced nonstationarity of downstream hydrological extreme frequencies (Ayalew et al., 2013; López and

Francés, 2013; Liang et al., 2017; Magilligan and Nislow, 2005; Su and Chen, 2018; Wang et al., 2017;

Zhang et al., 2015). There are two main approaches to incorporate reservoir effects into flood frequency analyses: the hydrological model simulation approach and the nonstationary frequency modeling approach. In the first approach, the regulated flood time series can be simulated using three model components: the stochastic rainfall generator, the rainfall-runoff model, and the reservoir flood operation module, which includes the reservoir storage capacity, the size of release structures, and the operation rules. The continuous simulation method can explicitly account for the reservoir effects on floods in the hypothetical case. However, it is difficult to apply this approach to a majority of real cases (Volpi et al., 2018) because the simplifying assumptions of this approach are only satisfied in a few of basins with single small reservoirs. Furthermore, even if the basins meet the simplifying assumptions, the detailed information required in this approach is likely unavailable. Thus, our attention is focused on the second method, the nonstationary frequency modeling approach. Nonstationary distribution models have been widely used to deal with the nonstationarity of extreme value series. In nonstationary distribution models, the distribution parameters are expressed as the functions of covariates to determine the conditional distributions of extreme value series. According to extreme value theory, the maxima series can generally be described using the generalized extreme value distribution (GEV). Thus, previous studies (El Adlouni et al., 2007; Ouarda and El‐Adlouni, 2011) have used the nonstationary generalized extreme value distribution to describe the nonstationary maxima series. Scarf (1992)

modeled the changes in the location and scale parameters of the GEV over time using the power function relationship. Coles (2001) introduced several time-dependent structures (e.g., trend, quadratic, and change-point) into the location, scale, and shape parameters of the GEV. El Adlouni et al. (2007)

provided a general nonstationary GEV model with an improved parameter estimate method. In recent years, "generalized additive models for location, scale, and shape" (GAMLSS) have been widely used in nonstationary hydrological frequency analyses (Du et al., 2015; Jiang et al., 2014; López and Francés,

2013; Rigby and Stasinopoulos, 2005; Villarini et al., 2009). GAMLSS provides various candidate distributions for frequency analysis, such as Weibull, gamma, Gumbel, and lognormal distributions.

However, the GEV has been rarely involved in the candidate distributions of GAMLSS. In terms of a parameter estimation method for the nonstationary distribution model, the maximum likelihood (ML)

method is the most common parameter estimate method. However, the ML method for a nonstationary distribution model can lead to very high quantile estimator variances when using numerical techniques to solve the likelihood function when using a small sample (El Adlouni et al., 2007). El Adlouni et al.

(2007) developed the generalized maximum likelihood (GML) method and demonstrated that the GML

method had better performance than the ML method in all their cases. Ouarda and El‑Adlouni (2011)

introduced the Bayesian nonstationary frequency analysis. The Bayesian inference can obtain multiple estimates, forming a posterior distribution of model parameters. Thus, the Bayesian method is able to conveniently describe the uncertainty of flood estimates associated with the uncertainty of model parameters.

In the nonstationary frequency modeling approach, a dimensionless reservoir index (RI) was proposed by López and Francés (2013) as an indicator of reservoir effects, and it generally is used as a covariate for the expression of the distribution parameters (e.g., location parameter) (Jiang et al., 2014;

López and Francés, 2013). Liang et al. (2017) modified the reservoir index by replacing the mean annual runoff in the expression of the RI with the annual runoff. Therefore, the modified reservoir index can reflect the impact of reservoirs on downstream flood extremes under various total inflow conditions each year. However, the precision and accuracy in the quantitative analysis of the reservoir effects on downstream floods need to be further improved. In fact, the effects of reservoirs may be closely related not only to the static reservoir storage capacity but also to the dynamic reservoir operations associated with multiple characteristics, such as the peak, the intensity, and the total volume of the multiday antecedent rainfall input (MARI), not just annual runoff.

Therefore, the aim of the study is to develop an indicator, referred to as the rainfall-reservoir composite index (RRIC), that combines the effects of reservoir storage capacity and the MARI on reservoir operation. This indicator is then used as a covariate to assess the reservoir effects on the downstream flood frequency. The specific objectives of this study are (1) to develop the RRCI; (2) to compare the RRCI with the RI using a covariate-based nonstationary flood frequency analysis; and (3)

to obtain the downstream flood estimation and its uncertainty based on the optimal nonstationary distribution using the Bayesian inference.

**2 Methods**

To quantify the effects of reservoirs on the frequency of the annual maximum daily flow series (AMDF) downstream of reservoirs, a three-step framework (Figure 1), termed the covariate-based flood frequency analysis using the RRIC as a covariate, was established. In this section, the methods of this framework are introduced. First, a reservoir index (RI) is defined by additionally considering the effects of reservoir sediment deposition on the storage capacity. Second, the RRCI is developed by combining the RI and a rainfall index. Next, the C-vine copula model is used to construct and calculate the rainfall index. Finally, the nonstationary distribution models that utilize the Bayesian estimation are clarified.

                                <Figure 1>

**2.1 Reservoir index (RI)**

Intuitively, the larger the reservoir capacity relative to the flow of a downstream gauging station, the greater the possible effects of the reservoir on the streamflow regime. To quantify reservoir-induced alterations to the downstream streamflow regime, Batalla et al. (2004) proposed an impounded runoff index (IRI), which is a ratio of reservoir capacity ($RC$) to (unimpaired) mean annual runoff ($\bar{Q}$) at the gauge station, indicated as $IRI = RC/\bar{Q}$. For a single reservoir, the IRI is a good indicator of the extent to which a reservoir alters streamflow. To analyze the effects of a multi-reservoir system on the downsream flood frequency, López and Francés (2013) proposed a dimensionless reservoir index. In this study, we additionally considered the effects of reservoir sediment deposition on the reservoir capacity. In accordance with López and Francés (2013), the reservoir index (RI) for a downstream gauging station is defined as

$$RI = \sum_{i=1}^{N}\left(\frac{A_i}{A_T}\right)\cdot\left(\frac{\left(1-LR_i\right)\cdot RC_i}{\bar{Q}}\right), \tag{1}$$

where $N$ is the total number of reservoirs upstream of the gauge station; $A_i$ is the total basin area upstream of the $i$-th reservoir; $A_T$ is the total basin area upstream of the gauge station; $RC_i$ is the total storage capacity of the $i$-th reservoir; and $LR_i$ is the loss rate (%) of $RC_i$ due to the sediment deposition (Appendix A). Equation (1) indicates that for a reservoir system consisting of small- and middle-sized reservoirs, the RI for the downstream gauging station is generally less than one. However, for a system with some large reservoirs, such as multi-year regulating storage reservoirs, the RI of the downstream gauging station near this system may be close to one or higher.

**2.2 Rainfall-reservoir composite index (RRCI)**

In addition to the reservoir capacity, the multiday antecedent rainfall input (MARI), which is an event of continuous multi-day multivariate rainfall that forms the inflow event that will be regulated by the reservoir system to become the downstream extreme flow, is a key constraint for scheduling the reservoir system. In this study, to add the antecedent rainfall effects into the new indicator of reservoir effects, five variables were used to describe the MARI: the maximum $M$ (the maximum daily rainfall in the MARI); the intensity $I$ (the mean daily rainfall in the MARI); the volume $V$ (the total daily rainfall in the MARI); the timing $T$ (the end time of MARI during that year); and the distance $L$ (the distance between the rainfall center and the outlet). The reason that $M$, $I$, $V$, and $L$ were selected is because these variables will determine the peak, the total volume, and the peak appearance time of an inflow event.

The variable, $T$, is utilized to capture information regarding the remaining storage capacity, due to staged operation strategies during flood season used in some reservoirs. For the operation strategy that consists of increasing the flood limit water level in stages, it is expected that if the timing of the MARI

is near the end of the flood season, the downstream AMDF will be less affected by reservoirs. This is because of the lesser remaining capacity during this period. The MARI variables that are selected to construct the new indicator are hereafter referred to as the scheduling-related MARI variables (denoted as $X_1, X_2, ..., X_d$ ). The extraction procedure of the MARI is detailed in section 3.2.

A new index is proposed in this study called the rainfall-reservoir composite index (RRIC) to more comprehensively assess the effects of reservoirs on floods by incorporating the effects of the

MARI. This index is defined as

$$
\text{RRCI} = \begin{cases} \left( P_{\text{MARI}}^{\vee} \left( \bigcup_{i=1}^{d} (X_i > x_i) \right) \right)^{(1/\text{RI}-1)} , 0 < \text{RI} \leq 1 \\ \text{RI}, \text{RI} > 1 \end{cases}, \tag{2}
$$

where $P_{\text{MARI}}^{\vee}$ is the OR-joint exceedance probability (OR-JEP); that is the probability that any one of the given set of values ( $x_1, x_2, ..., x_d$ ) for the scheduling-related MARI variables will be exceeded. Here, the

OR-JEP acts as a rainfall index for measuring the MARI effects. The lower this probability, the greater effects on reservoir operation the MARI has. Then, it is expected that downstream floods could possibly obtain relatively large values, and vice versa. Figure 2 illustrates the relationship in Equation (2), which shows that the RRCI is conditional on both the OR-JEP and the RI. Equation (2) can then be expressed as

$$
\text{RRCI} = \begin{cases} \left( 1 - F(x_1, x_2, ...x_d) \right)^{(1/\text{RI}-1)} , 0 < \text{RI} \leq 1 \\ \text{RI}, \text{RI} > 1 \end{cases}, \tag{3}
$$

where $F(\cdot)$ is the cumulative distribution function (CDF) that determines the dependence relationship of the variables. The expectation of the RRCI is as follows:

$$E(\text{RRCI}) = \int_{\mathbb{R}^d} \left(1 - F(x_1, x_2, \ldots x_d)\right)^{(1/\text{RI}-1)} dF(x_1, x_2, \ldots x_d) = \text{RI} . \qquad (4)$$

In addition, for the OR case, the following is true:

$$P_{\text{MARI}}^{\vee}\left(\bigcup_{i=1}^{d}(X_i > x_i)\right) \geq P_{\text{MARI}}^{\vee}(X_i > x_i) . \qquad (5)$$

Equations (3) and (5) indicate that, in addition to the RI, the RRCI is related to the number and the dependence relationship of the scheduling-related MARI variables. To obtain a reasonable RRCI, the unrelated MARI variables should not be incorporated. In this study, the number of MARI variables that were incorporated was no more than four to avoid a "dimension disaster" in modeling their dependence.

To select the scheduling-related MARI variables, a three-step selection procedure was used that included the following. (1) Selecting four variables from the five MARI variables by testing the significance of the Pearson correlation between the MARI variables and the AMDF. (2) Calculating the

RRCI for all possible subsets of the four variables using the $d$-dimensional ($d = 1, 2, 3, 4$) copulas. Then finally (3) identifying the variables by using the highest rank correlation coefficient between the RRCI

and the AMDF. The construction method of the $d$-dimensional ($d = 2, 3, 4$) distribution $F(x_1, x_2, \ldots, x_d)$

is described in the following subsection.

                              <Figure 2>

**2.3 C-vine Copula model**

In this subsection, a c-vine Copula model for the construction of the continuous $d$-dimensional distribution $F(x_1, x_2,..., x_d)$ is clarified. The Sklar's theorem (Sklar, 1959) showed that for a continuous

$d$-dimensional distribution, the one-dimensional marginals and dependence structure can be separated, and the dependence can be represented using a copula formula as follows:

$$F\left(x_1, x_2,...x_d \,|\, \boldsymbol{\theta}\right) = C\left(u_1, u_2,..., u_d \,|\, \boldsymbol{\theta}_c\right), u_i = F_{X_i}\left(x_i \,|\, \boldsymbol{\theta}_i\right) , \tag{6}$$

where $u_i$ is the univariate marginal distribution of $X_i$; $C(\cdot)$ is the copula function; $\boldsymbol{\theta}_c$ is the copula parameter vector; $\boldsymbol{\theta}_i$ is the parameter vector of the $i$-th marginal distribution; and $\boldsymbol{\theta} = \left(\boldsymbol{\theta}_c, \boldsymbol{\theta}_1, \boldsymbol{\theta}_2,..., \boldsymbol{\theta}_d\right)$

is the parameter vector of the entire $n$-dimensional distribution. Thus, the construction of $F(x_1, x_2,...x_d)$

can be separated into two steps: first is the modeling of the univariate marginals; and second is the modeling of the dependence structure. For the first step, the empirical distribution is used as the univariate marginal distributions, and the change-points of the variables are tested using the Pettitt test (Pettitt, 1979). Then, if there are any, the marginal and the change-point will be addressed using the estimation method (Xiong et al., 2015). Then, for the second step, the copula construction for the dependence modeling is based on the pair-copula construction method, which has been widely used in previous research (Aas et al., 2009; Xiong et al., 2015). According to Aas et al. (2009), the joint density function $f(x_1, x_2,..., x_d)$ is written as

$$f\left(x_1, x_2, ..., x_d \mid \boldsymbol{\theta}\right) = c_{1...n}\left(u_1, u_2, ..., u_d \mid \boldsymbol{\theta}_c\right) \prod_{i=1}^{d} f_{X_i}\left(x_i \mid \boldsymbol{\theta}_i\right), u_i = F_{X_i}\left(x_i \mid \boldsymbol{\theta}_i\right) \ . \tag{7}$$

The $n$-dimensional copula density $c_{1...d}\left(u_1, u_2, ..., u_d\right)$, which can be decomposed into $d(d-1)/2$

bivariate copulas, corresponding to a c-vine structure, is given by

$$c_{1...d}\left(u_1, u_2, ..., u_d \mid \boldsymbol{\theta}_c\right) = \prod_{j=1}^{d-1} \prod_{i=1}^{d-j} c_{j,i+j \mid 1,...,j-1}\left(F\left(u_j \mid u_1, ..., u_{j-1}\right), F\left(u_{i+j} \mid u_1, ..., u_{j-1}\right) \Big| \boldsymbol{\theta}_{j,i \mid 1,...,j-1}\right), \tag{8}$$

where $c_{j,i+j \mid 1,...,j-1}$ is the density function of a bivariate pair copula, and $\boldsymbol{\theta}_{j,i \mid 1,...,j-1}$ is a parameter vector of the corresponding bivariate pair copula. Therefore, the marginal conditional distribution is

$$F\left(u_{i+j} \mid u_1, ..., u_{j-1}\right) =$$
$$\frac{\partial C_{i+j,j-1 \mid 1,...,j-2}\left(F\left(u_{i+j} \mid u_1, ..., u_{j-2}\right), F\left(u_{j-1} \mid u_1, ..., u_{j-2}\right) \Big| \boldsymbol{\theta}_{i+j,j-1 \mid u_1,...,u_{j-2}}\right)}{\partial F\left(u_{j-1} \mid u_1, ..., u_{j-2}\right)}, \tag{9}$$
$$j = 2, ..., d-1; \ i = 0, ..., n-j$$

where $C_{i+j,j-1 \mid 1,...,j-2}$ is a bivariate copula distribution function. The maximum dimensionality covered in this study was four. Thus for a four-dimensional copula (of which the decomposition is shown in Figure

3), the general expression of Equation (8) is

$$c_{1234}\left(u_1, u_2, u_3, u_4 \mid \boldsymbol{\theta}_c\right) = c_{12}\left(u_1, u_2 \mid \boldsymbol{\theta}_{12}\right) c_{13}\left(u_1, u_3 \mid \boldsymbol{\theta}_{13}\right) c_{14}\left(u_1, u_4 \mid \boldsymbol{\theta}_{14}\right) \cdot$$
$$c_{23 \mid 1}\left(F\left(u_2 \mid u_1\right), F\left(u_2 \mid u_1\right) \mid \boldsymbol{\theta}_{23 \mid 1}\right) c_{24 \mid 1}\left(F\left(u_2 \mid u_1\right), F\left(u_4 \mid u_1\right) \mid \boldsymbol{\theta}_{24 \mid 1}\right) \cdot \tag{10}$$
$$c_{34 \mid 12}\left(F\left(u_3 \mid u_1, u_2\right), F\left(u_4 \mid u_1, u_2\right) \mid \boldsymbol{\theta}_{34 \mid 1}\right)$$

                      <Figure 3>

**240** **2.4 Covariate-based nonstationary frequency analysis using the Bayesian estimation**

**241** The covariate-based extreme frequency analysis has been widely used (Villarini et al., 2009;

**242** Ouarda and El‑Adlouni, 2011; López and Francés, 2013; Xiong et al., 2018). According to these

**243** studies, five distributions, gamma (GA), Weibull (WEI), lognormal (LOGNO), Gumbel (GU), and the

**244** generalized extreme value (GEV), were used as candidate distributions in this study. In addition, their

**245** density functions, the corresponding moments, and the used link functions are shown in Table 1. In the

**246** following, the nonstationary distribution models based on Bayesian estimation are developed for a

**247** covariate-based flood frequency analysis.

**248** <Table 1>

**249** Suppose that flood variable, $Y_t$, obeys the distribution $f_{Y_t}\left(y_t \mid \mathbf{\eta}_t\right)$ with the distribution

**250** parameters $\mathbf{\eta}_t = \left[\mu_t, \sigma_t, \xi\right]$. In this study, only the distribution parameters $\mu_t$ and $\sigma_t$ were allowed to be

**251** dependent on covariates because the shape parameter of the GEV is sensitive to the quantile estimation

**252** of rare events. According to the linear additive formulation of the generalized additive models for

**253** location, scale, and shape (GAMLSS) (Rigby and Stasinopoulos, 2005; Villarini et al., 2009), seven

**254** nonstationary scenarios for the formulas of the two distribution parameters, $\mu_t$ and $\sigma_t$, were

**255** investigated, as shown in Table 2. The constant scenario (S0) included one scenario (both $\mu_t$ and $\sigma_t$

**256** are constants). The RI-dependent scenarios (S1) included three scenarios: S11 ($\mu_t$ is RI-dependent and

**257** $\sigma_t$ is constant), S12 ($\mu_t$ is constant and $\sigma_t$ is RI-dependent), and S13 (both $\mu_t$ and $\sigma_t$ are RI- dependent). In addition, the RRCI-dependent scenarios (S2) including S21, S22, and S23 are similar to

S11, S12, and S13, respectively.

<Table 2>

In the following, the Bayesian inference is introduced. The GEV_S23 (representing the nonstationary GEV distribution with the S23 scenario) model was used as an example, and the model parameter vector $\mathbf{\theta}_{\text{GEV\_S23}} = [\alpha_0, \alpha_1, \beta_0, \beta_1, \xi]$ was used as the estimate. The Bayesian method was used to estimate $\mathbf{\theta}_{\text{GEV\_S23}}$. Let the prior probability distribution be $\pi(\mathbf{\theta}_{\text{GEV\_S23}})$, and the observations, $\mathbf{D}$, have the likelihood $l(\mathbf{D}|\mathbf{\theta}_{\text{GEV\_S23}})$. Then the posterior probability distribution $p(\mathbf{\theta}_{\text{GEV\_S23}}|\mathbf{D})$ can be calculated using Bayes' theorem as follows:

$$p(\mathbf{\theta}_{\text{GEV\_S23}}|\mathbf{D}) = \frac{l(\mathbf{D}|\mathbf{\theta}_{\text{GEV\_S23}})\pi(\mathbf{\theta}_{\text{GEV\_S23}})}{\int_{\Omega} l(\mathbf{D}|\mathbf{\theta}_{\text{GEV\_S23}})\pi(\mathbf{\theta}_{\text{GEV\_S23}})d\mathbf{\theta}_{\text{GEV\_S23}}} \propto l(\mathbf{D}|\mathbf{\theta}_{\text{GEV\_S23}})\pi(\mathbf{\theta}_{\text{GEV\_S23}}), \qquad (11)$$

where the integral is the normalizing constant, and $\mathbf{\Omega}$ is the entire parameter space. The obvious difference between the Bayesian method and the frequentist method is that the Bayesian method considers the parameters $\mathbf{\theta}_{\text{GEV\_S23}}$ to be random variables. In addition, the desired distribution of the random variables can be obtained using a Markov chain that can be constructed using various Markov chain Monte Carlo (MCMC) algorithms (Reis Jr and Stedinger, 2005; Ribatet et al., 2007) to process

Equation (11). In addition, in this study, the Metropolis-Hastings algorithm was used (Chib and

Greenberg, 1995; Viglione et al., 2013), which was done with the aid of the R package "MHadaptive"

(Chivers, 2012). A beta distribution function was used with the parameters $u = 6$ and $v = 9$, which were suggested by Martins and Stedinger (2000) and Martins and Stedinger (2001) as the prior distribution on the shape parameter $\xi$. For the other model parameters, $\alpha_0, \alpha_1, \beta_0, \beta_1$, the prior distributions were set to non-informative (flat) priors. There are two advantages of the Bayesian method. First, as noted by El

Adlouni et al. (2007), this method allows the addition of other information, such as historical and regional information, by defining the prior distribution. Second, the Bayesian method can provide an explicit way to account for the uncertainty of parameters estimates. In the nonstationary case in the *t*- year, the 95% credible interval for the estimation of the flood quantile corresponding to a given probability, $p$, can be obtained from a set of stable parameters estimations, $\hat{\boldsymbol{\theta}}_{\text{GEV\_S23}}^i (i = 1, 2, ..., M_c)$, in which $M_c$ is the length of the Markov chain.

The procedure of model selection can identify which of the five distributions is optimal, and which of the seven nonstationary scenarios is optimal. If all the distribution parameters are identified as constants (S0), this process will be a stationary frequency analysis. To select the optimal model, the

Schwarz Bayesian criterion (SBC) (Schwarz, 1978) for each fitted model object is calculated by the following:

$$\text{SBC} = -2\ln\left(\hat{l}\right) + \ln\left(n\right) * \text{df} , \tag{12}$$

where $\ln\left(\hat{l}\right)$ is the maximized log-likelihood of the model object; df is the freedom degree; and $n$ is the number of data points. SBC has a larger penalty on the over-fitting phenomenon than the Akaike information criterion (AIC) (Akaike, 1974). The model object with the lower SBC is preferred. The worm plot and the QQ plot were employed to check whether the model represented the data well.

**3 Study area and data**

**3.1 Study area**

Hanjiang River (Figure 4), with the coordinates of 30° 30′ –34° 30′ N, 106° 00′ –114°

00′ E and a catchment area of 159,000 km², is the largest tributary of the Yangtze River, China. This area has a warm temperate, semi-humid, continental monsoon climate. The temperature in the basin is not much different from upstream to downstream. Although the elevation range of the study area is quite wide (13–3493 m), the study area is a rainfall-dominated area, and the snowmelt contribution is quite limited. The Ankang gauging station was used as an example. The timing of the AMDF is primarily during the major rainfall period from June to September (Figure S3a, c, and d). In addition, the winter is warm, with mean temperature values of more than 2 ℃, as shown in Figure S3b. Since

1960, many reservoirs have been completed in the Hanjiang basin. Information of the five major reservoirs is shown in Table 3, including the longitude, latitude, control area, time for completion, and capability. The Danjiangkou Reservoir in central China's Hubei province is the largest one in this basin and was completed by 1967. As a multi-purpose reservoir, it primarily aims to supply water and control floods, and it is also used for electricity generation and irrigation. The reservoir has a total storage capacity of 21.0 billion m$^3$, a dead storage capacity of 7.23 billion m$^3$, an effective storage capacity of

10.2 billion m$^3$, and a flood control capacity of 7.72 billion m$^3$. After the Danjiangkou Dam Extension

Project in 2010, the Danjiangkou Reservoir gained an additional capacity of 13.0 billion m$^3$ and an extra flood control storage capacity of 3.3 billion m$^3$. In addition, this reservoir is operated using the strategy of staged increases in the flood limit water level during the flood control season (Zhang et al., 2009).

                        <Figure 4>

                        <Table 3>

**3.2 Data**

The assessment analysis of reservoir effects on flood frequency utilized streamflow data, reservoir data, and rainfall data. The annual maximum daily flood series (AMDF) was extracted from the daily streamflow records of the three gauges in the Hanjiang River basin; namely the Ankang (AK)

station with a drainage area of 38,600 km$^2$, the Huangjiagang (HJG) station with a drainage area of

90,491 km$^2$, and the Huangzhuang (HZ) station with a drainage area of 142,056 km$^2$. The streamflow and reservoir data were provided by the Hydrology Bureau of the Changjiang Water Resources

Commission, China (http://www.cjh.com.cn/en/index.html). The annual series of the maximum ($M$), the intensity ($I$), volume ($V$), the timing ($T$), and the distance ($L$) were extracted from the daily streamflow data to describe the MARI. Note that the timing of the MARI is equal to the occurrence time of the AMDF during the year. The MARI is a real-averaged event, and any two consecutive days of areal rainfall values in the MARI required more than 0.2 mm. Daily areal rainfall was calculated using the inverse distance weighting (IDW) method based on rainfall records from 16 stations (shown in

Figure 4). These rainfall data were downloaded from the National Climate Center of the China

Meteorological Administration (source: http://www.cma.gov.cn/). For the AK and HZ gauging stations, all the records were available from 1956 to 2015, while the HJG gauging station only had records available from 1956 to 2013.

**4 Results and discussion**

**4.1 Identification of reservoir effects**

To confirm the impact of reservoirs on the annual maximum daily flow (AMDF) in the study area, the mean and standard deviation of the AMDF before and after the construction of the two large reservoirs, the Danjiangkou reservoir (1967) upstream of the HJG and HZ stations and the Ankang reservoir (1992) upstream of the AK, HJG, and HZ stations, were compared. According to Table 4, the mean and standard deviation of the AMDF of the AK, HJG, and HZ stations were significantly reduced.

By using the HJG station as an example, the mean of the AMDF (1992–2013) is 4139 $m^3$/s, which is only 0.28 times 14,951 $m^3$/s (1956–1966), and the standard deviation is 4074 $m^3$/s, approximately 0.52

times 7896 $m^3$/s (1956–1966).

                                                              <Table 4>

Figure 5 presents the linear correlation between the five MARI variables (i.e., the maximum, $M$; the intensity, $I$; volume, $V$; the timing, $T$; and the distance $L$) and the AMDF. It was found that for $M$, $I$,

$V$, and $T$, except for $T$ in the AK station, the Pearson correlation coefficients between these four variables and the AMDF range from 0.27 to 0.71 (p-value<0.05), indicating that these four variables are significantly related to the AMDF. However, there is a Pearson correlation coefficient of no more than

0.24 between $L$ and the AMDF for each of the stations. Thus, $L$ was excluded from the calculation of the RRCI. A further analysis of the reservoir effects on the downstream AMDF will be performed in the following sections.

                                                                                                                   <Figure 5>

**4.2 Results for the rainfall-reservoir composite index (RRCI)**

To obtain the annual values of the RRCI, the RI was estimated first. The RI was affected by the loss of the reservoir capacity, but not to a great extent (Figure S2). This happened because the main reservoirs (Dangjiangkou and Ankang reservoirs) had a small loss rate of no more than 15% (Table S1

and Figure S1).

The C-vine copula model was applied to calculate the OR-JEP of the scheduling-related MARI

variables. In the modeling of the univariate marginal, the marginals of the intensity ($I$) of the AK and the HJG stations and the volume ($V$) of the HJG station were revised to deal with their significant change-points (Table S2). To identify the scheduling-related variables from $M$, $I$, $V$, and $T$, the RRCI for all the possible subsets of *M*, *I*, *V*, and *T* was calculated and compared. The Pearson, Kendall, and

Spearman correlation coefficients between the RRCI and the AMDF are listed in Table 5. Note that the entire decomposition structure of the C-vine copula for each RRCI of the same station was determined by the ordering of the variables of each subset (shown in the cells of the first column in Table 5). Figure

3 shows an example for the decomposition structure of the 4-dimensional copula. As shown in the first row in Table 5, there is a negative correlation between the AMDF and the RI for each station. The values of the Pearson correlation coefficients between the AMDF and the RI for the AK, HJG, and HZ

stations are -0.37, -0.55, and -0.53, respectively, demonstrating that there is a significant relation between the reservoir storage capacity and the reduction in the AMDF. For each station, with the exception of the RRCI of one-dimensional case, the values of the Pearson, Kendall, and Spearman correlation coefficients between the RRCI and the AMDF are higher than between the RI and the

AMDF. According to the highest Kendall correlation, the scheduling-related variables for the AK

station were *M*, *I*, *V* and *T*. Those for the HJG station were *I* and *T*, and those for the HZ station were *I*,

*V,* and *T*.

                                                  \<Table 5\>

Table 6 shows the results of the copula modeling of the scheduling-related variables using the aid of the R package "VineCopula" (https://CRAN.R-project.org/package=VineCopula). Note that for each bivariate pair in the third column in Table 6, three one-parameter bivariate Archimedean copula families (i.e., the Gumbel, Frank, and Clayton copulas) (Nelsen, 2006) were used to select from. As shown in Table 6, the results of the Cramer-von Mises test (Genest et al., 2009) shows that all the C- vine copula models passed the test at a significance level of 0.05. This result indicated that these models were effective for simulating the joint distribution of the scheduling-related variables for the three stations. Finally, the variation in the RI and the RRCI over time is displayed in Figure 6. It can be seen that for each station, after reservoir construction, in most cases, the annual values of the RRCI are larger (close to 1) than those of the RI. In contrast, in few cases, such as in 1983 at the HZ and HJG stations, the RRCI values were lower than the RI values.

<Figure 6>

<Table 6>

**4.3 Flood frequency analysis**

A nonstationary flood frequency analysis using the RRCI or the RI as the covariate was performed to investigate how the reservoirs affected the downstream flood frequency. A summary of results of fitting the nonstationary models to the flood data is shown in Table 7. Based on the SBC, the lowest values indicate that the best models for the AK, HJG, and HZ stations are the nonstationary WEI

distribution with S23, the nonstationary GA distribution with S21, and the nonstationary WEI

distribution with S21, respectively, hereafter referred to as WEI_S23, GA_S21, and WEI_S21, respectively. Note that for any one of the five distributions (GA, WEI, LOGNO, GU, and GEV), the

RRCI-dependent scenario had a lower SBC value than the RI-dependent scenario for each gauging station. Furthermore, for the RI-dependent and RRCI-dependent scenarios, using the HZ station as an example, the optimal formulas of the two distribution parameters, $\mu_t$ and $\sigma_t$, are given as follows:

(1) WEI_S11

$$\mu_t = \exp(9.94 - 2.79\text{RI})$$
$$\sigma_t = \exp(0.49)$$

$\qquad\qquad$ (13)

(2) WEI_S21

$$\mu_t = \exp(9.92 - 1.42\text{RRCI})$$
$$\sigma_t = \exp(0.73)$$

$\qquad\qquad$ (14)

It was found that in Equations (13) and (14), there were negative estimates of -2.79 and -1.42 for $\alpha_1$, respectively, revealing the decreasing degree of the frequency and magnitude of downstream floods due to the reservoir effects.

$\qquad$ Figure 7 compares the stationary scenario (S0), the RI-dependent scenario (S1), and the RRCI- dependent scenario (S2) of the same optimal distributions that explain all the flood values and the several largest flood values for each station. The QQ plots (Figure 7a1–c1) show that overall, the RRCI- dependent scenario more adequately captured the entire empirical quantiles (particularly the smallest and largest empirical quantiles) than the two other scenarios for each station. Furthermore, as shown in

Figure 7a2–c2, for the seven largest floods (observed) of each station, the RRCI-dependent scenario produced lower quantile residuals than the two other scenarios.

                          <Table 7>

                          <Figure 7>

Figure 8 shows the performance of the best models: WEI_S23 for the AK station, GA_S21 for the HJG station, and WEI_S21 for the HZ station. The points in the worm plots in Figure 8 are within the 95% confidence interval, indicating that the selected models are reasonable. In addition, according to the centile curves plots in Figure 8, the AMFD series is well fitted by the best models. Undoubtedly, with the incorporation of the effects of the MARI, the RRCI-dependent scenario well captured the presence of nonstationarity in the downstream flood frequency. The case of the HZ station was used for the analysis (Figure 8c1). After the construction of the Danjiangkou Reservoir (1967), due to reservoir operation, most of the values of the AMDF had been reduced in magnitude by this reservoir. However, some relatively large flood events still occurred several times, such as 25,600 m$^3$/s in 1983 and 19,900

m$^3$/s in 1975. Obviously, this phenomenon of flood occurrences was well explained by the RRCI.

                          <Figure 8>

The 100-year return levels at a 95% credible interval from WEI_S23 and WEI_S13 for the AK

station, GA_S21 and GA_S11 for the HJG station, and WEI_S21 and WEI_S11 for the HZ station are presented in Figure 9. For each station, compared to the optimal RI-dependent distribution, the optimal

RRCI-dependent distribution provided a lower 100-year return level. However, there existed exceptions.

In addition, after the construction of the main reservoir, the uncertainty range of the AK station was larger than that of the HJG and HZ stations. A possible explanation for the larger uncertainty range was that the sample size (1993–2015) of the regulated floods at the AK station was smaller. Furthermore, the dependent relationship between the RRCI and the AMDF at the AK station was weaker.

<Figure 9>

**4.4 Discussion**

The long-term variation in the AMDF series (Figure 8) indicates that the upstream reservoirs had evidently altered the downstream flood regimes. As an example, since the completion of the

Danjiangkou reservoir in 1967, the flood magnitude of the HZ station was evidently reduced overall.

This is consistent with the results of the effects of reservoirs on the hydrological regime in this area found in previous studies (Cong et al., 2013; GUO et al., 2008; Jiang et al., 2014; Lu et al., 2009). In this study, it was found that there was a significant difference between downstream floods affected by the same reservoir system (with the same RI value). In most cases, relatively small downstream floods were obtained. However, it is of interest to note that there still occurred unexpected large downstream floods in a few cases, in spite of a large RI value. For example, most values of the AMDF in the HZ

station have been less 10,000 m$^3$/s since 1967, but the values of the AMDF in 1983 and in 1975 were

25,600 m$^3$/s and 19,900 m$^3$/s, respectively. These unexpected large downstream floods were probably related to the MARI effects on reservoir operation. The five largest (unexpected) floods since 1967 and the corresponding values of the scheduling-related MARI variables in the HZ station are shown in Table

8. It was found that the largest floods from 1967 to 2015 occurred in 1983. For this flood event, the

MARI was a rare event (with an OR-JEP value of 0.435 ranking the second in 1967–2015) due to the largest mean intensity ($I = 20.2$ mm) and the second latest occurrence ($T = 281$). Surprisingly, all the timing values of the MARI for these five unexpected downstream floods showed high rankings (2–9th).

These timing values were near the end (approximately the 300th day of the year) of the flood control period (July–October) in this area. Actually, near the end of the major flood control period, the storage capacity should be decreased. This is because according to the operation rules of the Danjiangkou reservoir (Zhang et al., 2009), there is a staged increasing flood limit water level during the flood control season. One important cause for those unexpected large downstream floods was probably that the remaining storage capacity at the end of flood season was not sufficient to reduce some late floods.

Therefore, in addition to the storage capacity of reservoirs, the MARI effects should be indispensably considered when attempting to accurately quantify the effects of the reservoir on downstream floods.

                                    <Table 8>

With the combination of both the RI and the OR-JEP, the RRCI had a significant difference from RI (Figure 6). With a few exceptions, the RRCI values were higher than the RI values. This indicates that the real reservoir impact may be underestimated by the RI in most cases. Moreover, the RI

will also probably overestimate the real reservoir impact in a few cases because of not considering special rainfall events (i.e., the MARI with low values of the OR-JEP). The results of the covariate- based nonstationary flood frequency analysis (Table 7 and Figures 7 and 8) demonstrate that, compared to the RI-dependent scenario, the RRCI-dependent scenario for the optimal nonstationary distribution more completely captured the presence of nonstationarity in the downstream flood frequency. Therefore, the RRCI might be a useful index for accessing the reservoir effects on downstream flood frequency.

Finally, the estimation errors of the OR-JEP should be noted. (1) Only those MARI samples that corresponded to the timing of the AMDF were included to estimate the OR-JEP. This means that some extreme MARI samples that corresponded to the non-maximum flow were not included, resulting in an estimation error for the OR-JEP. To reduce this error, it might be worth considering the use of the peaks-over-threshold sampling method. (2) The areal-averaged MARI was based on the records from 16

rainfall stations using the IDW method. The estimation error of the areal-averaged rainfall can be transferred to the OR-JEP estimation error. Additional rainfall site data and spatial distribution information were needed to reduce the OR-JEP estimation error. Nonetheless, the good performance of the downstream flood frequency model results demonstrated that the MARI samples still remained representative in this study.

**5 Conclusions**

Accurately assessing the impact of reservoirs on downstream floods is an important issue for flood risk management. In this study, to evaluate the effects of reservoirs on the downstream flood frequency of the Hanjiang River, the rainfall-reservoir composite index (RRCI) was derived from Equation (2), which considers the combination of the reservoir index (RI) and the OR-joint exceedance probability (OR-JEP) of scheduling-related rainfall variables. The main findings are summarized as follows: (1) The magnitude of the downstream flood events has been reduced by the reservoir system in the study area. However, the long-term variation in the observed AMDF series showed that despite the large reservoirs, unexpected large flood events still occurred several times, such as at the Huangzhuang station in 1983. One important cause of the unexpected large floods at the Huangzhuang station may have been related to the operation strategy of staged increases in the flood limit water level of the Danjiangkou reservoir. (2) According to the results of the covariate-based nonstationary flood frequency analysis for each station, compared to the optimal RI-dependent distribution, the optimal RRCI-dependent distribution more completely captured the presence of nonstationarity in the downstream flood frequency. (3) Furthermore, in estimating the 100-year return level for each station, the optimal RRCI-dependent distribution provided a lower 100-year return level, but there existed exceptions. In addition, it provided a smaller uncertainty range associated with the uncertainty of the model parameter.

Consequently, this study demonstrated the necessity of including the antecedent rainfall effects, in addition to the effects of reservoir storage capacity, on reservoir operation to assess the reservoir effects on downstream flood frequency. This study provides a comprehensive approach for downstream flood risk management under the impacts of reservoirs.

**Acknowledgments**

This research is financially supported jointly by the National Natural Science Foundation of China (NSFC Grants 41890822 and 51525902), the Research Council of Norway (FRINATEK Project 274310), and the Ministry of Education "111 Project" Fund of China (B18037), all of which are greatly appreciated. We greatly appreciate the editor and the two reviewers for their insightful comments and constructive suggestions for improving the manuscript. No conflict of interest exists in the submission of the manuscript.

**Tables**

Table 1: Summary of the probability density functions, the corresponding moments, and the used link functions for nonstationary flood frequency analysis

| Distributions | Probability density functions | Moments | Link functions |
|---|---|---|---|
| Gamma (GA) | $f_Y\left(y\|\mu_t,\sigma_t\right)=\dfrac{(y)^{1/\sigma_t^2-1}}{\Gamma\left(1/\sigma_t^2\right)\left(\mu\sigma_t^2\right)^{1/\sigma_t^2}}\exp\left(-\dfrac{y}{\mu_t\sigma_t^2}\right)$
 $y>0,\mu_t>0,\sigma_t>0$ | $E(Y)=\mu_t$
 $Var(Y)=\mu_t^2\sigma_t^2$ | $g_1(\mu_t)=\ln(\mu_t)$
 $g_2(\sigma_t)=\ln(\sigma_t)$ |
| Weibull (WEI) | $f_Y\left(y\|\mu_t,\sigma_t\right)=\left(\dfrac{\sigma_t}{\mu_t}\right)\left(\dfrac{y}{\mu_t}\right)^{\sigma_t-1}\exp\left(-\left(\dfrac{y}{\mu_t}\right)^{\sigma_t}\right)$
 $y>0,\mu_t>0,\sigma_t>0$ | $E(Y)=\mu_t\Gamma\left(1+1/\sigma_t\right)$
 $Var(Y)=\mu_t^2\left[\Gamma\left(1+2/\sigma_t\right)-\Gamma^2\left(1+1/\sigma_t\right)\right]$ | $g_1(\mu_t)=\ln(\mu_t)$
 $g_2(\sigma_t)=\ln(\sigma_t)$ |
| Lognormal (LOGNO) | $f_Y\left(y\|\mu_t,\sigma_t\right)=\dfrac{1}{y\sigma_t\sqrt{2\pi}}\exp\left\{-\dfrac{\left[\log(y)-\mu_t\right]^2}{2\sigma_t^2}\right\}$
 $y>0,-\infty<\mu_t<\infty,\sigma_t>0$ | $E(Y)=w^{1/2}\exp(\mu_t)$
 $Var(Y)=w(w-1)\exp(2\mu_t)$
 $w=\exp(\sigma_t^2)$ | $g_1(\mu_t)=\ln(\mu_t)$
 $g_2(\sigma_t)=\ln(\sigma_t)$ |
| Gumbel (GU) | $f_Y\left(y\|\mu_t,\sigma_t\right)=\dfrac{1}{\sigma_t}\exp\left\{\left(\dfrac{y-\mu_t}{\sigma_t}\right)-\exp\left(\dfrac{y-\mu_t}{\sigma_t}\right)\right\}$
 $-\infty<y<\infty,-\infty<\mu_t<\infty,\sigma_t>0$ | $E(Y)=\mu_t-0.57722\sigma_t$
 $Var(Y)=\left(\pi^2/6\right)\sigma_t^2$ | $g_1(\mu_t)=\mu_t$
 $g_2(\sigma_t)=\ln(\sigma_t)$ |
| Generalized extreme value (GEV) | $f_Y\left(y\|\mu_t,\sigma_t,\xi\right)=\dfrac{1}{\sigma_t}\left[1+\xi\left(\dfrac{y-\mu_t}{\sigma_t}\right)\right]^{-1/\xi-1}\exp\left\{-\left[1+\xi\left(\dfrac{y-\mu_t}{\sigma_t}\right)\right]^{-1/\xi}\right\}$
 $y>\mu_t-\sigma_t/\xi,-\infty<\mu_t<\infty,\sigma_t>0,-\infty<\xi<\infty$ | $E(Y)=\mu_t-\dfrac{\sigma_t}{\xi}+\dfrac{\sigma_t}{\xi}\eta_1$
 $Var(Y)=\sigma_t^2\left(\eta_2-\eta_1^2\right)/\xi$
 $\eta_m=\Gamma\left(1-m\xi\right)$ | $g_1(\mu_t)=\mu_t$
 $g_2(\sigma_t)=\ln(\sigma_t)$ |

Table 2: Seven nonstationary scenarios for the formulas of the two distribution parameters (i.e.,

$\mu_t$ and $\sigma_t$ )

| Scenario classification | Scenario codes | The formula of distribution parameters | |
|---|---|---|---|
| | | $g_1(\mu_t)$ | $g_2(\sigma_t)$ |
| Stationary (S0) | S0 | $\alpha_0$ | $\beta_0$ |
| RI-dependent (S1) | S11 | $\alpha_0 + \alpha_1 RI$ | $\beta_0$ |
| | S12 | $\alpha_0$ | $\beta_0 + \beta_1 RI$ |
| | S13 | $\alpha_0 + \alpha_1 RI$ | $\beta_0 + \beta_1 RI$ |
| RRCI-dependent (S2) | S21 | $\alpha_0 + \alpha_1 RRCI$ | $\beta_0$ |
| | S22 | $\alpha_0$ | $\beta_0 + \beta_1 RRCI$ |
| | S23 | $\alpha_0 + \alpha_1 RRCI$ | $\beta_0 + \beta_1 RRCI$ |

Table 3: Information of the five major reservoirs in the Hanjiang River basin.

| Reservoirs | Longitude | Latitude | Area (km$^2$) | Year | Capacity ($10^9$ m$^3$) |
|---|---|---|---|---|---|
| Shiquan | 108.05 | 33.04 | 23,400 | 1974 | 0.566 |
| Ankang | 108.83 | 32.54 | 35,700 | 1992 | 3.21 |
| Huanglongtan | 110.53 | 32.68 | 10,688 | 1978 | 1.17 |
| Dangjiangkou | 111.51 | 32.54 | 95,220 | 1967 | 34.0 |
| Yahekou | 112.49 | 33.38 | 3030 | 1960 | 1.32 |

Table 4: Change in the mean and standard deviation of the AMDF after the construction of the two large reservoirs (Danjiangkou reservoir completed by 1967, and the Ankang reservoir built by

1992).

| Stations | Mean (m³/s) | | | Standard deviation (m³/s) | | |
|---|---|---|---|---|---|---|
| | 1956–1966 | 1967–1991 | 1992–2015 | 1956–1966 | 1967–1991 | 1992–2015 |
| AK | 9451 | 10,468 | 6506 | 4341 | 4623 | 4454 |
| HJG | 14,951 | 7524 | 4139 | 7896 | 5482 | 4074 |
| HZ | 16,603 | 10,120 | 5958 | 8833 | 5420 | 4721 |

Table 5: Correlation coefficients between the RRCI and the AMDF.

[revised manuscript text omitted]

*The model parameters in the optimal formulas are the posterior mean from the Bayesian inference.

Table 8: Summary of the rainfall information for the five largest floods after the construction (1967) of the Danjiangkou reservoir in the HZ station

| Year | Values (Ranking in 1967-2015) | | | | |
|------|------|------|------|------|------|
|      | AMDF [m³/s] | OR_JEP [-] | $I$ [mm] | $V$ [mm] | $T$ [day of the year] |
| 1983 | 25,600 (1) | 0.435 (2) | 20.2 (1) | 121.4 (19) | 281 (2) |
| 1975 | 19,900 (2) | 0.557 (7) | 9.6 (18) | 163.6 (13) | 277 (6) |
| 1974 | 18,200 (3) | 0.506 (4) | 12.0 (7) | 120.4 (20) | 278 (4) |
| 2005 | 16,800 (4) | 0.651 (11) | 8.2 (27) | 179.7 (10) | 278 (4) |
| 1984 | 16,100 (5) | 0.461 (3) | 9.9 (15) | 256.3 (4) | 273 (9) |

**Figures**

[Figure]

Figure 1: Flowchart of the nonstationary covariate-based flood frequency analysis using the rainfall-reservoir composite index (RRCI)

[Figure]

Figure 2: Relationship in Equation (2). (a) The contour plot of the RRCI against both the RI and the OR-JEP; and (b) is the function curves of the RRCI against the OR-JEP under different values of RI

[Figure]

Figure 3: Decomposition of a C-vine copula using four variables and three trees (denoted by T1,

T2, and T3)

[Figure]

Figure 4: Geographic location of the reservoirs, gauging stations, and rainfall stations along the

Hanjiang River.

[Figure]

Figure 5: Linear correlation between the five MARI variables and the AMDF for (a) the AK station, (b) the HJG station, and (c) the HZ station

[Figure]

Figure 6: Variation of the RI and the RRCI for (a) the AK station, (b) the HJG station, and (c)

the HZ station

[Figure]

Figure 7: Comparison of the stationary (S0), the RI-dependent (S1), and the RRCI-dependent (S2) scenarios of the same optimal distributions for (a) the AK station, (b) the HJG station, and (c) the

HZ station. The left panels (a1, b1, and c1) are the QQ plots for the entire AMDF series in each station.

The right panels (a2, b2, and c2) are the plots of the quantile residuals for the seven largest floods (their values and occurrence years have been listed) in each station, and the means of their quantile residuals (points) and the corresponding standard errors are indicated by the lines

[Figure]

Figure 8: Performance of (a) WEI_S23 for the AK station, (b) GA_S21 for the HJG station, and (c) WEI_S21 for the HZ station. The left panels (a1, b1, and c1) are the centile curves plots in each station (the 50th centile curves are indicated by the thick blue lines; the light gray-filled areas are between the 5th and 95th centile curves; the dark grey-filled areas are between the 25th and 75th centile curves; and the filled red points indicate the observed series). The right panels (a2, b2, and c2) are the worm plots. A reasonable model should have the plotted points within the 95% confidence intervals (between the two blue dashed curves)

[Figure]

Figure 9: Statistical inference of the 100-year return levels with a 95% uncertainty interval using the optimal RI-dependent and the RRCI-dependent distributions: (a) WEI_S13 and WEI_S23 for the

AK station, (b) GA_11 and GA_S21 for the HJG station, and (c) WEI_S11 and WEI_S21 for the HZ

station. In nonstationary case, the 95% credible interval in the t-year is calculated by a set of the (99th)

percentile estimations which are obtained by the flood distribution functions determined by the values of both covariate in that year and posterior parameter samples.